# Accelerating with FlyBrainLab the discovery of the functional logic of the *Drosophila* brain in the connectomic and synaptomic era

**Aurel A Lazar†\*, Tingkai Liu†, Mehmet Kerem Turkcan†, Yiyin Zhou†**

Department of Electrical Engineering, Columbia University, New York, United States

**Abstract** In recent years, a wealth of *Drosophila* neuroscience data have become available including cell type and connectome/synaptome datasets for both the larva and adult fly. To facilitate integration across data modalities and to accelerate the understanding of the functional logic of the fruit fly brain, we have developed FlyBrainLab, a unique open-source computing platform that integrates 3D exploration and visualization of diverse datasets with interactive exploration of the functional logic of modeled executable brain circuits. FlyBrainLab's User Interface, Utilities Libraries and Circuit Libraries bring together neuroanatomical, neurogenetic and electrophysiological datasets with computational models of different researchers for validation and comparison within the same platform. Seeking to transcend the limitations of the connectome/synaptome, FlyBrainLab also provides libraries for molecular transduction arising in sensory coding in vision/olfaction. Together with sensory neuron activity data, these libraries serve as entry points for the exploration, analysis, comparison, and evaluation of circuit functions of the fruit fly brain.

**\*For correspondence:**
aurel@ee.columbia.edu

†*The authors' names are listed in alphabetical order.*

**Competing interests:** The authors declare that no competing interests exist.

## Introduction

The era of connectomics/synaptomics ushered in the advent of large-scale availability of highly complex fruit fly brain data (*Chiang et al., 2011*; *Berck et al., 2016*; *Takemura et al., 2017a*; *Scheffer et al., 2020*), while simultaneously highlighting the dearth of computational tools with the speed and scale that can be effectively deployed to uncover the functional logic of fly brain circuits. In the early 2000's, automation tools introduced in computational genomics significantly accelerated the pace of gene discovery from the large amounts of genomic data. Likewise, there is a need to develop tightly integrated computing tools that automate the process of 3D exploration and visualization of fruit fly brain data with the interactive exploration of executable circuits. The fruit fly brain data considered here includes neuroanatomy, genetics, and neurophysiology datasets. Due to space limitations, we mostly focus here on exploring, analyzing, comparing, and evaluating executable circuits informed by wiring diagrams derived from neuroanatomy datasets currently available in public domain.

To meet this challenge, we have built an open-source interactive computing platform called FlyBrainLab. FlyBrainLab is uniquely positioned to accelerate the discovery of the functional logic of the *Drosophila* brain. It is designed with three main capabilities in mind: (1) 3D exploration and visualization of fruit fly brain data, (2) creation of executable circuits directly from the explored and visualized fly brain data in step (1), and (3) interactive exploration of the functional logic of the executable circuits devised in step (2) (see *Figure 1*).

To achieve tight integration of the three main capabilities sketched in *Figure 1* into a single working environment, FlyBrainLab integrates fly brain data in the NeuroArch Database (*Givon et al.,*

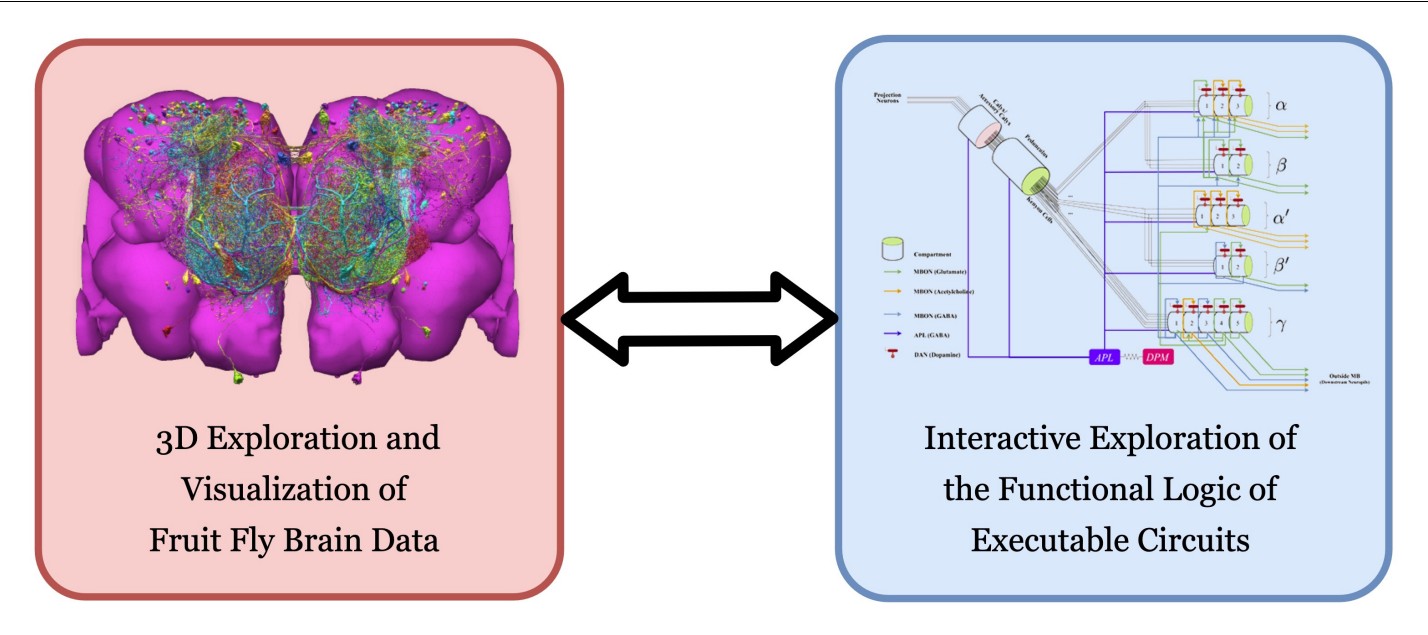

**Figure 1.** FlyBrainLab provides, within a single working environment, (left) 3D exploration and visualization of fruit fly brain data, and (right) creation of executable circuit diagrams from the explored and visualized circuit on the left followed by an interactive exploration of the functional logic of executable circuits.

*2015*) and provides circuit execution with the Neurokernel Execution Engine (*Givon and Lazar, 2016*) (see *Figure 2a*). The NeuroArch Database stores neuroanatomy datasets provided by for example, FlyCircuit (*Chiang et al., 2011*), Larva L1EM (*Ohyama et al., 2015*), the Medulla 7 Column (*Takemura et al., 2017a*) and Hemibrain (*Scheffer et al., 2020*), genetics datasets published by for example, FlightLight (*Jenett et al., 2012*) and FlyCircuit (*Chiang et al., 2011*), and neurophysiology datasets including the DoOR (*Münch and Galizia, 2016*) and our own in vivo recordings (*Lazar and Yeh, 2020*; *Kim et al., 2011*; *Kim et al., 2015*). The Neurokernel Execution Engine (see *Figure 2a*) supports the execution of fruit fly brain circuits on GPUs. Finally, the NeuroMynerva front-end exhibits an integrated 3D graphics user interface (GUI) and provides the user a unified view of data integration and computation (see *Figure 2a* (top) and *Figure 2b*). The FlyBrainLab software architecture is depicted in the *Appendix 1—figure 1*.

To accelerate the generation of executable circuits from fruit fly brain data, NeuroMynerva supports the following workflow.

First, the 3D GUI, called the NeuroNLP window (see *Figure 2b*, top middle-left), supports the visual exploration of fly brain data, including neuron morphology, synaptome, and connectome from all available data sources, stored in the NeuroArch Database (*Givon et al., 2015*). With plain English queries (see *Figure 2b*, top middle-left), a layperson can perform sophisticated database queries with only knowledge of fly brain neuroanatomy (*Ukani et al., 2019*).

Second, the circuit diagram GUI, called the NeuroGFX window (see *Figure 2b*, top middle-right) enables the interactive exploration of executable circuits stored in the NeuroArch Database. By retrieving tightly integrated biological data and executable circuit models from the NeuroArch Database, NeuroMynerva supports the interaction and interoperability between the biological circuit (or pathway for short) built for morphological visualization and the executable circuit created and represented as an interactive circuit diagram, and allows them to build on each other. This helps circuit developers to more readily identify the modeling assumptions and the relationship between neuroanatomy, neurocircuitry, and neurocomputation.

Third, the GUIs can operate in tandem with command execution in Jupyter notebooks (see also *Figure 2b*, bottom center). Consequently, fly brain pathways and circuit diagrams can be equivalently processed using API calls from Python, thereby ensuring the reproducibility of the exploration of similar datasets with minimal modifications. The Neurokernel Execution Engine (*Givon and Lazar,*

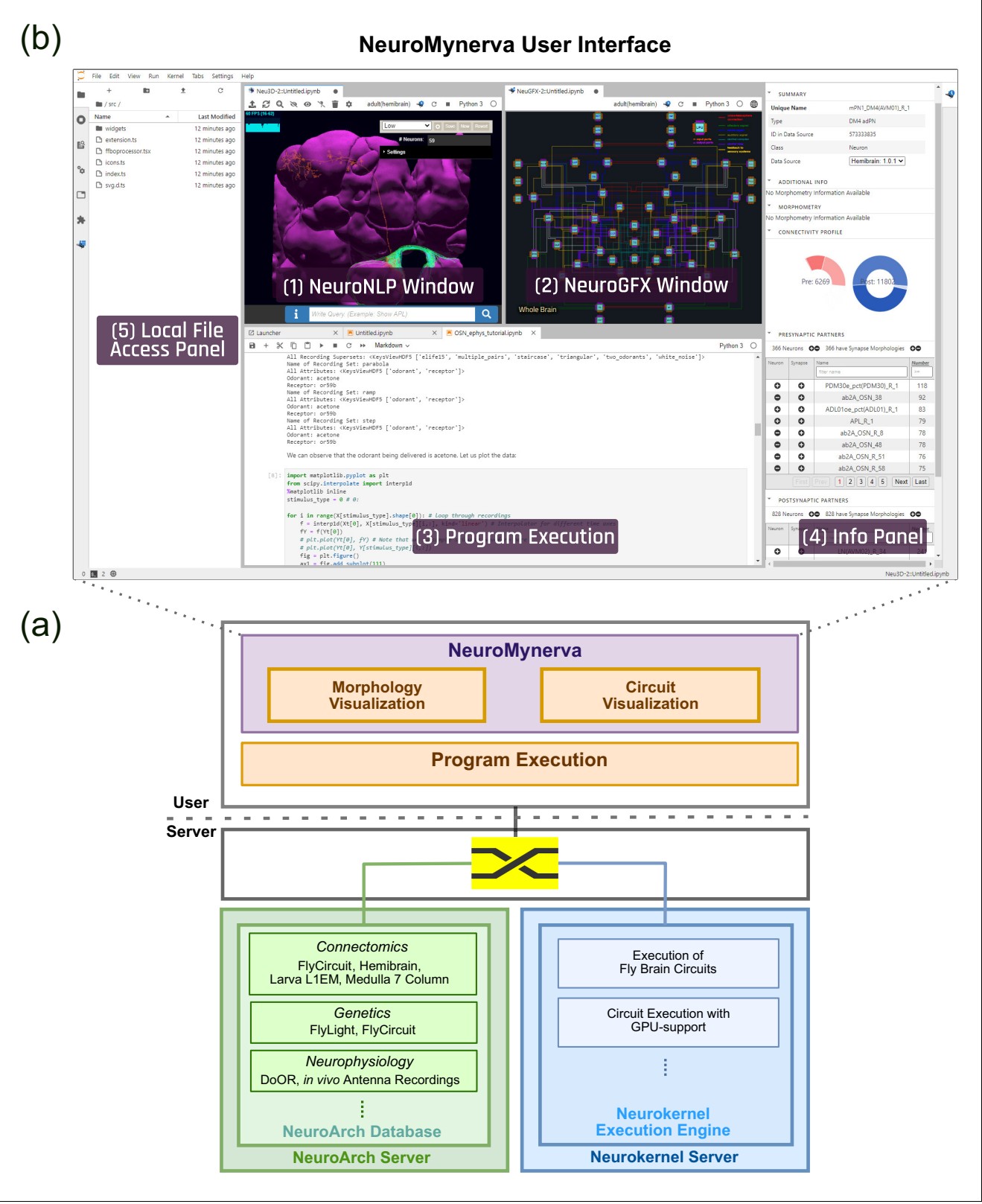

**Figure 2.** The software architecture and the user interface of FlyBrainLab. (a) The main components of the architecture of FlyBrainLab: (top) NeuroMynerva user-side frontend, (bottom left) NeuroArch Database for storage of fruit fly brain data and executable circuits, (bottom right) Neurokernel Execution Engine for execution of fruit fly brain circuits on GPUs (see also *Appendix 1—figure 1* for a schematic diagram of the FlyBrainLab software architecture). (b) NeuroMynerva User Interface. The UI typically consists five blocks, including a (1) NeuroNLP 3D Visualization

*Figure 2 continued on next page*

*Figure 2 continued*

Window with a search bar for NLP queries, providing capabilities for displaying and interacting with fly brain data such as the morphology of neurons and position of synapses. (2) NeuroGFX Executable Circuits Window, for exploring executable neural circuits with interactive circuit diagrams. (3) Program Execution Window with a built-in Jupyter notebook, executing any Python code including calls to the FlyBrainLab Client (see also Appendix 1.2), for direct access to database queries, visualization, and circuit execution, (4) Info Panel displaying details of highlighted neurons including the origin of data, genetic information, morphometric statistics and synaptic partners, etc. (5) Local File Access Panel with a built-in Jupyter file browser for accessing local files.

*2016*) provides circuit execution on multiple computing nodes/GPUs. The tight integration in the database also allows the execution engine to fetch executable circuits directly from the NeuroArch Database. The tight integration between NeuroArch and Neurokernel is reinforced and made user transparent by NeuroMynerva.

Exploration, analysis, execution, comparison, and evaluation of circuit models, either among versions developed by one's own, or among those published in literature, are often critical steps toward discovering the functional logic of brain circuits. Six types of explorations, analyses, comparisons, and evaluations are of particular interest. First, build and explore the structure of fly brain circuits with English queries (Use Case 1). Second, explore the structure and function of yet to be discovered brain circuits (Use Case 2). Third, interactively explore executable circuit models (Use Case 3). Fourth, starting from a given dataset and after implementing a number of circuit models published in the literature, analyze and compare these under the same evaluation criteria (Use Case 4). Fifth, automate the construction of executable circuit models from datasets gathered by different labs and analyze, compare, and evaluate the different circuit realizations (Use Case 5). Sixth, analyze, compare, and evaluate fruit fly brain circuit models at different developmental stages (Use Case 6).

In what follows, we present results, supported by the FlyBrainLab Circuits Libraries (see Materials and methods), demonstrating the comprehensive exploration, execution, analysis, comparison, and evaluation capability of FlyBrainLab. While our emphasis here is on building executable circuit models informed by the connectome/synaptome of the fruit fly brain, these libraries together with sensory neuron activity data serve as entry points for an in-depth exploration, execution, analysis, comparison, and evaluation of the functional logic of the fruit fly brain.

## Results

### Use Case 1: building fly brain circuits with english queries

FlyBrainLab is equipped with a powerful and versatile user interface to build fruit fly brain circuits from connectome and synaptome datasets. The interface is designed to accommodate users with widely different expertise, such as neurobiologists, computational neuroscientists or even college or high school students. Knowledge of the nomenclature of the fruit fly brain is assumed.

The simplest way to build a fly brain circuit is via the NeuroNLP natural language query interface (*Ukani et al., 2019*). By specifying in plain English cell types, synaptic distribution, pre- and post-synaptic partners, neurotransmitter types, etc, neurons and synapses can be visualized in the NeuroNLP window (see also *Figure 2b*).

The motion detection pathway in the fruit fly Medulla has been, in part, mapped out thanks to the Medulla 7 Column dataset (*Takemura et al., 2015*). While much research has focussed on the direct, feedforward pathway feeding into the motion sensitive T4 neurons (*Takemura et al., 2017b*; *Haag et al., 2017*), the contribution of the feedback pathways and the neighboring columnar neurons to the motion detection circuit has largely been ignored. To study the circuit that mediates the lateral feedback into the motion detection pathway, we used English queries to quickly visualize the neurons involved. Starting from a T4a neuron in the 'home' column that is sensitive to front-to-back-motion (*Maisak et al., 2013* 'show T4a in column home'; 'color lime'), we queried its presynaptic neurons ('add presynaptic neurons'; 'color gray') as well as their presynaptic neurons that are non-columnar, in particular, the Dm and Pm cells (*Fischbach and Dittrich, 1989*) ('add presynaptic $Dm$ neurons with at least five synapses'; 'color cyan'; 'add presynaptic $Pm$ neurons with at least five synapses'; 'color yellow'). The resulting visualization of the circuit is depicted in *Figure 3 (a1)*, with

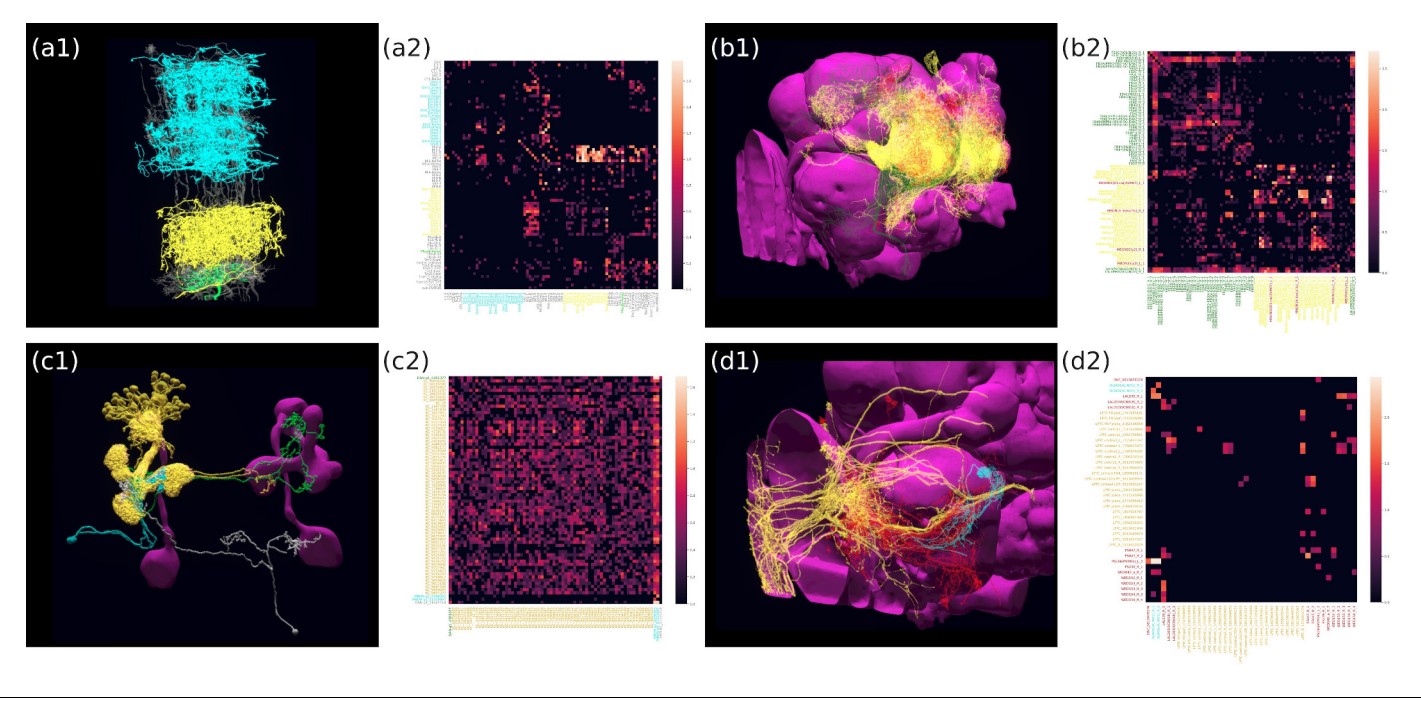

**Figure 3.** Building fly brain circuits with English queries. (**a1**) Lateral feedback pathways in the visual motion detection circuit. (green) a T4a neuron, (gray) neurons presynaptic to the T4a neuron, (cyan) glutamatergic and GABAergic Dm neurons that are presynaptic to the neurons in gray, (yellow) Pm neurons that are presynaptic to the neurons in gray. (**a2**) Connectivity matrix of the pathways in (**a1**). (**b1**) Pathways between MBONs and neurons innervating FB layer 3. (yellow) MBONs that are presynaptic to neurons that have outputs in FB layer 3. (green) Neurons that have outputs in FB layer 3 that are postsynaptic to the MBONs in yellow. (red) MBONs postsynaptic to neurons in green. (**b2**) Connectivity matrix of the pathways in (**b1**). (**c1**) The pathways of the g compartment of the larva fruit fly. (cyan) g compartment MBONs, (yellow) KCs presynaptic to the g compartment MBONs, (green) a DAN presynaptic to the g compartment MBONs, (white) an OAN presynaptic to the g compartment MBONs, (**c2**) Connectivity matrix of the pathways in (**c1**). (**d1**) Pathways between LPTCs and a potential translational motion-sensitive neuron GLN. (yellow) LPTCs, (cyan) GLNs, (red) neurons that form the path between LPTCs to GLNs. (**d2**) Connectivity matrix of the pathways in (**d1**). color bar in a2, b2, c2, and d2: $log_{10}(N+1)$, where $N$ is the number of synapses between 2 neurons. (**a1**)–(**d1**) are screenshots downloaded from the NeuroNLP Window. The sequence of queries that generates these visualizations is listed in Materials and methods Use Case 1.

neurons mediating cross-columnar interaction highlighted. The retrieved connectivity matrix is shown in *Figure 3 (a2)* (see also Materials and methods, Use Case 1).

The mushroom body (MB) has been known to be an associative olfactory memory center (*Modi et al., 2020*), whereas the fan-shaped body (FB) shown to be involved in visual pattern memory (*Liu et al., 2006*). Recently, it has been shown that the Kenyon cells in the MB also receive visual inputs (*Li et al., 2020a*), and that the MB and FB are interconnected (*Li et al., 2020b*). The pathway between the MB and the FB, or a particular layer in the FB can be easily visualized using NeuroNLP. We used English queries to establish and visualize the circuit that directly connects the MB with the layer 3 of the FB in the Hemibrain dataset, as depicted in *Figure 3 (b1)*. The connectivity matrix is shown in *Figure 3 (b2)* (see also Materials and methods, Use Case 1, for the sequence of queries that created this visualization).

Natural language queries supplemented by the NeuroNLP 3D interface and the Info Panel (see also *Figure 2b*) enable us to inspect, add and remove neurons/synapses. For example, in *Figure 3 (c1)*, we built a simple circuit around the g compartment of the mushroom body (*Saumweber et al., 2018*) of the Larva L1EM dataset (*Ohyama et al., 2015*) starting from the MBONs that innervate it. We then inspected these MBONs in the Info Panel and added all KCs presynaptic to each of them by filtering the name 'KC' in the presynaptic partner list. Similarly, we added the dopaminergic neurons (DANs) and octopaminergic neurons (OANs) presynaptic to these MBONs. *Figure 3 (c2)* depicts the connectivity matrix of this MB circuit (see also Materials and methods, Use Case 1, for the full sequence of queries/operations that created this visualization).

The FlyBrainLab UI provides users a powerful yet intuitive tool for building fly brain circuits at any scale, requiring no knowledge of programming or the data model of the underlying NeuroArch Database. For more advanced users, FlyBrainLab also exposes the full power of NeuroArch API for directly querying the NeuroArch database using the NeuroArch JSON format. Utilizing this capability, we built a circuit pathway that potentially carries translational visual motion information into the Noduli (NO) in *Figure 3 (d1)*. The search for this circuit was motivated by a type of cells in honey bees, called TN neurons, that are sensitive to translational visual motion and provide inputs to the NO (*Stone et al., 2017*). In the Hemibrain dataset, a cell type 'GLN' resembles the TN neurons in the honey bee and is potentially a homolog in the fruit fly. We therefore asked if there exists a pathway to these neurons from visual output neurons that are sensitive to wide-field motion, in particular, the lobula plate tangential cells (LPTCs). Using a NeuroArch query, we found all paths between LPTCs and GLNs that are less than three hops and have at least five synapses in each hop (see also Materials and methods), Use Case 1, for the complete listing of the invoked NeuroArch JSON query. Only the HS cells and H2 cells, but not CH and VS cells (*Hausen, 1984*) have robust paths to the GLNs. The connectivity of this circuit is shown in *Figure 3(d2)* (see also Materials and methods, Use Case 1).

## Use Case 2: exploring the structure and function of yet to be discovered brain circuits

Here, we further demonstrate the capabilities of FlyBrainLab in the quest of exploring the structure and function of yet to be discovered fly brain circuits. In particular, we demonstrate several use cases of the Utility Libraries (see Appendix 2) and their interaction with the rest of the FlyBrainLab components.

In the first example, we explore the structure of densely-connected brain circuits in the Hemibrain dataset. Such an exploration is often the starting point in the quest of understanding the function of a brain circuit without any prior knowledge of neuropil boundaries, or the identity of each neuron (see also Materials and methods, Use Case 2). By invoking the NeuroGraph Library on the Hemibrain dataset (see Appendix 2), we extracted eight densely connected neuron groups, as shown in *Figure 4a*. We then visualized subsets of neurons pseudocolored by group membership as shown in *Figure 4b* and assigned six of the eight groups to several known brain regions/neuropils. These neuropils include the AL, the MB, the lateral horn (LH), the central complex (CX), the anterior ventrolateral protocerebrum (AVLP), and the superior protocerebrum (SP). The remaining two brain regions correspond to the anterior optic tubercle with additional neurons of the posterior brain (AOTUP) and the anterior central brain (ACB). A circuit diagram depicting the connections between these groups of neurons is shown in *Figure 4c*.

In the second example, we sought to define cell types not just by visually inspecting the morphology of the neurons, but also by taking into account the underlying graph structure of the circuit pathways. This is useful when a new dataset is released without explicit definitions of cell types and/ or when there is a need for refining such definitions. Here, to automatically analyze neuron cell types in the VA1v glomerulus dataset (*Horne et al., 2018*), we applied the Adjacency Spectral Embedding algorithm (*Sussman et al., 2012*) of the NeuroGraph library (see Appendix 2 and Materials and methods, Use Case 2). The embedding is visualized using UMAP (*McInnes et al., 2018*) and depicted in *Figure 4d*, and it is validated by annotations from the original dataset. We note that the overlap between PNs and some LNs is due to the restricted volume of the traced tissue. For an additional adjustment of their cell-type assignment, the resulting clusters of neurons can be further visually inspected as shown in *Figure 4e*. Outliers that lie far away from their clusters may guide future inquiries into cell types that have not been previously described or provide new descriptions for existing cell types contingent on their connectivity. Finding new neuron subtypes, for example, LNs that cluster with OSNs or neurons that cluster with LNs can be further investigated. Finally, a circuit diagram can be automatically generated using the NeuroGraph Library, as shown in *Figure 4f*.

Lastly, we demonstrate the process of automatic circuit diagram generation of explored brain circuits. Here, we explored the lateral horn subcircuit downstream of the V glomerulus projection neurons, as well as the neuropils that the lateral horn output neurons (LHONs) project to *Varela et al., 2019*. The circuit can be easily specified and visualized by NeuroNLP queries (see Materials and methods, Use Case 2), and individual neurons can be further added/removed from the GUI. The resulting circuit is depicted in *Figure 4h*. We then inspected the innervation pattern of each neuron,

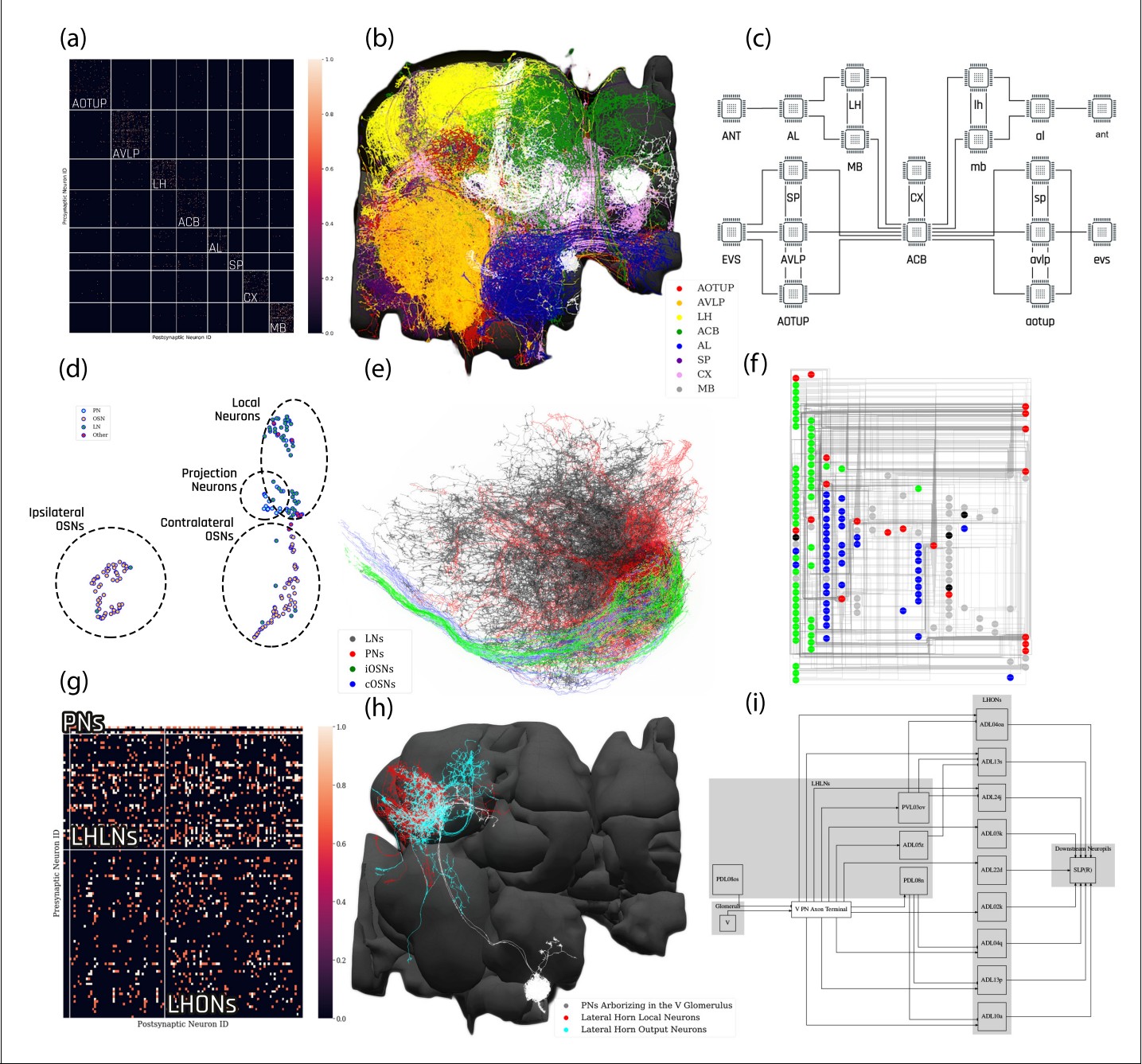

**Figure 4.** Exploratory analysis of the fly brain circuits. (**a**) Louvain algorithm applied to all neurons in the Hemibrain dataset showing eight groups of densely connected neurons. Color indicates the value of $\log_{10}(n + 1)$, where $n$ is the number of synapses; values larger than one are shown in the same color as value 1. AOTUP: anterior optic tubercle with additional neurons of the posterior brain, AVLP: anterior ventrolateral protocerebrum, LH: lateral horn, ACB: neurons in the anterior central brain, AL: antennal lobe, SP: superior protocerebrum, CX: central complex, MB: mushroom body. Labels were added after visually inspecting the neurons in each group of neurons in (**b**). (**b**) A subset of neurons pseudo-colored according to the group they belong to in (**a**). (**c**) A brain-level circuit diagram created by hand according to the grouping of neurons and the inter-group edge information obtained in (**a**). Visual and olfactory inputs from, respectively, the early visual system (EVS) and antenna (ANT) were added. Groups in the left hemisphere were added by symmetry. (**d**) Adjacency Spectral Embedding algorithm applied to the VA1v connectome dataset using the NeuroGraph library. The color of each circle indicates the cell-type labeling from the original dataset. Groups of neurons labeled by dashed circles are based on validated cell types. (**e**) Visualization of neurons analyzed in (**d**). Neuron colors were assigned according to the groups in (**d**). (**f**) A circuit diagram of the VA1v circuit analyzed in (**d**) automatically generated by the NeuroGraph Library. (**g**) Connectivity matrix of the lateral horn neurons downstream the V glomerulus projection neurons of the antennal lobe. Colorbar configured in the same way as in (**a**). (**h**) Morphology of the neurons in (**g**). (white) PNs arborizing in the V

*Figure 4 continued on next page*

Figure 4 continued

glomerulus, (red) LHLNs, (cyan) LHONs. (i) A circuit diagram automatically generated by the circuit visualization utilities of NeuroGraph starting with the circuit in (g) and (h), and the superior lateral protocerebrum (SLP), the primary neuropil that the LHONs project to.

either visually, or by querying its arborization data from the NeuroArch Database, and classified it either as a lateral horn local neuron (LHLN) or a LHON. The connectivity of neurons of the resulting circuit is shown in *Figure 4g*, where the rows and columns are grouped by neuron type. Using this collection of information, we invoked the NeuroGraph Library to create the circuit diagram shown in *Figure 4i* (see also Materials and methods, Use Case 2). The circuit diagram can then be used for computational studies as outlined in the previous examples.

## Use Case 3: interactive exploration of executable fruit fly brain circuit models

Beyond exploring the structure of fruit fly brain circuits, a primary objective supported by FlyBrain-Lab is the study of the function of executable circuits constructed from fly brain data. FlyBrainLab provides users with rapid access to executable circuits stored on the NeuroArch Database. During program execution, these circuits can also be directly accessed by the Neurokernel Execution Engine.

In *Figure 5a*, we depict the pathways of a cartridge of the Lamina neuropil (*Rivera-Alba et al., 2011*) visualized in the NeuroNLP window. The circuit diagram modeling the cartridge visualized in the NeuroGFX window is shown in *Figure 5b*. With proper labels assigned to blocks/lines representing the neurons and synapses, we made the circuit diagram interactive. For example, by clicking on the block representing a neuron, the neuron can be inactivated, an operation corresponding to the silencing/ablating a neuron in the fly brain. *Figure 5d* depicts a modified cartridge circuit in which several neurons have been silenced. As a result, the visualized neural pathways in the NeuroNLP window automatically reflect these changes, as shown in *Figure 5c*. The circuit components can also be disabled/reenabled by selecting through hiding/displaying visualized neurons in the NeuroNLP window.

In the same interactive diagram, models of the circuit components and their parameters can be viewed/specified from a Model Library with all the available model implementations in the Neurokernel Execution Engine. In addition to these simple interactive operations further detailed in Materials and methods, Use Case 3, FlyBrainLab APIs support bulk operations such as updating models and parameters of an arbitrary number of circuit components (see also Appendix 4).

## Use Case 4: analyzing, evaluating, and comparing circuit models of the fruit fly central complex

We first demonstrate the workflow supported by FlyBrainLab for analyzing, evaluating and comparing circuit models of the fruit fly Central Complex (CX) based on the FlyCircuit dataset (*Chiang et al., 2011*). The circuit connecting the ellipsoid body (EB) and the protocerebral bridge (PB) in the CX has been shown to exhibit ring attractor dynamics (*Seelig and Jayaraman, 2015*; *Kim et al., 2017*; *Skaggs et al., 1995*). Recently, a number of researchers investigated circuit mechanisms underlying these dynamics. Here, we developed a CXcircuits Library for analyzing, evaluating and comparing various CX circuit realizations. Specifically, we implemented three of the circuit models published in the literature, called here model A (*Givon et al., 2017*), model B (*Kakaria and de Bivort, 2017*), and model C (*Su et al., 2017*), and compared them in the same FlyBrainLab programming environment.

In *Figure 6 (a1, b1, c1)*, the anatomy of the neuronal circuits considered in models A, B, and C is depicted, respectively. The corresponding interactive circuit diagram is shown in *Figure 6 (a2, b2, c2)*. Here, model A provides the most complete interactive CX circuit, including the core subcircuits for characterizing the PB-EB interaction with the EB-LAL-PB, PB-EB-LAL, PB-EB-NO, PB local, and EB ring neurons (see Materials and methods, Use Case 4, and *Givon et al., 2017* for commonly used synonyms). Models B and C exhibit different subsets of the core PB-EB interaction circuit in model A. While no ring neurons are modeled in model B, PB local neurons are omitted in model C. They,

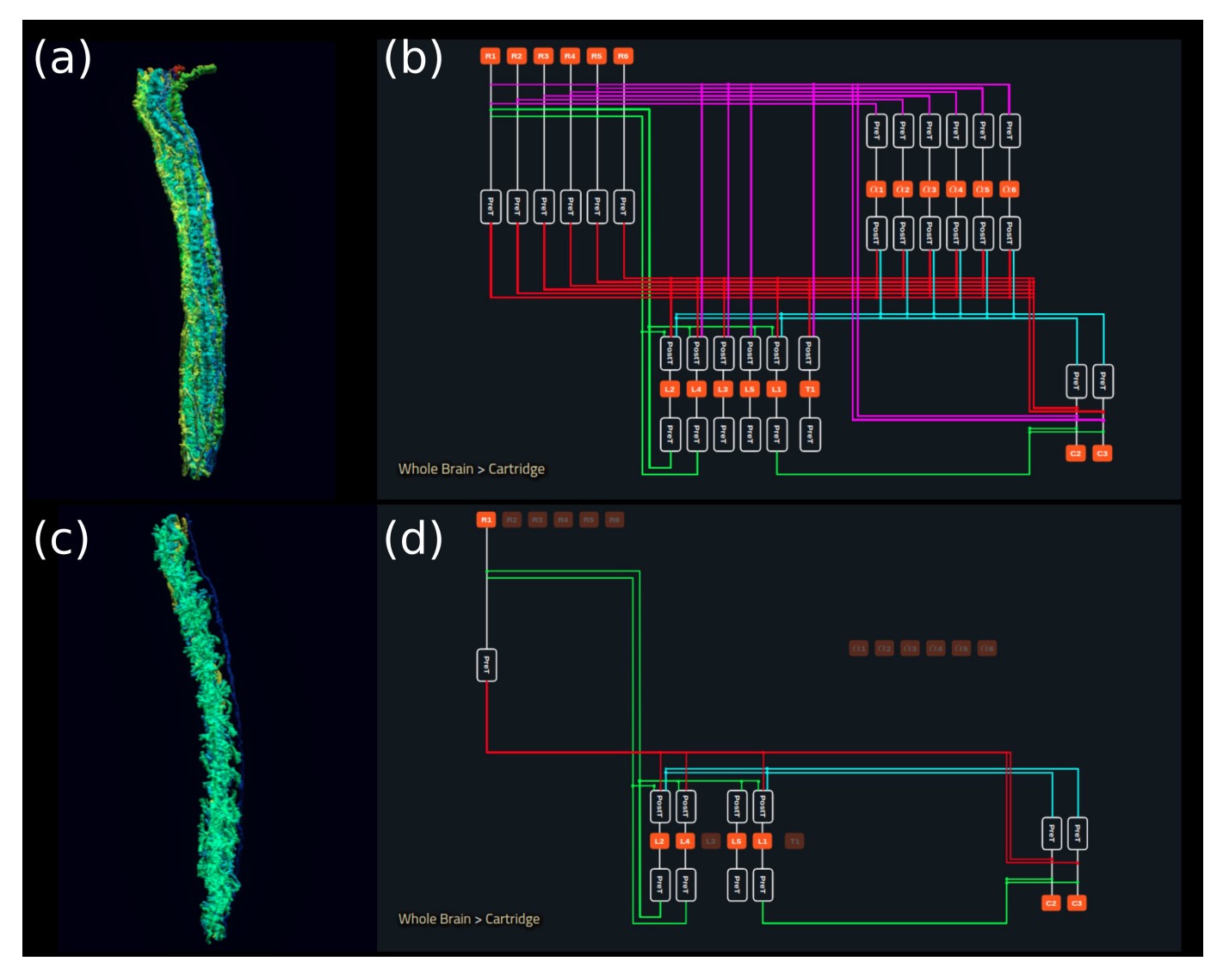

**Figure 5.** Interactive exploration of executable circuit models. (**a**) The pathways of a Lamina cartridge visualized in the NeuroNLP window. (**b**) A circuit diagram of the cartridge in (**a**) displayed in NeuroGFX window. (**c**) The cartridge pathways modified interactively using the circuit diagram in (**b**) that results in the circuit diagram in (**d**). (**d**) The circuit diagram modeling the chosen pathways in (**c**).

however, do not model other neurons in the CX, for example, those that innervate the Fan-shaped Body (FB).

In *Video 1*, we demonstrate the interactive capabilities of the three models side-by-side, including the visualization of the morphology of CX neurons and the corresponding executable circuits, user interaction with the circuit diagram revealing connectivity pattern, and the execution of the circuit. In the video, the visual stimulus depicted in *Figure 6 (d3)* was presented to three models (see Materials and methods, Use Case 4, for the details of generating the input stimulus for each model). The responses, measured as the mean firing rate of EB-LAL-PB neurons within contiguous EB wedges, are shown in *Figure 6 (a4, b4, c4)*, respectively. Insets depict the responses at 10, 20, and 30 s. During the first second, a moving bar in its fixed initial position and a static bar are presented. The moving bar displays a higher brightness than the static bar. All three models exhibited a single-bump (slightly delayed) response tracking the position of the moving bar. The widths of the bumps were different, however. After 30 s, the moving bar disappeared and models A and B shifted to track the location of the static bar, whereas the bump in model C persisted in the same position

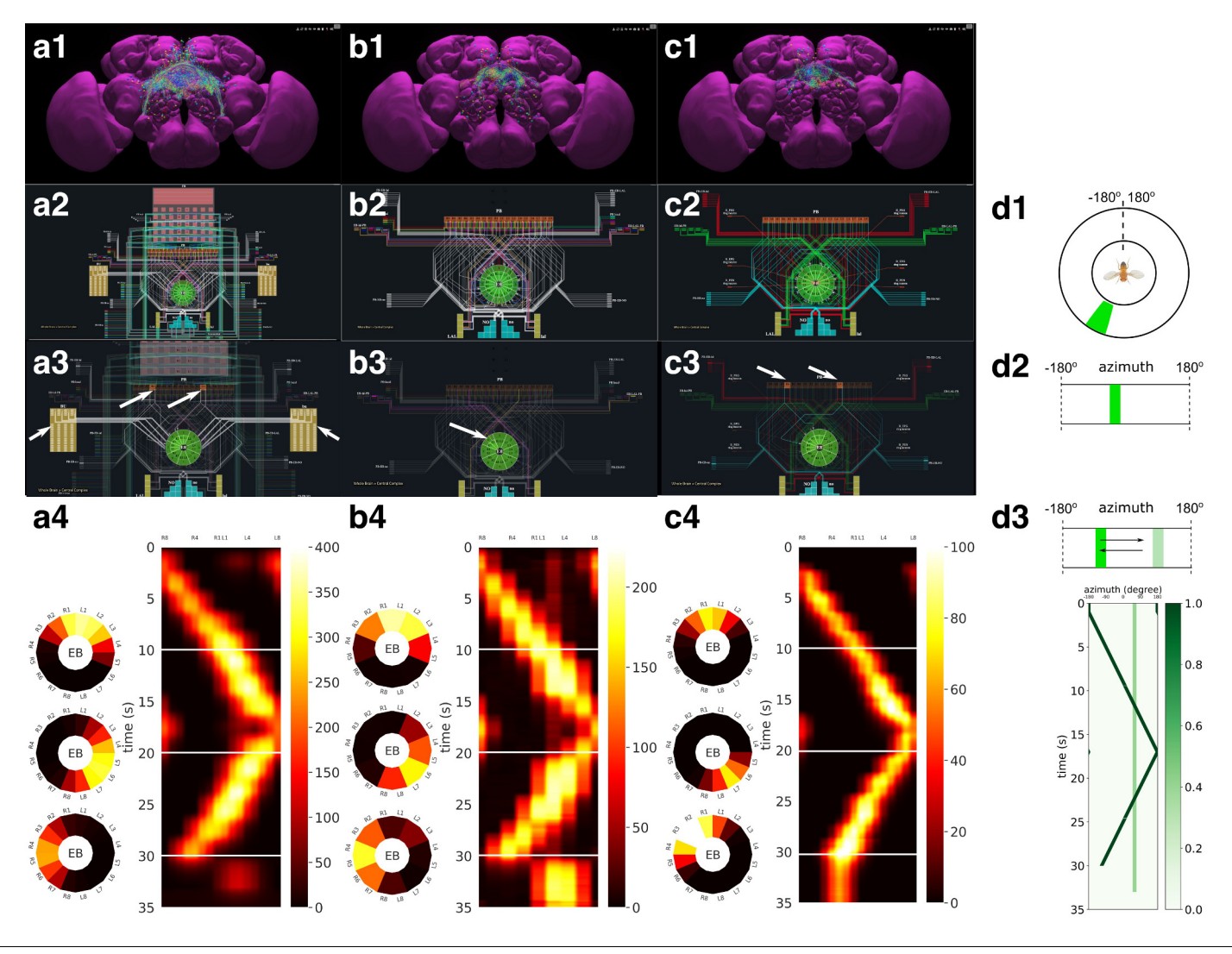

**Figure 6.** Analysis, evaluation and comparison of three models of CX published in the literature. (a1–a4) Model A (*Givon et al., 2017*), (b1–b4) Model B (*Kakaria and de Bivort, 2017*), (c1–c4) Model C (*Su et al., 2017*). (a1, b1, c1) Morphology of the neurons visualized in the NeuroNLP window (see *Figure 2b*). Displayed number of neurons in: (a1) 366, (a2) 87, (a3) 54. (a2, b2, c2) Neuronal circuits in the NeuroNLP window depicted in the NeuroGFX window (see *Figure 2b*) as abstract interactive circuit diagrams. The naming of the ring neurons in (c2) follows *Su et al., 2017*. Number of neurons in the diagram: (b1) 348, (b2) 60, (b3) 56. As the FlyCircuit dataset contains duplicates, some neurons in the diagrams may correspond to multiple neurons in the dataset and some do not have correspondences due to the lack of morphology data. (a3, b3, c3) When a single vertical bar is presented in the visual field (d1/d2), different sets of neurons/subregions (highlighted) in each of the models, respectively, receive either current injections or external spike inputs. (a4, b4, c4) The mean firing rates of the EB-LAL-PB neurons innervating each of the EB wedges of the three models (see Materials and methods, Use Case 4), in response to the stimulus shown in (d3). Insets show the rates at 10, 20, and 30 s, respectively, overlaid onto the EB ring. (d1) A schematic of the visual field surrounding the fly. (d2) The visual field flattened. (d3) Input stimulus consisting of a bar moving back and forth across the screen, and a second fixed bar at 60° and with lower brightness.

where the moving bar disappeared. Furthermore, for models B and C but not for model A, the bumps persisted after the removal of the visual stimulus (after 33 s), as previously observed in vivo (*Seelig and Jayaraman, 2015*; *Kim et al., 2017*).

By comparing these circuit models, we notice that, to achieve the ring attractor dynamics, it is critical to include global inhibitory neurons, for example, PB local neurons in models A and B, and ring neurons in models A and C. The model A ring neurons featuring a different receptive field and the ring neurons in model C receiving spike train input play a similar functional role. However, to achieve the ring attractor dynamics characterized by a single bump response to multiple bars and

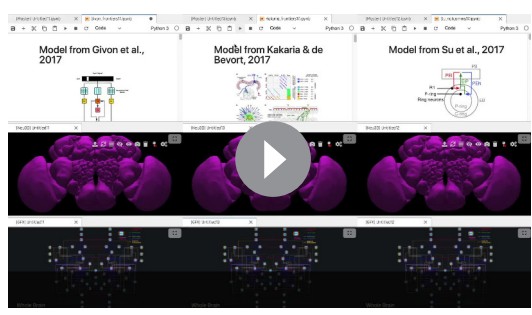

**Video 1.** Running three CX executable circuits in the FlyBrainLab. (left) Model A (*Figure 6a*). (middle) Model B (*Figure 6b*). (right) Model C (*Figure 6c*).
https://elifesciences.org/articles/62362#video1

persistent bump activity after the removal of the vertical bar, model C only required three out of the five core neuron types (see Materials and methods, Use Case 4), whereas model B requires all four neuron types included.

## Use Case 5: analyzing, evaluating, and comparing adult antenna and antennal lobe circuit models based upon the FlyCircuit and hemibrain datasets

In the second example, we demonstrate the effect on modeling the antenna and antennal lobe circuits due to, respectively, the FlyCircuit (*Chiang et al., 2011*) and the Hemibrain (*Scheffer et al., 2020*) datasets (see also Materials and methods, Use Case 5).

We start by exploring and analyzing the morphology and connectome of the olfactory sensory neurons (OSNs), antennal lobe projection neurons (PNs), and local neurons (LNs) of the FlyCircuit (*Chiang et al., 2011*) and the Hemibrain (*Scheffer et al., 2020*) datasets (see *Figure 7a*). Compared with the antennal lobe data in the FlyCircuit dataset, the Hemibrain dataset reveals additional connectivity details between OSNs, PNs, and LNs that we took into account when modeling the antennal lobe circuit (see Materials and methods, Use Case 5). Following (*Lazar et al., 2020a*), we first constructed the two layer circuit based on the FlyCircuit dataset shown in *Figure 7b* (left) and then constructed a more extensive connectome/synaptome model of the adult antennal lobe based on the Hemibrain dataset as shown in *Figure 7b* (right).

Execution of and comparison of the results of these two circuit models show quantitatively different PN output activity in steady-state (*Figure 7c*) and for transients (*Figure 7d*). A prediction (*Lazar and Yeh, 2019*; *Lazar et al., 2020a*) made by the antenna and antennal lobe circuit shown in *Figure 7b* (left) using the FlyCircuit data has been that the PN activity, bundled according to the source glomerulus, is proportional to the vector characterizing the affinity of the odorant-receptor pairs (*Figure 7c*, left column).

The transient and the steady state activity response are further highlighted in *Figure 7d* for different amplitudes of the odorant stimulus waveforms. The initial results show that the circuit on the right detects with added emphasis the beginning and the end of the odorant waveforms.

The complex connectivity between OSNs, LNs, and PNs revealed by the Hemibrain dataset suggests that the adult antennal lobe circuit encodes additional odorant representation features (*Scheffer et al., 2020*).

## Use Case 6: analyzing, evaluating, and comparing early olfactory circuit models of the larva and the adult fruit flies

In the third example, we investigate the difference in odorant encoding and processing in the *Drosophila* Early Olfactory System (EOS) at two different developmental stages, the adult and larva (see also Materials and methods, Use Case 6).

We start by exploring and analyzing the morphology and connectome for the Olfactory Sensory Neurons (OSNs), Antennal Lobe Projection Neurons (PNs) and Local Neurons (LNs) of the adult Hemibrain (*Scheffer et al., 2020*) dataset and the LarvaEM (*Berck et al., 2016*) dataset (see *Figure 8a*).

Detailed connectivity data informed the construction of the model for both the adult and larva EOS, that we developed based on parameterized versions of the previous literature (*Lazar et al., 2020a*). In particular, the larval model includes fewer number of OSNs, PNs, and LNs in Antenna and Antennal Lobe circuit as shown in *Figure 8b* right.

The adult and larval EOS models were simultaneously evaluated on a collection of mono-molecular odorants whose binding affinities to odorant receptors have been estimated from physiological recordings (see also Materials and methods Use Case 6). In *Figure 8c* (left), the affinity values are

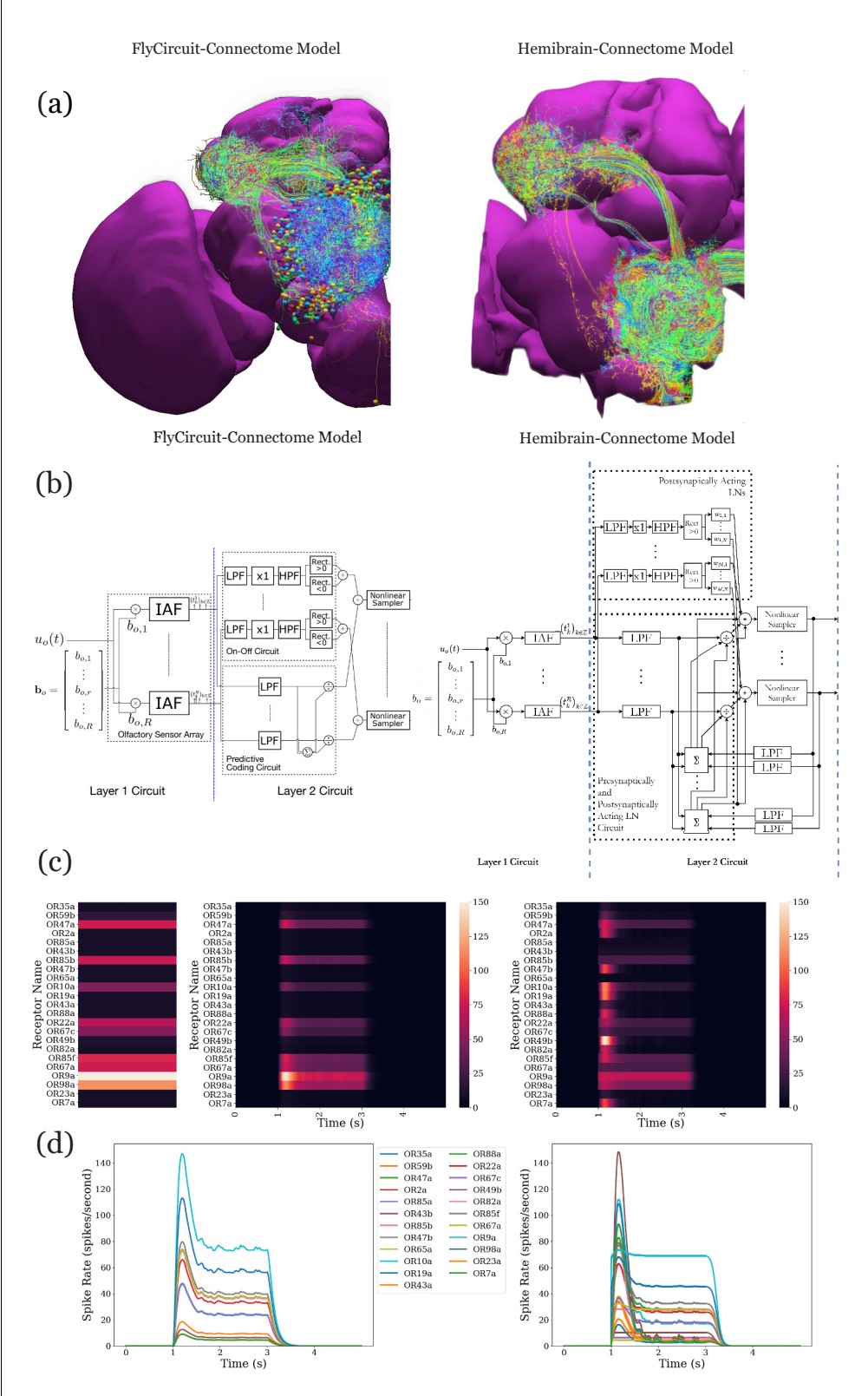

**Figure 7.** Analysis, evaluation, and comparison between two models of the antenna and antennal lobe circuit of the adult fly based on the FlyCircuit (left) dataset (*Chiang et al., 2011*) and an exploratory model based on the Hemibrain (right) dataset (*Scheffer et al., 2020*). (a) Morphology of olfactory sensory neurons, local neurons, and projection neurons in the antennal lobe for the two datasets. The axons of the projection neurons and their projections to the mushroom body and lateral horn are also visible. (b) Circuit diagrams depicting the antenna and antennal lobe circuit motifs derived

*Figure 7 continued on next page*

*Figure 7 continued*

from the two datasets. (c) Response of the antenna/antennal lobe circuit to a constant ammonium hydroxide step input applied between 1 s and 3 s of a 5 s simulation; (left) the interaction between the odorant and 23 olfactory receptors is captured as the vector of affinity values; (middle and right) a heatmap of the uniglomerular PN PSTH values (spikes/second) grouped by glomerulus for the two circuit models. (d) The PN response transients of the two circuit models for uniform noise input with a minimum of 0ppm and a maximum of 100 ppm preprocessed with a 30 Hz low-pass filter (*Kim et al., 2011*) and delivered between 1 s and 3 s.

shown for the odorant receptors that are only in the adult fruit fly (top panel), that appear in both the adult and the larva (middle panel) and, finally, that are only in the larva. The steady-state responses of the Antenna and Antennal Lobe circuit for both models are computed and shown in *Figure 8c* (middle and right, respectively). Visualized in juxtaposition alongside the corresponding affinity vectors, we observe a stark contrast in odorant representation at all layers of the circuit between adult and larva, raising the question of how downstream circuits can process differently represented odorant identities and instruct similar olfactory behavior across development. Settling such questions requires additional physiological recordings, that may improve the accuracy of the current FlyBrainLab EOS circuit models.

## Discussion

Historically, a large number of visualization and computational tools have been developed primarily designed for either neurobiological studies (see *Figure 1* (left)) or computational studies (see *Figure 1* (right)). These are briefly discussed below.

The computational neuroscience community has invested a significant amount of effort in developing tools for analyzing and evaluating model neural circuits. A number of simulation engines have been developed, including general simulators such as NEURON (*Hines and Carnevale, 1997*), NEST (*Gewaltig and Diesmann, 2007*), Brian (*Stimberg et al., 2019*), Nengo (*Bekolay et al., 2014*), Neurokernel (*Givon and Lazar, 2016*), DynaSim (*Sherfey et al., 2018*), and the ones that specialize in multi-scale simulation, for example MOOSE (*Ray and Bhalla, 2008*), in compartmental models, for example ARBOR (*Akar et al., 2019*), and in fMRI-scale simulation for example The Virtual Brain (*Sanz Leon et al., 2013*; *Melozzi et al., 2017*). Other tools improve the accessibility to these simulators by (i) facilitating the creation of large-scale neural networks, for example BMTK (*Dai et al., 2020a*) and NetPyNE (*Dura-Bernal et al., 2019*), and by (ii) providing a common interface, simplifying the simulation workflow and streamlining parallelization of simulation, for example PyNN (*Davison et al., 2008*), Arachne (*Aleksin et al., 2017*), and NeuroManager (*Stockton and Santamaria, 2015*). To facilitate access and exchange of neurobiological data worldwide, a number of model specification standards have been worked upon in parallel including MorphML (*Crook et al., 2007*), NeuroML (*Gleeson et al., 2010*), SpineML (*Tomkins et al., 2016*), and SONATA (*Dai et al., 2020b*).

Even with the help of these computational tools, it still takes a substantial amount of manual effort to build executable circuits from real data provided, for example, by model databases such as ModelDB/NeuronDB (*Hines et al., 2004*) and NeuroArch (*Givon et al., 2015*). Moreover, with the ever expanding size of the fruit fly brain datasets, it has become more difficult to meet the demand of creating executable circuits that can be evaluated with different datasets. In addition, with very few exceptions, comparisons of circuit models, a standard process in the computer science community, are rarely available in the computational neuroscience literature.

Substantial efforts by the system neuroscience community went into developing tools for visualizing the anatomy of the brain. A number of tools have been developed to provide interactive, web-based interfaces for exploring, visualizing and analyzing fruit fly brain and ventral nerve cord datasets, for both the adult (*Chiang et al., 2011*; *Scheffer et al., 2020*) and the larva (*Ohyama et al., 2015*). These include the FlyCircuit (*Chiang et al., 2011*), the Fruit Fly Brain Observatory (FFBO/NeuroNLP) (*Ukani et al., 2019*), Virtual Fly Brain (*Milyaev et al., 2012*), neuPrintExplorer (*Clements et al., 2020*), FlyWire (*Dorkenwald et al., 2020*), and CATMAID (*Saalfeld et al., 2009*). Similar tools have been developed for other model organisms, such as the Allen Mouse Brain Connectivity Atlas (*Oh et al., 2014*), the WormAtlas for *C. elegans* (https://www.wormatlas.org) and the Z Brain for zebra fish (*Randlett et al., 2015*). A number of projects, for example (*Bates et al., 2020*), offer a more specialized capability for visualizing and analyzing neuroanatomy data.

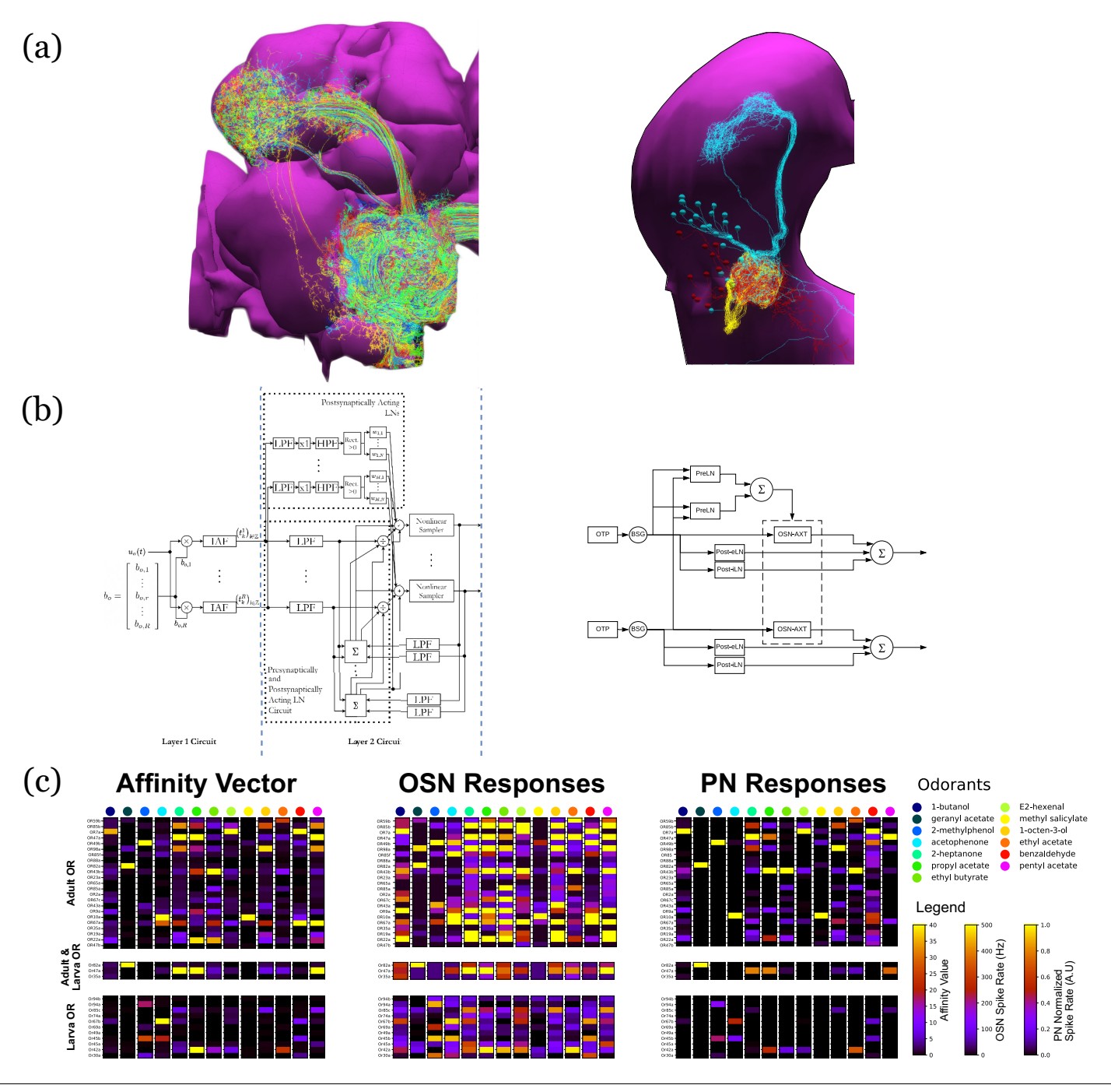

**Figure 8.** Evaluation and Comparison of two *Drosophila* Early Olfactory System (EOS) models describing adult (*left*, developed based on Hemibrain dataset) and larval (*right*, developed based on LarvaEM dataset) circuits. (a) Morphology of Olfactory Sensory Neurons (OSNs) in the Antenna (ANT), Local Neurons (LNs) in the Antennal Lobe (AL) and Projection Neurons in the AL. (b) Circuit diagrams depicting the Antenna and Antennal Lobe circuit motifs. (c) (left) Interaction between 13 odorants and 37 odorant receptors (ORs) characterized by affinity values. The ORs expressed only in the adult fruit flies are grouped in the top panel; the ones that are expressed in both the adult and the larva are grouped in the middle panel; and those expressed only in the larva are shown in the bottom panel. Steady-state outputs of the EOS models to a step concentration waveform of 100 ppm are used to characterize combinatorial codes of odorant identities at the OSN level (middle) and the PN level (right).

While these tools have significantly improved the access to and exploration of brain data a number of recent efforts started to bridge the gap between neurobiological data and computational modeling including the Geppetto (*Cantarelli et al., 2018*), the OpenWorm (*Szigeti et al., 2014*) and the Open Source Brain (*Gleeson et al., 2019*) initiatives and the Brain Simulation Platform of the Human Brain Project (*Einevoll et al., 2019*). However, without information linking circuit activity/computation to the structure of the underlying neuronal circuits, understanding the function of brain circuits remains elusive. Lacking a systematic method of automating the process of creating and exploring the function of executable circuits at the brain or system scale levels hinders the application of these tools when composing more complex circuits. Furthermore, these tools fall short of offering the capability of generating static circuit diagrams, let alone interactive ones. The experience of VLSI design, analysis, and evaluation of computer circuits might be instructive here. An electronic circuit engineer reads a circuit diagram of a chip, rather than the 3D structure of the tape-out, to understand its function, although the latter ultimately realizes it. Similarly, visualization of a biological circuit alone, while powerful and intuitive for building a neural circuit, provides little insights into the function of the circuit. While simulations can be done without a circuit diagram, understanding how an executable circuit leads to its function remains elusive.

The tools discussed above all fall short of offering an integrated infrastructure that can effectively leverage the ever expanding neuroanatomy, genetic and neurophysiology data for creating and exploring executable fly brain circuits. Creating circuit simulations from visualized data remains a major challenge and requires extraordinary effort in practice as amply demonstrated by the Allen Brain Observatory (*de Vries et al., 2020*). The need to accelerate the pace of discovery of the functional logic of the brain of model organisms has entered a center stage in brain research.

FlyBrainLab is uniquely positioned to accelerate the discovery of the functional logic of the *Drosophila* brain. Its interactive architecture seamlessly integrates and brings together computational models with neuroanatomical, neurogenetic, and neurophysiological data, changing the organization of fruit fly brain data from a group of independently created datasets, arrays, and tables, into a well-structured data and executable circuit repository, with a simple API for accessing data in different datasets. Current data integration extensively focuses on connectomics/synaptomics datasets that, as demonstrated, strongly inform the construction of executable circuit models. We will continue to expand the capabilities of the NeuroArch database with genetic Gal4 lines (https://gene.neuronlp.fruitflybrain.org) and neurophysiology recordings including our own (http://antenna.neuronlp.fruitflybrain.org/). How to construct executable models of brain circuits using genetic and neurophysiology data sets is not the object of this publication and will be discussed elsewhere. Pointers to our initial work are given below.

As detailed here, the FlyBrainLab UI supports a highly intuitive and automated workflow that streamlines the 3D exploration and visualization of fly brain circuits, and the interactive exploration of the functional logic of executable circuits created directly from the analyzed fly brain data. In conjunction with the capability of visually constructing circuits, speeding up the process of creating interactive executable circuit diagrams can substantially reduce the exploratory development cycle.

The FlyBrainLab Utility and Circuit Libraries accelerate the creation of models of executable circuits. The Utility Libraries (detailed in the Appendix 2) help untangle the graph structure of neural circuits from raw connectome and synaptome data. The Circuit Libraries (detailed in the Appendix 3) facilitate the exploration of neural circuits of the neuropils of the central complex and, the development and implementation of models of the adult and larva fruit fly early olfactory system.

Importantly, to transcend the limitations of the connectome, FlyBrainLab is providing Circuit Libraries for molecular transduction in sensory coding (detailed in the Appendix 3), including models of sensory transduction and neuron activity data (*Lazar et al., 2015a*; *Lazar et al., 2015b*; *Lazar and Yeh, 2020*). These libraries serve as entry points for discovery of circuit function in the sensory systems of the fruit fly (*Lazar and Yeh, 2019*; *Lazar et al., 2020a*). They also enable the biological validation of developed executable circuits within the same platform.

The modular software architecture underlying FlyBrainLab provides substantial flexibility and scalability for the study of the larva and adult fruit fly brain. As more data becomes available, we envision that the entire central nervous system of the fruit fly can be readily explored with FlyBrainLab. Furthermore, the core of the software and the workflow enabled by the FlyBrainLab for accelerating discovery of *Drosophila* brain functions can be adapted in the near term to other model organisms including the zebrafish and bee.

# Materials and methods

The FlyBrainLab interactive computing platform tightly integrates tools enabling the morphological visualization and exploration of large connectomics/synaptomics datasets, interactive circuit construction and visualization and multi-GPU execution of neural circuit models for in silico experimentation. The tight integration is achieved with a comprehensive open software architecture and libraries to aid data analysis, creation of executable circuits and exploration of their functional logic.

## Architecture of FlyBrainLab

FlyBrainLab exhibits a highly extensible, modularized architecture consisting of a number of interconnected server-side and user-side components (see *Appendix 1—figure 1*) including the NeuroArch Database, the Neurokernel Execution Engine and the NeuroMinerva front-end. The architecture of FlyBrainLab and the associated components are described in detail in Appendix 1.

## FlyBrainLab Utilities Libraries

FlyBrainLab offers a number of utility libraries to untangle the graph structure of neural circuits from raw connectome and synaptome data. These libraries provide a large number of tools including high level connectivity queries and analysis, algorithms for discovery of connectivity patterns, circuit visualization in 2D or 3D and morphometric measurements of neurons. These utility libraries are described in detail in Appendix 2.

## FlyBrainLab Circuit Libraries

FlyBrainLab provides a number of libraries for analysis, evaluation and comparison of fruit fly brain circuits. The initial release of FlyBrainLab offers libraries for exploring neuronal circuits of the central complex, early olfactory system, and implementations of olfactory and visual transduction models. These circuit libraries are described in detail in Appendix 3.

## Loading publicly available datasets into NeuroArch Database

All datasets are loaded into the NeuroArch database (*Givon et al., 2015*; *Givon et al., 2014*) using the NeuroArch API (https://github.com/fruitflybrain/neuroarch).

For the FlyCircuit dataset (*Chiang et al., 2011*) version 1.2, all 22,828 female *Drosophila* neurons were loaded, including their morphology, putative neurotransmitter type, and other available metadata. The original name of the neurons was used. These names also serve as the 'referenceId' pointing to the record in the original dataset. Connectivity between neurons was inferred according to *Huang et al., 2018* and loaded as a different, inferred class of synapses, totaling 4,538,280 connections between pairs of neurons. The metadata was provided by the authors (*Huang et al., 2018*).

For the Hemibrain dataset (*Scheffer et al., 2020*), release 1.0.1. Attributes of the neurons, synapses and connections were obtained from the Neuprint database dump available at (https://storage.cloud.google.com/hemibrain-release/neuprint/hemibrain_v1.0.1_neo4j_inputs.zip). The neuropil boundary mesh and neuron morphology were obtained by invoking the neuprint-python API (*Clements et al., 2020*) of the database server publicly hosted by the original dataset provider. The former was post-processed to simplify the mesh object in MeshLab (https://www.meshlab.net) using quadric edge collapse decimation with a percentage of 0.05. All coordinates were scaled by 0.008 to a [μm] unit. It included a total of 24,770 neurons that were designated in the Neuprint database as 'Traced', 'Roughly Traced', as well as the neurons that were assigned a name or a cell type. Cell type and neuron name follow the 'type' and 'instance' attributes, respectively, in the original dataset. To create a unique name for each neuron, neurons with the same instance names were padded with a sequential number. The BodyIDs of neurons in the original dataset use the 'referenceId'. A total of 3,604,708 connections between pairs of neurons were loaded, and included the positions of 14,318,675 synapses.

At the time of publication, the Hemibrain dataset release 1.2 (https://storage.cloud.google.com/hemibrain-release/neuprint/hemibrain_v1.2_neo4j_inputs.zip) was also loaded into the NeuroArch Database. It included a total of 25,842 neurons, 3,817,700 connections between pairs of these neurons and the positions of 15,337,617 synapses.

For the Larva L1EM dataset (*Ohyama et al., 2015*), a total of 1,051 neurons characterized by their morphology and metadata were loaded from the publicly served database server at https://l1em.

catmaid.virtualflybrain.org. The IDs of neurons in the original dataset were used as 'referenceId'. A total of 30,350 connections between pairs of neurons were loaded, including the position of 121,112 synapses. All coordinates were scaled by 0.001 to a [$\mu m$] unit.

For the Medulla 7 Column dataset (*Takemura et al., 2015*), the attributes of the neurons, synapses and connections were obtained from the Neuprint database server export available at https://storage.cloud.google.com/hemibrain-release/neuprint/fib25_neo4j_inputs.zip. Neuron morphology was obtained from https://github.com/janelia-flyem/ConnectomeHackathon2015 commit 81e94a9. Neurons without a morphology were omitted during loading. The rest of the procedure is the same as for loading the Hemibrain dataset. A total of 2365 neurons, 42,279 connections between pairs of neurons, and the positions of 130,203 synapses were loaded. Neurotransmitter data was obtained from the Gene Expression Omnibus accession GSE116969 of the transcriptome study published in *Davis et al., 2020*.

Extra annotations were avoided as much as possible when loading these datasets to the Neuro-Arch database for public download. If any, they were used to comply with the NeuroArch data model. The complete loading scripts are available at https://github.com/FlyBrainLab/datasets.

## Use Case 1: building fly brain circuits with English queries

The circuit in *Figure 3 (a1)* was built using the Medulla 7 Column dataset. The following English queries were used to construct the circuit: (1) 'show T4a in column home', (2) 'color lime', (3) 'add presynaptic neurons', (4) 'color gray', (5) 'add presynaptic $Dm$ neurons with more than five synapses', (6) 'color cyan', (7) 'add presynaptic $Pm$ neurons with more than five synapses', (8) 'color yellow', (9) 'pin T4a in column home', (10) 'pin $Dm$', (11) 'pin $Pm$'.

The circuit in *Figure 3 (b1)* was built using the Hemibrain dataset release 1.2. The following English queries were used to construct the circuit: (1) 'show MBON presynaptic to neurons that has outputs in FB layer 3 with at least 10 synapses', (2) 'color mbon yellow', (3) 'add postsynaptic neurons with at least 10 synapses that has output in FB layer 3', (4) 'color forest green', (5) 'add mbon postsynaptic to neurons that have input in FB layer 3 with at least 10 synapses', (6) 'color red'.

The circuit in *Figure 3 (c1)* was built using the Larva L1EM dataset. We first query the neuron MBON that innervate the g compartment by 'show $MBON-g$ in right mb'. The Information Panel in the FlyBrainLab UI provides a list of presynaptic partners and a list of postsynaptic partners of the neuron selected. After filtering the list by name and by the number of synapses, each neuron and the synapses to/from the neuron can be added to the NeuroNLP window for visualization. Finally, the collection of all filtered results can be added to the NeuroNLP window for visualization by clicking a single button. The circuit in *Figure 3 (c1)* was constructed by leveraging this capability.

The circuit in *Figure 3 (d1)* was built using the Hemibrain dataset release 1.2. First, the LPTCs and GLNs in the right hemisphere were added with the NLP queries 'show LPTC' and 'add /rGLN(.*)R(.*)/r'. Second, to obtain the pathway between the two neuron types, the following query was invoked:

```
# query for LPTC

res1 = client.executeNLPquery("show LPTC")

# color the neurons in the previous query
_ = client.executeNLPquery('color orange')

# query for GLN on the right hemisphere using regular expression
res2 = client.executeNLPquery("add /rGLN(.*)R(.*)/r")
```

```
# color the neurons in the previous query
_ = client.executeNLPquery("color cyan")
```

```
# get the unique names of the GLNs
gln = [v['uname'] for v in res2.neurons.values()]
```

```
# query using NeuroArch JSON format
task = {"query": [
                {"action": {"method": {"path_to_neurons": {
                        "pass_through": gln,
                        "max_hops": 2,
                        "synapse_threshold": 5
                }}},
                "object": {"state": 1}}],
            "format": "morphology",
            "verb": "add"
}
```

```
res3 = client.executeNAquery(task)
_ = client.executeNLPquery("color red")
```

After building up the circuit in the NeuroNLP window, the connectivity matrices of the four circuits were retrieved using the 'get_neuron_adjacency_matrix' method.

## Use Case 2: exploring the structure and function of yet to be discovered brain circuits

To investigate the overall brain level structure from Hemibrain neurons (*Figure 4a–c*), NeuroArch Database was queried for all neurons in the Hemibrain dataset and connectivity information (in the form of a directed graph) was extracted using the FlyBrainLab Client API (see Appendix 1.2). The Louvain algorithm (*Blondel et al., 2008*) of the NeuroGraph Library (see Appendix 2) was used to detect the structure of the graph. Apart from the connectivity of the neurons, any annotation of the neurons was excluded from the analysis. A random subset of neurons from each densely connected group are visualized and colored in the NeuroNLP. Group names are assigned by visually inspecting the results displayed by the NeuroNLP window and known neuropil structure of the fly brain. The circuit diagram was created by hand according to the groups and inter-group connections.

To define cell types in the VA1v glomerulus connectome dataset (*Figure 4d–f*), NeuroArch Database was queried for all neurons in the VA1v dataset and connectivity information (in the form of a directed graph) was extracted by using the FlyBrainLab Client API (see Appendix 1.2). The Adjacency Spectral Embedding algorithm (*Sussman et al., 2012*) of the NeuroGraph library (see Appendix 2) was applied to calculate embeddings via the $GMM \circ ASE$ approach (*Priebe et al., 2017*). Annotations of the identity of the neurons, if any, were not used in this step of the analysis. To check the quality of the embeddings result, human-annotated data from the original dataset was used to color the neurons according to their cell types. Neurons from each cell type were visualized and colored by NeuroNLP commands. The circuit diagram was generated using the NeuroGraph Library. Connections between neurons were established only if more than 10 synapses from a presynaptic neuron to a postsynaptic neuron could be identified. Coloring of the cell types matches the NeuroNLP commands.

To investigate the downstream neurons of V glomerulus projection neurons of the AL (*Figure 4g–i*), the latter neurons and their postsynaptic partners with more than 10 synapses were visualized

with NeuroNLP queries. Arborization data of each neuron was queried to determine whether it is a local neuron or an output neuron of the lateral horn. NeuroGraph Library was used to generate the circuit diagram. The circuit diagram generation is based on the GraphViz library (*Ellson et al., 2001*), and additional information such as group names were used for arranging the diagram.

## Use Case 3: interactive exploration of executable fruit fly brain circuit models

Connectome data published by *Rivera-Alba et al., 2011* was uploaded into the NeuroArch Database, including the six photoreceptors, eight neurons, and six neurites of multiple amacrine cells that innervate the cartridge. For simplicity, each of the amacrine cell neurites was considered as a separate neuron. The traced neuron data were obtained from authors of *Rivera-Alba et al., 2011* and subsequently converted into neuron skeletons in SWC format. The connectivity between these neurons was provided in *Rivera-Alba et al., 2011* as supplementary information.

Loading data into NeuroArch Database was achieved with the FlyBrainLab 'query' module. The 'query' module provides a mirror of the functionality available in the high-level NeuroArch API. The pathway was then explored in the NeuroNLP window and the connectivity matrix extracted by FlyBrainLab Client API call, as described above.

The circuit diagram in *Figure 5* was created manually using Inkscape. All blocks representing the neurons were added with attributes that had the neuron's unique name in the database as a label value, and added with '.neuron' class designation. Similarly, all synapses were added with '.synapse' class designation. The diagram was made interactive by loading a javascript file available in the NeuGFX library. The runtime interaction is controlled by the circuit module of the FlyBrainLab Client API.

Appendix 4 provides a walk through of the process of creating and operating an interactive circuit diagram of a Lamina cartridge circuit starting from raw connectomic and synaptomic data (*Rivera-Alba et al., 2011*). Some of the core FlyBrainLab capabilities (see also Appendix 1.2) are also highlighted. The walk through is accompanied by a Jupyter notebook available at https://github.com/FlyBrainLab/Tutorials/tree/master/tutorials/cartridge/Cartridge.ipynb.

In what follows, the usage of FlyBrainlab in analyzing, evaluating and comparing more complex circuit models is demonstrated. For brevity, Jupyter notebooks are only provided on Github repositories disseminated at https://github.com/FlyBrainLab/FlyBrainLab/wiki/FlyBrainLab-Resources.

## Use Case 4: analyzing, evaluating, and comparing circuit models of the fruit fly central complex

Model A (*Givon et al., 2017*; *Figure 6a*), Model B (*Kakaria and de Bivort, 2017*; *Figure 6b*) and Model C (*Su et al., 2017*; *Figure 6c*) were implemented using the CXcircuits Library (see also Appendix 3). A wild-type fruit fly CX circuit diagram based on model A was created in the SVG format and made interactive in the NeuroGFX window. The neurons modeled in the circuit diagram were obtained by querying all the neurons of the CX neuropils in the FlyCircuit dataset. The innervation pattern of each neuron was obtained from *Lin et al., 2013* and visually examined in the NeuroNLP window. Based on the assumptions made for each model, a standard name was assigned to the model of the neuron according to the naming scheme adopted in the CXcircuits Library. The neurons with missing morphologies in the FlyCircuit dataset were augmented with the data available in the literature (*Wolff et al., 2015*; *Lin et al., 2013*).

This all encompassing circuit diagram was then adapted to the other models. Since the assumptions about the subregions that neurons receive inputs from and output to are different in each circuit model, slightly different names may be assigned to neurons in different circuit models. The complete list of modeled neurons of the three models are provided in *Supplementary file 1* 'CX_models.xlsx', along with their corresponding neurons in the FlyCircuit dataset. The CXcircuits Library also uses this list to enable synchronization of the neurons visualized in the NeuroNLP window with the neurons represented in the NeuroGFX window.

All three models include the same core subcircuits for modeling the Protocerebral Bridge - Ellipsoid Body (PB-EB) interaction. The core subcircuits include three cell types, namely, the PB-EB-LAL, PB-EB-NO, and EB-LAL-PB neurons (NO - Noduli, LAL - Lateral Accessory Lobe, see also *Givon et al., 2017* for a list of synonyms of each neuron class). These cells innervate three neuropils,

either PB, EB and LAL or PB, EB, and NO. Note that only synapses within PB and EB are considered. For model A, this is achieved by removing all neurons that do not belong to the core PB-EB circuit. This can be directly performed on the circuit diagram in the NeuroGFX window or by using the CXcircuits API. Model B includes an additional cell type, the PB local neurons that introduce global inhibition to the PB-EB circuit. Model C does not include PB local neurons, but models 3 types of ring neurons that innervate the EB. Both PB local neurons and ring neurons are present in model A. However, except for their receptive fields, all ring neurons in model A are the same (see below). *Figure 9* depicts the correspondence established between the morphology of example neurons and their respective representation in the overall circuit diagram.

In Model C, the subcircuit consisting of the PB-EB-LAL and EB-LAL-PB neurons was claimed to give rise to the persistent bump activity while the interconnect between PB-EB-NO and EB-LAL-PB allowed the bump to shift in darkness. To better compare with the other two models that did not model the shift in darkness, PB-EB-NO neurons were interactively disabled from the diagram.

For a single vertical bar presented to the fly at the position shown in *Figure 6 (d1)* (see also the flattened visual input in *Figure 6 (d2)*, the PB glomeruli or the EB wedges innervated by neurons of each of the three circuit models that receive injected current or external spike inputs are, respectively, highlighted in *Figure 6 (a3, b3, c3)*. The CXcircuit Library generates a set of visual stimuli and computes the spike train and/or the injected current inputs to each of the three models.

For model A (*Givon et al., 2017*), each PB glomerulus is endowed with a rectangular receptive field that covers 20° in azimuth and the entire elevation. Together, the receptive fields of all PB glomeruli tile the 360° azimuth. All PB neurons with dendrites in a glomerulus, including the PB-EB-LAL and PB-EB-NO neurons, receive the visual stimulus filtered by the receptive field as an injected current. Additionally, each Bulb (BU) microglomerulus is endowed with a Gaussian receptive field with a standard deviation of 9° in both azimuth and elevation. The ring neuron innervating a microglomerulus receives the filtered visual stimulus as an injected current (see also the arrows in *Figure 6 (a3)*). Neuron dynamics follow the Leaky Integrate-and-Fire (LIF) neuron model

$$C^i \frac{dV^i}{dt} = -\frac{V^i - V_0^i}{R^i} + I^i,$$

(1)

where $V^i$ is the membrane voltage of the $i$ th neuron, $C^i$ is the membrane capacitance, $V_0^i$ is the resting potential, $R^i$ is the resistance, and $I^i$ is the synaptic current. Upon reaching the threshold voltage $V_{th}^i$, each neuron's, membrane voltage is reset to $V_r^i$. Synapses are modeled as $\alpha$-synapses with dynamics given by the differential equations

$$
\begin{aligned}
g^i(t) &= \bar{g}^i s^i(t) \\
\frac{ds^i}{dt}(t) &= h^i(t) 1_{[t \geq 0]}(t) \\
\frac{dh^i}{dt}(t) &= -(a_r^i + a_d^i)h(t) - a_r^i a_d^i s^i + a_r^i a_d^i \sum_k \delta(t - t_k^i),
\end{aligned}
$$

(2)

where $\bar{g}^i$ is a scaling factor, $a_r^i$ and $a_d^i$ are, respectively, the rise and decay time of the synapse, $1_{[t \geq 0]}(t)$ is the Heaviside function and $\delta(t)$ is the Dirac function. $\delta(t - t_k^i)$ indicates an input spike to the $i$ th synapse at time $t_k^i$.

For Model B (*Kakaria and de Bivort, 2017*), the receptive field of each of the 16 EB wedges covers 22.5° in azimuth. All EB-LAL-PB neurons that innervate a wedge receive a spike train input whose rate is proportional to the filtered visual stimulus (see also arrow in *Figure 6 (b3)*). The maximum input spike rate is 120 Hz when the visual stimulus is a bar of width 20° at maximum brightness 1. A 5 Hz background firing is always added even in darkness. Neurons are modeled as LIF with a refractory period of 2.2 ms as suggested in *Kakaria and de Bivort, 2017*. For synapses, instead of using the postsynaptic current (PSC)-based model described in *Kakaria and de Bivort, 2017*, the $\alpha$-synapse described above was used and its parameters were chosen such that the time-to-peak and peak value approximately matched that of the PSC-based synapse.

For Model C (*Su et al., 2017*), the receptive field of each of the 16 EB wedges covers 22.5° in azimuth. Two PB-EB-LAL neurons project to a wedge each from a different PB glomerulus. Input to the Model C circuit is presented to pairs of PB glomeruli (see also arrows in *Figure 6 (c3)*), and all

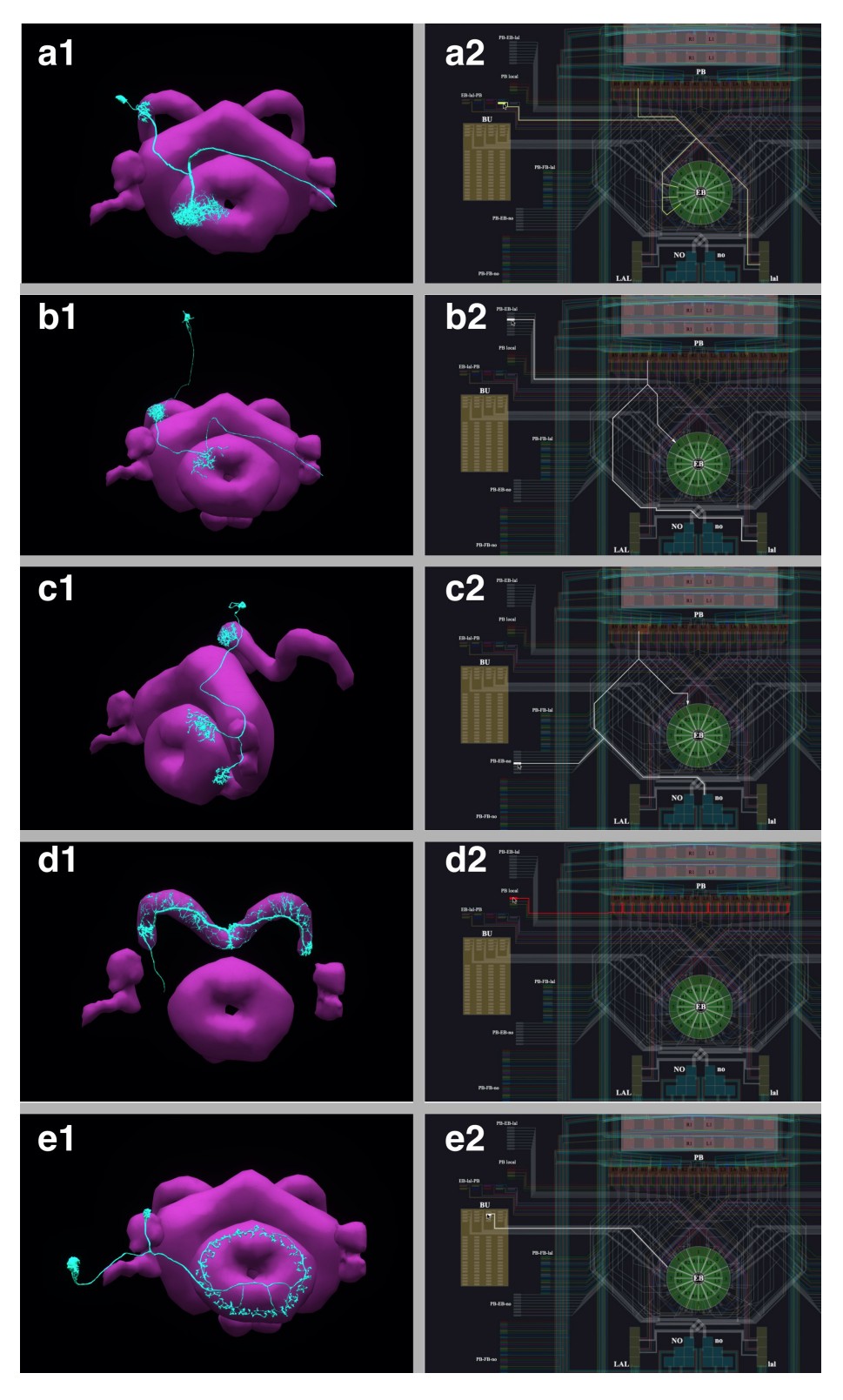

**Figure 9.** The correspondence between the morphology and the circuit diagram representation of 5 classes of neurons that determine the PB-EB interaction. (**a1, a2**) EB-LAL-PB neuron and its wiring in the circuit diagram. (**b1, b2**) PB-EB-LAL neuron and its wiring in the circuit diagram. (**c1, c2**) PB-EB-NO neuron and its wiring in the circuit diagram. (**d1, d2**) PB local neuron and its wiring in the circuit diagram. (**e1, e2**) Ring neuron and its wiring in the circuit diagram.

neurons with dendrites in these two PB glomeruli receive a spike train input at a rate proportional to the filtered visual stimuli, with a maximum 50 Hz when the bar is at maximum brightness 1. Neurons are modeled as LIF neurons with a refractory period of 2 ms (as suggested in *Su et al., 2017*). Synapses are either modeled by the AMPA/GABAA receptor dynamics as

$$
\begin{aligned}
g^i(t) &= \overline{g}^i s^i(t) \\
\frac{ds^i}{dt}(t) &= -\frac{s^i(t)}{\tau^i} + \sum_k \delta(t - t^i_k),
\end{aligned}
\tag{3}
$$

where $g^i(t)$ is the synaptic conductance, $\tau^i$ is the time constant, and $s^i(t)$ is the state variable of the $i$ th synapse, or modeled by the NMDA receptor dynamics (*Su et al., 2017*)

$$
\begin{aligned}
g^i(t) &= \frac{\overline{g}^i s^i(t)}{1 + [Mg^{2+}]^i e^{-\frac{0.062V}{3.56}}} \\
\frac{ds^i}{dt}(t) &= -\frac{s^i(t)}{\tau^i} + \alpha^i(1 - s^i(t)) \sum_k \delta(t - t^i_k),
\end{aligned}
\tag{4}
$$

where $g^i(t)$ is the synaptic conductance, $\overline{g}^i$ is the maximum conductance, $s^i(t)$ is the state variable, $\tau^i$ is the time constant, $\alpha^i > 0$ is a constant, $[Mg^{2+}]^i$ is the extracellular concentration of $Mg^{2+}$, respectively, of the $i$ th synapse and $V$ is the membrane voltage of the postsynaptic neuron.

Parameters of the above models can be found in *Givon et al., 2017*; *Kakaria and de Bivort, 2017*; *Su et al., 2017* and in the CXcircuit Library.

Commonly used models, such as the LIF neuron and the α-synapse, are built-into the Neurokernel Execution Engine. Only the model parameters residing in the NeuroArch Database need to be specified via NeuroArch API. The Neurokernel automatically retrieves the circuit models and their parameters from the NeuroArch Database based on the last queried circuit model. For models that are not yet built-into the Neurokernel Execution Engine, such as the PSC-based model in Model B, users must provide an implementation supported by the Neurodriver API.

The 35 s visual stimulus, depicted in *Figure 6 (d3)*, was presented to all three models. A bright vertical bar moves back and forth across the entire visual field while a second bar with lower brightness is presented at a fixed position. *Figure 6 (d3)* (bottom) depicts the time evolution of the visual input.

To visualize the response of the three executable circuits, the mean firing rate $r^j(t)$ of all EB-LAL-PB neurons that innervate the $j$ th EB wedge was calculated following *Su et al., 2017*.

$$
r^j(t) = \frac{1}{N^j} \sum_{i \in \mathbb{P}^j} \left( \sum_k \delta(t - t^i_k) \star e^{-\frac{t}{0.7215}} \right),
\tag{5}
$$

where $\star$ denotes the convolution operator, $\mathbb{P}^j$ is the index set of EB-LAL-PB neurons that innervate the $j$ th EB wedge, whose cardinality is $N^j$, and $t^i_k$ is the time of $k$th spike generated by $i$th neuron. CXcircuit Library provides utilities to visualize the circuit response.

Jupyter notebooks for Models A, B and C used to generate *Video 1* are available at https://github.com/FlyBrainLab/CXcircuits/tree/master/notebooks/elife20.

## Use Case 5: analyzing, evaluating, and comparing adult antenna and antennal lobe circuit models based upon the FlyCircuit and hemibrain datasets

The Early Olfactory System models based on the FlyCircuit and the Hemibrain datasets were implemented using the EOScircuits Library (see also Appendix 3). The circuit architecture, shown in *Figure 7b* (left), follows previous work (*Lazar et al., 2020a*) based upon the inferred connectivity between LNs and PNs in the FlyCircuit dataset and the functional connectivity between LNs and OSNs observed in experiments (*Olsen and Wilson, 2008*) (see also *Figure 7a* (left)). Specifically, LNs are separated into two groups: a group of presynaptically-acting LNs assumed to receive inputs from all OSNs and to project to the axon terminals of each of the OSNs; another group of postsynaptically acting LNs assumed to receive inputs from all OSNs and to provide inputs to the PNs that arborize the same glomerulus. Only uniglomerular PNs are modeled and their

characterization is limited to their connectivity. For the Hemibrain dataset, the exact local neurons and their connectivity within the AL circuit are used. Specifically, LNs are divided into presynaptically acting and postsynaptically acting ones based on the number of synaptic contacts onto OSNs and PNs, respectively. If the majority of synapses of an LN is targeting OSNs, it is modeled as a presynaptically acting LN. Otherwise, it is modeled as a postsynaptically acting LN. Note that the connectivity pattern in the circuit model based on FlyCircuit dataset is inferred (*Lazar et al., 2020a*). whereas in the circuit model based on Hemibrain dataset is extracted from the data.

At the input layer (the Antenna Circuit), the stimulus model for the adult EOS circuit builds upon affinity data from the DoOR dataset (*Münch and Galizia, 2016*), with physiological recordings for 23/51 receptor types. Receptors for which there is no affinity data in the DoOR dataset were assumed to have zero affinity values. Two input stimuli were used. The initial input stimulus was 5 s long; between 1 and 3 s, ammonium hydroxide with a constant concentration of 100 ppm was presented to the circuits in *Figure 7b* and the responses are shown in (*Figure 7c*). The same odorant waveform was used here as in *Figure 7d*. To generate the concentration waveform of the odorant, values were drawn randomly from the uniform distribution between 0 and 100 ppm every $10^{-4}$ seconds between 1 and 3 s in *Figure 7d*. The sequence was then filtered by a lowpass filter with a 30 Hz bandwidth (*Kim et al., 2011*) to obtain the concentration of the odorant.

Olfactory Sensory Neurons expressing each one receptor type processed the input odorant in parallel. The Antennal Lobe model based on FlyCircuit data is divided into two sub-circuits: (1) the ON-OFF circuit and (2) the Predictive Coding circuit (*Lazar and Yeh, 2019*). The ON-OFF circuit describes odorant gradient encoding by Post-synaptic LNs in the AL, while the Predictive Coding circuit describes a divisive normalization mechanism by Pre-synaptic LNs that enable concentration-invariant odorant identity encoding by Projection Neurons in the AL.

The EOS model based on Hemibrain dataset takes advantage of the detailed connectivity between neurons (see *Figure 7a* (right)) and introduces a more extensive connectome-synaptome model of the AL (see *Figure 7b* (right)). FlyBrainLab utility libraries were used to (1) access the Hemibrain data, (2) find PNs and group them by glomeruli, (3) use this data to find the OSNs associated with each glomerulus, (4) find LNs and group connectivity between OSNs, LNs and PNs. Multiglomerular PNs were not included. Contralateral LN connections were ignored. All PNs were assumed to be excitatory. An executable circuit was constructed in FlyBrainLab using the Hemibrain data. In addition to the baseline model in *Figure 7b* (left), the following components were introduced (1) LNs that innervate specific subsets of glomeruli, (2) LNs that provide inputs to both OSN axon terminals and to PNs dendrites, (3) synapses from PNs onto LNs.

## Use Case 6: evaluating, analyzing, and comparing early Olfactory circuit models of the larva and the adult fruit flies

The Early Olfactory System models for both the adult and the larval flies were implemented using the EOScircuits library (see also Appendix 3). The circuit of the adult EOS follows the one described above. Similarly, the larval model is implemented using physiological recording on 14/21 receptor types (*Kreher et al., 2005*). In both the adult and larval physiology datasets, 13 common monomolecular odorants were employed (see *Figure 8c* (legend)). Together, 13/23 odorant/receptor pairs for adult and 13/14 odorant/receptor pairs for larva were used for model evaluation, where each odorant was carried by a 100 ppm concentration waveform. In both adult and larva Antenna circuits, Olfactory Sensory Neurons expressing each receptor type processed an odorant waveform in parallel.

The adult Antennal Lobe model follows the one built on the Hemibrain data (*Scheffer et al., 2020*). Both the adult and the larva circuit components are parameterized by the number of LNs per type, where for instance there were 28 LNs used in the larval model in accordance to connectome data (*Berck et al., 2016*). In addition to neuron types, the AL circuit was modified in terms of connectivity from (1) LNs to Projection Neurons (PNs), (2) PNs to LNs and (3) LNs to LNs. Refer to *Table 1* for more details.

The evaluation of both EOS models focused on the Input/Output relationship comparison between the adult and the larval EOS models. For each of the 13 odorants, the input stimulus is a 5 s concentration waveform that is 100 ppm from 1 to 4 s and 0 ppm otherwise. Both adult and larval models reach steady-state after 500 ms and the steady-state population responses averaged across 3–4 s are computed as odorant combinatorial code at each layer (i.e. OSN response, PN response).

**Table 1.** Neurons and neuron types used for visualization and simulation in *Figure 8*.

| Neuropil | Neuron Type | Organism | Number (Model in *Figure 8b*) | Number (Visualization in *Figure 8a*) |
|---|---|---|---|---|
| Antenna | Olfactory Sensory Neuron | Adult | 51 receptor types (channels), 1357 total olfactory sensory neurons | 1357 |
| | | Larva | 14 receptor types (channels), 1 neuron expressing the same receptor type (14 neurons in total) | 21 |
| Antennal Lobe | Uniglomerular Projection Neuron | Adult | 1 neuron per channel, 51 total | 141 (Different number per glomerulus) |
| | | Larva | 1 neuron per channel (14 neurons in total) | 21 |
| | Presynaptic Local Neuron | Adult | 97 neurons | 97 |
| | | Larva | 6 pan-glomerular neurons | 5 |
| | Postsynaptic Inhibitory Local Neuron | Adult | 77 neurons | 77 |
| | | Larva | 0-1 neuron per channel (11 neurons in total) | 7 |
| | Postsynaptic Excitatory Local Neuron | Adult | 77 (assumed to be the same as the Postsynaptic Inhibitory Local Neuron population) | 77 |
| | | Larva | 0-1 neuron per channel (11 neurons in total) | 7 |

## Code availability and installation

Stable and tested FlyBrainLab installation instructions for user-side components and utility libraries are available at https://github.com/FlyBrainLab/FlyBrainLab for Linux, MacOS, and Windows. The installation and use of FlyBrainLab does not require a GPU, but a service-side backend must be running, for example, on a cloud service, that the user-side of FlyBrainLab can connect to. By default, the user-side-only installation will access the backend services hosted on our public servers. Note that users do not have write permission to the NeuroArch Database, nor will they be able to access a Neurokernel Server for execution. The server-side backend codebase is publicly available at https://github.com/fruitflybrain and https://github.com/neurokernel.

A full installation of FlyBrainLab, including all backend and frontend components, is available as a Docker image at https://hub.docker.com/r/fruitflybrain/fbl. The image requires a Linux host with at least 1 CUDA-enabled GPU and the nvidia-docker package (https://github.com/NVIDIA/nvidia-docker) installed. For a custom installation of the complete FlyBrainLab platform, a shell script is available at https://github.com/FlyBrainLab/FlyBrainLab.

To help users get started, a number of tutorials are available written as Jupyter notebooks at https://github.com/FlyBrainLab/Tutorials, including a reference to English queries at https://github.com/FlyBrainLab/Tutorials/blob/master/tutorials/getting_started/1b_nlp_queries.ipynb. An overview of the FlyBrainLab resources is available at https://github.com/FlyBrainLab/FlyBrainLab/wiki/FlyBrainLab-Resources.

## Data availability

The NeuroArch Database hosting publicly available FlyCircuit (RRID:SCR_006375), Hemibrain, Medulla 7-column and Larva L1EM datasets can be downloaded from https://github.com/FlyBrainLab/datasets. The same repository provides Jupyter notebooks for loading publicly available datasets, such as the FlyCircuit dataset with inferred connectivity (*Huang et al., 2018*), the Hemibrain dataset, the Medulla 7-column dataset and the Larva L1 EM dataset.

## Acknowledgements

The research reported here was supported by AFOSR under grant #FA9550-16-1-0410 and DARPA under contract #HR0011-19-9-0035. The authors thank the reviewers for their constructive comments that significantly improved the presentation of the manuscript.

## Additional information

### Funding

| Funder | Grant reference number | Author |
|---|---|---|
| Air Force Office of Scientific Research | FA9550-16-1-0410 | Aurel A Lazar |
| Defense Advanced Research Projects Agency | HR0011-19-9-0035 | Aurel A Lazar |

The funders had no role in study design, data collection and interpretation, or the decision to submit the work for publication.

### Author contributions

Aurel A Lazar, Conceptualization, Resources, Formal analysis, Supervision, Funding acquisition, Investigation, Methodology, Writing - original draft, Project administration, Writing - review and editing, Conceived the study and FlyBrainLab software architecture. Developed comparative models of the central complex. Developed comparative models of the early olfactory system.; Tingkai Liu, Conceptualization, Software, Formal analysis, Validation, Investigation, Visualization, Methodology, Writing - original draft, Writing - review and editing, Conceived the study and FlyBrainLab software architecture. Developed the FlyBrainLab platform. Developed user-side libraries. Developed comparative models of the early olfactory system.; Mehmet Kerem Turkcan, Conceptualization, Software, Formal analysis, Validation, Investigation, Visualization, Methodology, Writing - original draft, Writing - review and editing, Conceived the study and FlyBrainLab software architecture. Developed the FlyBrainLab platform. Developed user-side libraries and utility libraries. Developed comparative models of the central complex. Developed comparative models of the early olfactory system.; Yiyin Zhou, Conceptualization, Software, Formal analysis, Funding acquisition, Validation, Investigation, Visualization, Methodology, Writing - original draft, Project administration, Writing - review and editing, Conceived the study and FlyBrainLab software architecture. Developed the FlyBrainLab platform. Updated the server-side components of the existing FFBO architecture. Developed comparative models of the central complex.

### Author ORCIDs

Aurel A Lazar ![ORCID] https://orcid.org/0000-0003-4261-8709
Tingkai Liu ![ORCID] https://orcid.org/0000-0003-3075-7648
Mehmet Kerem Turkcan ![ORCID] https://orcid.org/0000-0001-9273-7293
Yiyin Zhou ![ORCID] https://orcid.org/0000-0003-4618-4039

### Decision letter and Author response

Decision letter https://doi.org/10.7554/eLife.62362.sa1
Author response https://doi.org/10.7554/eLife.62362.sa2

## Additional files

### Supplementary files

• Supplementary file 1. Full list of neurons used in the three CX models in Use Case 4 and their correspondence in the FlyCircuit dataset.

• Transparent reporting form

### Data availability

General information about the FlyBrainLab is available at https://www.fruitflybrain.org. Stable and tested FlyBrainLab installation instructions are available at https://github.com/FlyBrainLab/FlyBrainLab. An overview of the FlyBrainLab resources can be found at the FlyBrainLab Resource wiki page at https://github.com/FlyBrainLab/FlyBrainLab/wiki/FlyBrainLab-Resources. It includes links to individual code repositories for components, libraries and tutorials. The NeuroArch Database hosting

publicly available FlyCircuit, Hemibrain, Medulla 7-column and Larva L1EM datasets can be downloaded from https://github.com/FlyBrainLab/datasets. The same repository provides Jupyter notebooks for loading publicly available datasets, such as the FlyCircuit dataset with inferred connectivity, the Hemibrain dataset, the Medulla 7-column dataset and the Larva L1 EM dataset.

The following previously published datasets were used:

| Author(s) | Year | Dataset title | Dataset URL | Database and Identifier |
|---|---|---|---|---|
| Chiang AS, et al. | 2011 | FlyCircuit | https://doi.org/10.1016/j.cub.2010.11.056 | FlyCircuit DB, 10.1016/j.cub.2010.11.056 |
| Ohyama T, et al. | 2015 | L1EM | https://doi.org/10.1038/nature14297 | CATMAID, 10.1038/nature14297 |
| Scheffer LK, et al. | 2020 | Hemibrain | https://doi.org/10.7554/eLife.57443 | neuPrint, 10.7554/eLife.57443 |
| Takemura SY, et al. | 2015 | Medulla 7 Column Data | https://github.com/janelia-flyem/Connectome-Hackathon2015 | github, 10.1073/pnas.1509820112 |

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

## Appendix 1

## The architecture of FlyBrainLab

To support the study of the function of brain circuits FlyBrainLab implements an extensible, modularized architecture that tightly integrates fruit fly brain data and models of executable circuits. ***Appendix 1—figure 1*** depicts the architecture of FlyBrainLab on both the user- and backend server-side.

The backend server-side components are described in Appendix 1.1. The user-side components are presented in Appendix 1.2.

### 1.1 The server-side components

The server-side backend consists of four components: FFBO Processor, NeuroArch, Neurokernel and NeuroNLP servers. They are collectively called the FFBO servers. A brief description of each of the components is given below.

**FFBO Processor** implements a Crossbar.io router (https://crossbar.io/) that establishes the communication path among connected components. Components communicate using routed Remote Procedure Calls (RPCs) and a publish/subscribe mechanism. The routed RPCs enable functions implemented on the server-side to be called by the user-side backend (see also Section 1.2). After an event occurs, the publisher immediately informs topic subscribers by invoking the publish/subscribe mechanism. This enables, for example, the FFBO processor to inform the user-side and other servers when a new backend server is connected. The FFBO processor can be hosted locally or in the cloud. It can also be hosted by a service provider for, for example, extra data sharing. The open source code of the FFBO processors is available at https://github.com/fruitflybrain/ffbo.processor.

**NeuroArch Server** hosts the NeuroArch graph database (***Givon et al., 2015***) implemented with OrientDB (https://orientdb.org). The NeuroArch Database provides a novel data model for representation and storage of connectomic, synaptomic, cell type, activity, and genetic data of the fruit fly brain with cross-referenced executable circuits. The NeuroArch data model is the foundation of the integration of fruit fly brain data and executable circuits in FlyBrainLab. Low-level queries of the NeuroArch Database are supported by the NeuroArch Python API (https://github.com/fruitflybrain/neuroarch). The NeuroArch Server provides high level RPC APIs for remote access of the NeuroArch Database. The open source code of the NeuroArch Server is available at https://github.com/fruitflybrain/ffbo.neuroarch_component.

**Neurokernel Server** provides RPC APIs for code execution of model circuits by the Neurokernel Execution Engine (***Givon and Lazar, 2016***). Neurokernel supports the easy combination of independently developed executable circuits towards the realization of a complete whole brain emulation. The Neurokernel Execution Engine features:

- the core Neurokernel services (https://github.com/neurokernel/neurokernel) providing management capabilities for code execution, and communication between interconnected circuits,
- the Neurodriver services (https://github.com/neurokernel/neurodriver) providing low level APIs for code execution on GPUs according to user-specified circuit connectivity, biological spike generators and synapses, and input stimuli.

The Neurokernel Server directly fetches the specification of executable circuits from the NeuroArch Server, instantiates these circuits and transfers them for execution to the Neurokernel Execution Engine. The open source code of the Neurokernel Server is available at https://github.com/fruitflybrain/ffbo.neurokernel_component.

**NeuroNLP Server** provides an RPC API for translating queries written as English sentences, such as 'add dopaminergic neurons innervating the mushroom body', into database queries that can be interpreted by the NeuroArch Server API. This capability increases the accessibility of the NeuroArch Database to researchers without prior exposure to database programming, and facilitates research by simplifying the often-demanding task of writing database queries. The open source code of the NeuroNLP Server is available at https://github.com/fruitflybrain/ffbo.nlp_component.

## 1.2 The user-side components

The FlyBrainLab user-side consists of the NeuroMynerva frontend and the FlyBrainLab Client and Neuroballad backend components. A brief description of each of the components is given below.

**NeuroMynerva** is the user-side frontend of FlyBrainLab. It is a browser-based application that substantially extends upon JupyterLab by providing a number of widgets, including a Neu3D widget for 3D visualization of fly brain data, a NeuGFX widget for exploring executable neural circuits with interactive circuit diagrams, and an InfoPanel widget for accessing individual neuron/synapse data. All widgets communicate with and retrieve data from the FlyBrainLab Client. A master widget keeps track of the instantiated widgets by the user interface. With the JupyterLab native notebook support, APIs of the FlyBrainLab Client can be directly called from notebooks. Such calls provide Python access to NeuroArch queries and circuit execution. Interaction between code running in notebooks and widgets is fully supported.

**FlyBrainLab Client** is a user-side backend implemented in Python that connects to the FFBO processor and accesses services provided by the connected backend servers. FlyBrainLab Client provides program execution APIs for handling requests to the server-side components and parsing of data coming from backend servers. The FlyBrainLab Client also exhibits a number of high-level APIs for processing data collected from the backend servers, such as computing the adjacency matrix from retrieved connectivity data or retrieving morphometrics data. In addition, it handles the communication with the frontend through the Jupyter kernel.

**NeuroBallad** is a Python library that simplifies and accelerates executable circuit construction and simulation using Neurokernel in Jupyter notebooks in FlyBrainLab. NeuroBallad provides classes for specification of neuron or synapse models with a single line of code and contains functions for adding and connecting these circuit components with one another. NeuroBallad also provides capabilities for compactly specifying inputs to a circuit on a per-experiment basis.

**Core Functionalities Provided by FlyBrainLab** Examples of capabilities that users can directly invoke are: (1) Query using plain English and 3D graphics for building and visualizing brain circuits; (2) Query the NeuroArch Database using NeuroArch JSON format for building and visualizing brain circuits and constructing executable circuits; (3) Retrieval of the connectivity of the brain circuit built/visualized; (4) Retrieval of the graph representing the circuit built/visualized; (5) Retrieval of the circuit model corresponding to the brain circuit built; (6) User interface and API for circuit diagram interaction; (7) Specification of models and parameters for each circuit components; (8) Execution of the circuits represented/stored in the NeuroArch Database.

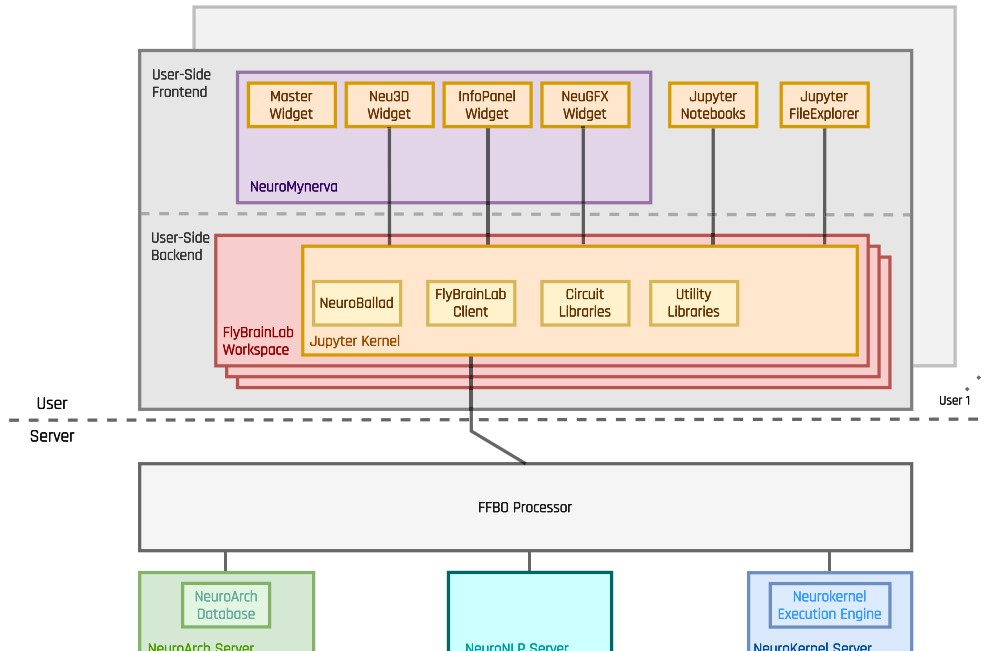

*Appendix 1—figure 1 continued on next page*

*Appendix 1—figure 1 continued*

**Appendix 1—figure 1.** The architecture of FlyBrainLab. The server-side architecture (*Ukani et al., 2019*) consists of the FFBO Processor, the NeuroNLP Server, the NeuroArch Server and the Neurokernel Server. The user-side provides the local execution environment as well as an easy-to-use GUI for multi-user access to the services provided by the server-side. The FlyBrainLab Utility Libraries and Circuit Libraries (see Sections 2 and 3 for details) can be loaded into the FlyBrainLab workspace of the user-side backend.

## Appendix 2

### Utility libraries for the fruit fly connectome/synaptome

Different connectome and synaptome datasets are often available at different levels of abstraction. For example, some datasets come with cell types labeled and some only provide raw graph level connectivity. Without extensive analysis tools, it takes substantial manual effort to construct and test a neural circuit. FlyBrainLab offers a number of utilities to explicate the graph structure of neural circuits from raw connectome and synaptome data. In conjunction with the capability of visually constructing circuits enabled by the NeuroMynerva front-end, speeding up the process of creating interactive executable circuit diagrams can substantially reduce the exploratory development cycle.

The FlyBrainLab Utility Libraries include:

- **NeuroEmbed**: Cell Classification and Cell Type Discovery,
- **NeuroSynapsis**: High Level Queries and Analysis of Connectomic and Synaptomic Data,
- **NeuroGraph**: Connectivity Pattern Discovery and Circuit Visualization Algorithms,
- **NeuroWatch**: 3D Fruit Fly Data Visualization in Jupyter Notebooks,
- **NeuroMetry**: Morphometric Measurements of Neurons.

In this section, we outline the capabilities enabled by the Utility Libraries listed above.

### NeuroEmbed: cell classification and cell type discovery

The NeuroEmbed library implements a set of algorithms for structure discovery based on graph embeddings into low-dimensional spaces providing capabilities for:

- Cell type classification based on connectivity, and optionally morphometric features,
- Searching for neurons that display a similar connectivity pattern,
- Standard evaluation functions for comparison of embedding algorithms on clustering and classification tasks.

### NeuroSynapsis: high-level queries and analysis of connectomic and synaptomic data

The NeuroSynapsis Library offers a large set of utilities to accelerate the construction of circuits and analysis of connectomic and synaptomic data. It provides capabilities for

- Retrieval of neuron groups according to user-defined criteria, such as cell type, innervation pattern and connectivity, etc.,
- Retrieval of connectivity between neurons, cell types, or user-defined neuron groups, through direct or indirect connections,
- Retrieval of synapse positions and partners for groups of neurons and the capability to filter synapses by brain region, partnership or spatial location,
- Statistical analysis of retrieved synapses, such as the synaptic density in a brain region,

### NeuroGraph: connectivity pattern discovery and circuit visualization algorithms

The NeuroGraph Library offers a set of tools to discover and analyze any connectivity pattern among cell groups within a circuit. Capabilities include:

- Discovery of connectivity patterns between cell populations by automatic generation of connectivity dendrograms with different linkage criteria (such as Ward or average) (*Ward, 1963*; *Sokal, 1958*),
- Analysis of the structure of circuits by community detection algorithms such as Louvain, Leiden, Label Propogation, Girvan-Newman and Infomap,
- Analysis of neural circuit controllability, for example discovery of driver nodes (*Liu et al., 2011*),
- Comparing observed connectivity between groups of cells with models of random connectivity.

In addition, the NeuroGraph Library provides utilities to visualize the connectivity of a neural circuit to aid the creation of interactive circuit diagrams. Further capabilities include

- Force-directed layout for the architecture-level graph of the whole brain or circuit-level graph of circuits specified by NeuroSynapsis,
- Semi-automated generation of 2D circuit diagrams from specified connectome datasets either at single-neuron or cell-type scale by separating circuit components into input, local and output populations for layouting.

## NeuroWatch: 3D fruit fly data visualization in Jupyter notebooks

The NeuroWatch Library offers utilities to enable visualization of neuron morphology data using Neu3D in Jupyter Notebook cells. Capabilities include:

- Loading brain regions in 3D mesh format,
- Recoloring, rescaling, repositioning and rotating neuropils, neurons and synapses for visualization,
- Interactive alignment of new neuromorphology datasets into FlyBrainLab widgets.

## NeuroMetry: morphometric measurements of neurons

Morphometric measurements of neurons can be extracted from neuron skeleton data available in connectome datasets in .swc format. NeuroMetry provides utilities for

- Calculating morphometric measurements of neurons that are compatible with NeuroMorpho.org (*Ascoli et al., 2007*), such as total length, total surface area, total volume, maximum euclidean distance between two points, width, height, depth, average diameter, number of bifurcations and the maximum path distance,
- Accessing precomputed measurements in currently available datasets in FlyBrainLab, including FlyCircuit and Hemibrain data.

An application of the morphometric measurements is the estimation of energy consumption arising from spike generation in axon-hillocks (*Sengupta et al., 2010*).

# Appendix 3

## Libraries for analyzing, evaluating, and comparing Fruit FlyBrain circuits

The Circuit Libraries are built on top of the core FlyBrainLab architecture and provide tools for studying the functional logic of a specific or a set of distributed brain regions/circuits. The FlyBrainLab Circuit Library includes:

- **CXcircuits**: Library for Central Complex Circuits,
- **EOScircuits**: Library for Larva and Adult Early Olfactory Circuits,
- **MolTrans**: Library for Molecular Transduction in Sensory Encoding.

These are respectively described in Appendix 3.1, 3.2 and 3.3 below.

## 3.1 CXcircuits: library for central complex circuits

The CXcircuits library facilitates the exploration of neural circuits of the central complex (CX) based on the FlyCircuit dataset (*Chiang et al., 2011*). It supports the evaluation and direct comparison of the state-of-the-art of executable circuit models of the CX available in the literature, and accelerates the development of new executable CX circuit models that can be evaluated and scrutinized by the research community at large in terms of modeling assumptions and biological validity. It can be easily expanded to account for the Hemibrain dataset (*Scheffer et al., 2020*). The main capabilities of the CX Library include programs for:

- Constructing *biological* CX circuits featuring
  - A naming scheme for CX neurons that is machine parsable and facilitates the extraction of innervation patterns (*Givon et al., 2017*),
  - Algorithms for querying CX neurons in the NeuroArch database, by neuron type, by subregions they innervate, and by connectivity,
  - An inference algorithm for identifying synaptic connections between CX neurons in NeuroArch according to their innervation patterns,
- Constructing *executable* CX circuit diagrams that
  - Are interactive for CX circuits in wild-type fruit flies,
  - Interactively visualize neuron innervation patterns and circuit connectivity,
  - Are interoperable with 3D visualizations of the morphology of CX neurons,
  - Easily reconfigure CX circuits by enabling/disabling neurons/synapses, by enabling/disabling subregions in any of the CX neuropils, and by adding neurons,
  - Readily load neuron/synapse models and parameters,
- Evaluation of the executable CX circuits with
  - A common set of input stimuli, and the
  - Visualization of the execution results with a set of plotting utilities for generating raster plots and a set of animation utilities for creating videos.

## 3.2 EOScircuits library for larva and adult early olfactory circuits

The EOScircuits Library accelerates the development of models of the fruit fly early olfactory system (EOS), and facilitates structural and functional comparisons of olfactory circuits across developmental stages from larva to the adult fruit fly. Built upon FlyBrainLab's robust execution backend, the EOScircuits Library enables rapid iterative model development and comparison for Antenna (ANT), Antennal Lobe (AL) and Mushroom Body (MB) circuits across developmental stages. ANTcircuits Modeled after the first layer of the olfactory pathway, the ANTcircuits Library builds upon the Olfactory Transduction (OlfTrans) library (see Section 3.3 below) and describes interactions between odorant molecules and Olfactory Sensory Neurons (OSNs). The library provides parameterized ANT circuits, that support manipulations including

- Changing the affinity values of each of the odorant-receptor pairs characterizing the input of the Odorant Transduction Process (*Lazar and Yeh, 2020*),
- Changing parameter values of the Biological Spike Generators (BSGs) associated with each OSN (*Lazar and Yeh, 2020*),
- Changing the number of OSNs expressing the same Odorant Receptor (OR) type.

ALcircuits Modeled after the second layer of the olfactory pathway, the ALcircuits Library describes the interaction between OSNs in ANT, Projection Neurons in AL and Local Neurons in AL. The library provides parameterized AL circuits, that support manipulations including

- Changing parameter values of Biological Spike Generators (BSGs) associated with each of the Local and Projection Neurons,
- Changing the number and connectivity of Projection Neurons innervating a given AL Glomerulus,
- Changing the number and connectivity of Local Neurons in the Predictive Coding and ON-OFF circuits of the AL (*Lazar and Yeh, 2019*).

MBcircuits Modeled after the third neuropil of the olfactory pathway, the MBcircuits Library describes the expansion-and-sparsification circuit consisting of a population of Antennal Lobe Projection Neurons and Mushroom Body Kenyon Cells (KCs) (*Lazar et al., 2020a*). The library provides a parameterized MB subcircuit involving Kenyon Cells and the Anterior Posterior Lateral (APL) neuron, and supports circuit manipulations including

- Generating and changing random connectivity patterns between PNs and KCs with varying degree of fan-in ratio (number of PNs connected to a given KC),
- Changing the strength of feedback inhibition of the APL neuron.

## 3.3 MolTrans library for molecular transduction in sensory encoding

The Molecular Transduction Library accelerates the development of models of early sensory systems of the fruit fly brain by providing (1) implementations of transduction on the molecular level that accurately capture the encoding of inputs at the sensory periphery, and (2) activity data of the sensory neurons such as electrophysiology recordings for the validation of executable transduction models. The MolTrans Library includes the following packages:

- **Olfactory Transduction** (**OlfTrans**): Molecular Transduction in Olfactory Sensory Neurons,
- **Visual Transduction** (**VisTrans**): Molecular Transduction in Photoreceptors.

The capabilities of the two libraries are discussed in what follows.

### OlfTrans: Odorant Transduction in Olfactory Sensory Neurons

The OlfTrans Library (https://github.com/FlyBrainLab/OlfTrans) provides the following capabilities (see also *Lazar and Yeh, 2020*):

- Defines a model of odorant space for olfactory encoding in the adult and larva olfactory system,
- Hosts a large number of electrophysiology data of OSNs responding to different odorants with precisely controlled odorant waveforms (*Kim et al., 2011*).

Moreover, the OlfTrans Library offers

- The model of an odorant transduction process (OTP) validated by electrophysiology data and executable on Neurokernel/NeuroDriver,
- Algorithms for fitting and validation of OTP models with electrophysiology data of the Olfactory Sensory Neurons,
- Algorithms for importing odorant transduction models and data into NeuroArch and execution on Neurokernel.

The OlfTrans Library provides critical resources in the study of any subsequent stages of the olfactory system. It serves as an entry point for discovering the function of the circuits in the olfactory system of the fruit fly.

### VisTrans: PhotoTransduction in Photoreceptors

The VisTrans Library exhibits the following features and/or capabilities (see also *Lazar et al., 2015a*):

- A geometrical mapping algorithm of the visual field onto photoreceptors of the retina of the fruit fly,

- A molecular model of the phototransduction process described and biologically validated in *Song et al., 2012*,
- A parallel processing algorithm emulating the visual field by the entire fruit fly retina,
- Algorithms for importing phototransduction models into the NeuroArch Database and for program execution on the Neurokernel Execution Engine.
- Algorithms for visually evaluating photoreceptor models.

The VisTrans Library accelerates the study of the contribution of photoreceptors towards the overall spatiotemporal processing of visual scenes. It also serves as an entry point for discovering circuit function in the visual system of the fruit fly (*Lazar et al., 2020b*).

# Appendix 4

## Creating an interactive executable circuit model of the lamina cartridge

In this appendix section we walk through an example of creating an interactive executable circuit of the lamina cartridge. The starting point of the example is the connectomic data of a lamina cartridge. Through the example, we will highlight core FlyBrainLab capabilities to load data, to query and analyze data, to create interactive circuit diagram and to execute the resulting circuit.

The example here is accompanied by a Jupyter notebook available at https://github.com/Fly-BrainLab/Tutorials/blob/master/tutorials/cartridge/Cartridge.ipynb. The notebook is intended to be used inside NeuroMynerva. Running the code requires a full FlyBrainLab installation (see Code Availability and Installation), and write access to the NeuroArch server (default for a full installation).

For simplicity, start a new NeuroArch server connected to an empty database folder that will be populated with the cartridge data. After running 'start.sh' to start FlyBrainLab (see also https://github.com/FlyBrainLab/FlyBrainLab/wiki/How-to-use-FlyBrainLab-Full-Installation for instructions), run 'run_neuroarch.sh lamina lamina', where the first 'lamina' refers to the database folder and the second refers to the dataset name. Start now an NLP server using any of the named applications (such as hemibrain, flycircuit or medulla) by running 'run_nlp.sh medulla lamina'. Here 'medulla' refers to the NLP application name, and 'lamina' refers to the dataset name. The latter should match the dataset name of the NeuroArch Server. In NeuroMynerva, configure a new FlyBrainLab workspace, called 'adult (lamina)', to connect to the lamina dataset (see https://github.com/FlyBrainLab/FlyBrainLab/wiki/Installation for instructions regarding how to add new servers/datasets).

In NeuroMynerva, start a new FlyBrainLab workspace using the lamina configuration. Connect the Python kernel of the notebook to the kernel of the new FlyBrainLab workspace. The code below is ready to run in the created workspace.

```
[1]: import flybrainlab as fbl
     import flybrainlab.query as fbl_query
     import flybrainlab.circuit as circuit
     import pandas as pd
     import numpy as np
     import seaborn as sns
     %matplotlib inline
     import matplotlib.pyplot as plt
```

The following code obtains the FlyBrainLab Client object that is automatically created when launching a new workspace. It also makes sure that NeuroNLP, NeuroGFX and the client object can communicate with each other.

```
[2]: client = fbl.get_client()
     for i in fbl.widget_manager.widgets:
         if fbl.widget_manager.widgets[i].widget_id not in\
                 fbl.client_manager.clients[
                     fbl.widget_manager.widgets[i].client_id
                 ]['widgets']:
             fbl.client_manager.clients[
                 fbl.widget_manager.widgets[i].client_id
             ]['widgets'].append(
                 fbl.widget_manager.widgets[i].widget_id)
```

## 4.1 Loading the connectome datasets into the NeuroArch database

Data can be loaded into NeuroArch database from the FlyBrainLab frontend by using the Neuro-Arch_Mirror class in the query module that mirrors the high-level NeuroArch API.

```
[3]: db = fbl_query.NeuroArch_Mirror(client)
```

We first create a species of *Drosophila melanogaster*:

```
[4]: species=db.add_Species(Drosophila melanogaster,
                            stage = 'adult',
```

```
                                     sex = 'female',
                                     synonyms = [
                                             'fruit fly',
                                             'common fruit fly',
                                             'vinegar fly'
                                     ])
```

Then we create a data source under the species:

```
[5]: data_source = db.add_DataSource(
                     'cartridge',
                     version = '1.0',
                     url = 'https://doi.org/10.1016/j.cub.2011.10.022',
                     description = 'Rivera-Alba et al.,\
                             Current Biology 2011',
                     species = list(species.keys())[0])
```

Make the above data source default.

```
[6]: db.select_DataSource(list(data_source.keys())[0])
```

[FBL NA 2021-01-18 10:53:11,717] Default datasource set
Add the lamina neuropil.

```
[7]: lam = db.add_Neuropil('LAM(L)',
                     synonyms = ['left lamina'])
```

Create function to read neuron skeleton data:

```
[8]: def load_swc(file_name):
         df = pd.read_csv(file_name,
                     sep = ' ',
                     header = None,
                     comment = '#',
                     index_col = False,
                     names = [
                             'sample', 'identifier',
                             'x', 'y', 'z', 'r', 'parent'],
                     skipinitialspace = True)
         return df
     neuron_list = ['R1', 'R2', 'R3', 'R4', 'R5', 'R6',
             'L1', 'L2', 'L3', 'L4', 'L5', 'T1',
             'a1', 'a2', 'a3', 'a4', 'a5', 'a6',
             'C2', 'C3']
     swc_dir = 'swc'
```

Reading synapse data:

```
[9]: connections = pd.read_csv('connection.csv', index_col = 0)
     neuron_order = connections.columns.to_list()
     adjacency = connections.to_numpy()
```

Loading Neuron data into database

```
[10]: for neuron in neuron_list:
          df = load_swc('{}/{}.swc'.format(swc_dir, neuron))
          morphology = {'x': (df['x']*0.04).tolist(),
                  'y': (df['y']*0.04).tolist(),
                  'z': (df['z']*0.04).tolist(),
                  'r': (df['r']*0.04).tolist(),
                  'parent': df['parent'].tolist(),
                  'identifier': [0]*(len(df['x'])),
                  'sample': df['sample'].tolist(),
                  'type': 'swc'}
          arborization = []
```

```
arborization.append(
    {'type': 'neuropil',
    'dendrites': {
        'LAM(L)': int(connections.loc[neuron].sum())},
    'axons': {
        'LAM(L)': int(connections[neuron].sum())}
    })
db.add_Neuron(neuron, # uname
        neuron, # name
        referenceId = neuron, #referenceId
        morphology = morphology,
        arborization = arborization)
```

Loading synapse data into database

```
[11]: for post_ind, pre_ind in zip(*np.nonzero(adjacency)):
        pre_neuron = neuron_order[pre_ind]
        post_neuron = neuron_order[post_ind]
        db.add_Synapse(pre_neuron, post_neuron,
        int(adjacency[post_ind][pre_ind]))
```

## 4.2 Building and exploring cartridge pathways

```
[12]: # provide a wild-card regular expression to match all neuron names
      res1 = client.executeNLPquery('show all')
```

[FBL NLP 2021-01-18 10:53:34,363] NLP successfully parsed query.

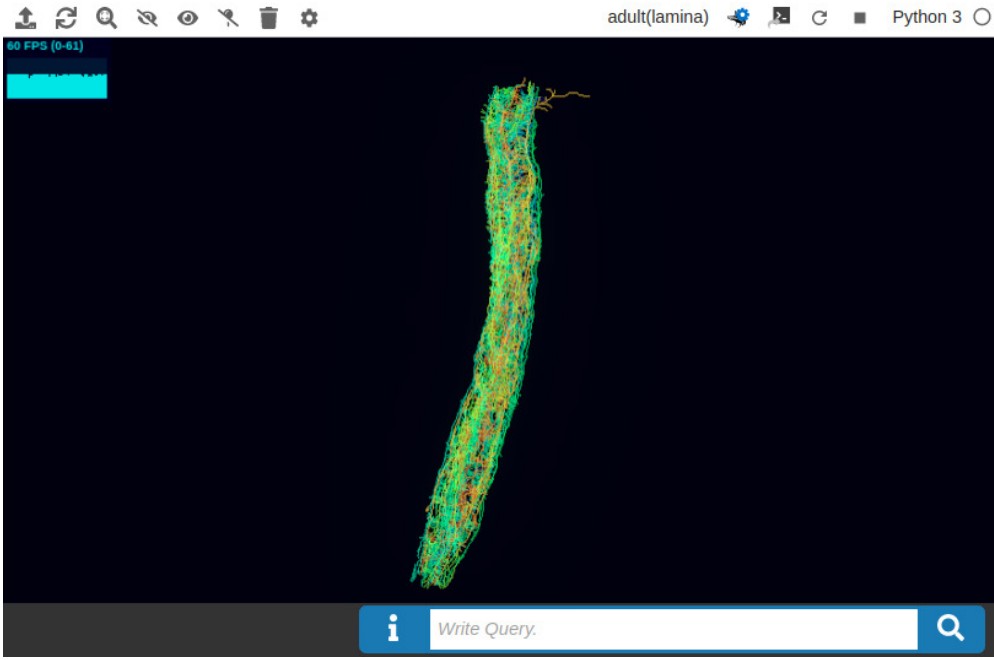

**Appendix 4—figure 1.** A lamina cartridge visualized in the NeuroNLP window.

Obtaining the connectivity matrix between neurons that are displayed in the NeuroNLP window.

```
[13]: g = client.get_neuron_graph(synapse_threshold = 0)
      M, order = g.adjacency_matrix()
      sns.heatmap(M, xticklabels = order, yticklabels = order);
```

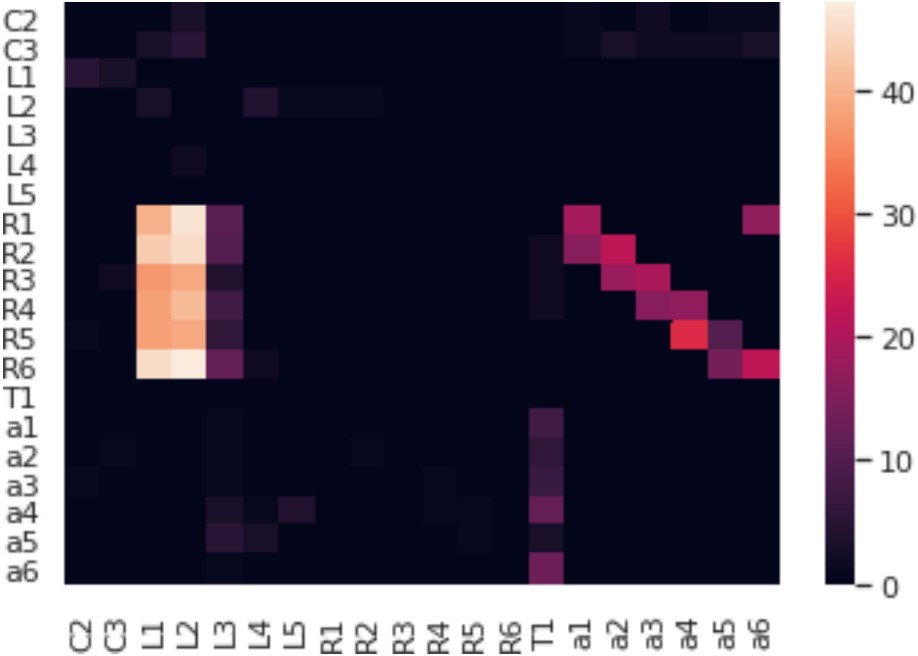

**Appendix 4—figure 2.** The connectivity matrix of the lamina cartridge.

## 4.3 Interactive exploration of the cartridge circuit diagram

Next, we interactively build an executable circuit using a circuit diagram manually created based on the connectivity above. We construct an ExecutableCircuit object from the NLP query result above. In this case, there is no model associated with these neurons yet. It initializes a new executable circuit.

```
[14]: c = circuit.ExecutableCircuit(client, res1,

                  model_name = 'cartridge', version = '1.0')
```

### Initializing a new executable circuit

We then load an SVG circuit diagram manually created and make it interactive through injecting a piece of standardized JavaScript code into the neuGFX widget. These can all be done easily using the ExecutableCircuit API. The first JavaScript file below defines the additional neuron models needed, the PhotoreceptorModel from the VisTrans Library. The second governs the interaction on the diagram in the NeuroGFX window.

```
[15]: filename = 'cartridge.svg'
      jsmodeldef = 'update_available_models.js'
      jsfilename = 'onCartridgeLoad.js'
      c.load_diagram(filename)
      c.load_js(jsmodeldef)
      c.load_js(jsfilename)
```

sending circuit configuration to GFX

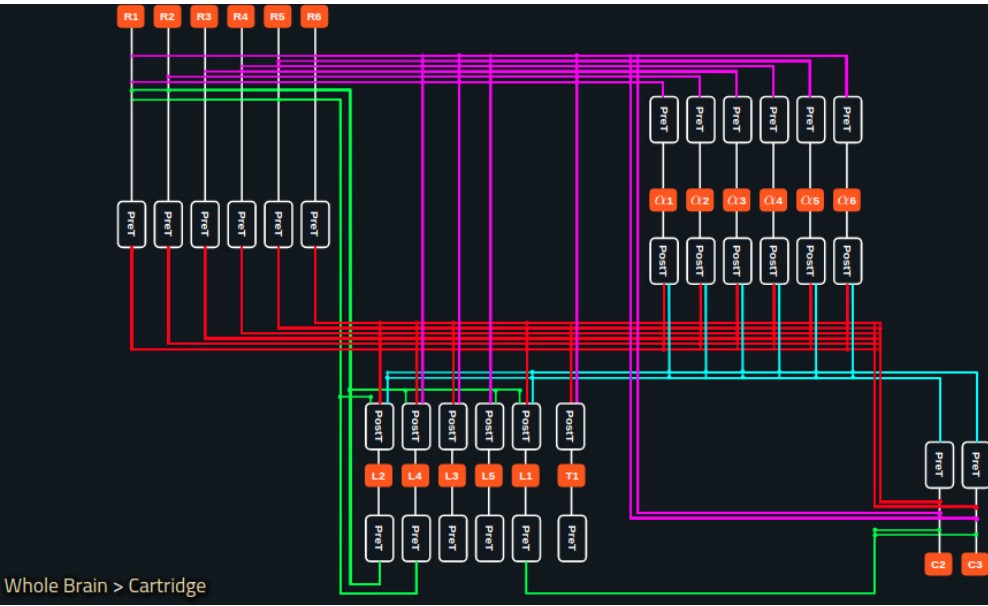

**Appendix 4—figure 3.** A circuit diagram of the lamina cartridge.

Now, we can interact with the diagram to highlight a neuron and its connected neurons, choose the model implementation for each of the cells by right clicking it, single click to silent/rescue the neuron. For example, right clicking on R1 neuron and choose PhotoreceptorModel as its model.

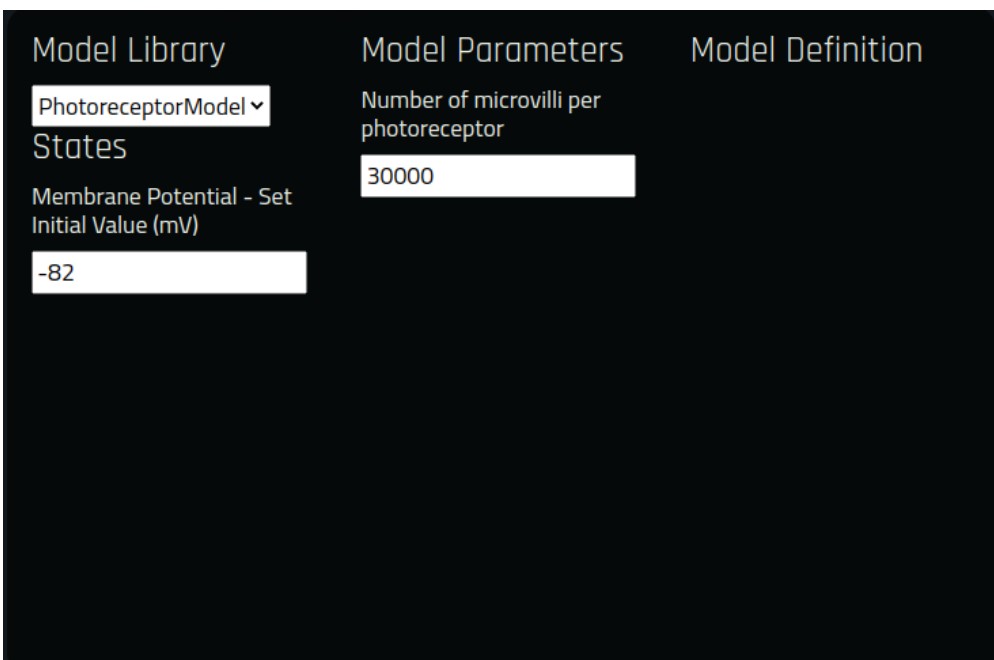

**Appendix 4—figure 4.** A screenshot of the model library in NeuroGFX for selecting the neuron model and specifying the parameters.

The same model can be populated to other photoreceptors R2-R6 by sending circuit configuration to GFX

```
[16]: c.update_model_like(['R{}'.format(i) for i in range(2,7)], 'R1')
```

Repeat this for other neurons but use a non-spiking MorrisLecar model. You can update the model and parameters on the diagram or by the code below.

```
[17]: c.update_model('L2', {'V1': -20.0,
                            'V2': 50.0,
```

```
                              'V3': -40.0,
                              'V4': 20.0,
                              'phi': 0.1,
                              'offset': 0.0,
                              'V_L': -40.,
                              'V_Ca': 80.0,
                              'V_K': -80.0,
                              'g_L': 15.0,
                              'g_Ca': 2.0,
                              'g_K': 10.,
                              'name': 'MorrisLecar'},
                    states = {'V': -46.08, 'n': 0.3525})

          c.update_model_like(['L1', 'L3', 'L4', 'L5', 'T1',
                              'C2', 'C3', 'a1', 'a2', 'a3',
                              'a4', 'a5', 'a6'],
                              'L2')
```

sending circuit configuration to GFX
sending circuit configuration to GFX

Now we get all the synapses in the circuit. In particular, the photoreceptors express the histamine neurotransmitter that is inhibitory.

```
[18]: update_models = {}
      for rid, v in c.get('Synapse').items():
          update_models[v['uname']] = {
              'params': {'name': 'SigmoidSynapse',
                    'reverse': -80.0 if \
                        v['uname'].split('-')[0][0] == 'R'\
                        else 0,
                    'threshold': -50.5,
                    'slope': 0.05,
                    'gmax': 0.04,
                    'scale': c.graph.nodes[rid]['N']
                    },
              'states': {'g': 0.0}}
      c.update_models(update_models)
```

sending circuit configuration to GFX

Finally, all the neurons and synapses have been configured. We write the executable circuit to the database with name cartridge and version 1.0.

```
[19]: c.flush_model()
```

## 4.4 Execution of the model circuit with the neurokernel execution engine

With the modeling data stored in the database, we can issue commands to execute the circuit in the Neurokernel Execution Engine. First we remove components that have been disabled.

```
[20]: res = c.remove_components()
```

We define the duration and time step of simulation

```
[21]: dur = 2.0
      dt = 1e-4
      steps = int(dur/dt)
```

We then define inputs to the circuit. Here we present a step input from time 0.5 s to 1.5 s at the light intensity equivalent to 10,000 photons per second, to the six photoreceptors. We also specify to return the inputs to the frontend.

```
[22]: input_processors = {'LAM(L)':
          [{'class': 'StepInputProcessor',
```

```
            'name': 'LAM(L)',
            'module': 'neurokernel.LPU.InputProcessors.StepInputProcessor',
            'variable': 'photon',
            'uids': [c.find_model(c.uname_to_rid['R{}'.format(i)]).
    popitem()[0] for i in range(1,7)],
            'val': 1e4,
            'start': 0.5,
            'stop': 1.5,
            'input_file': 'LAM_input.h5',
            'input_interval': 10}
            ]}
```

Next, we choose to record responses of the circuit with the 'Record' class and specify the variables and components to record. Here we request to return the membrane voltage 'V' of all neurons as specified by None in the 'uids' field.

```
[23]: output_processors = {'LAM(L)':
            [{'class': 'Record',
            'uid_dict': {'V': {'uids': None}},
            'sample_interval': 10}
            ]}
```

Execute the circuit. The execution will be queued and returned immediately to allow you to further explore the circuit or to work on other circuits while waiting for the execution result.

```
[24]: c.execute(input_processors = input_processors,
            output_processors = output_processors,
            steps = steps, dt = dt)
```

[FBL NK 2021-01-18 11:04:35,464] Execution request sent. Please wait.
[FBL NK 2021-01-18 11:04:35,464] Job received. Currently queued #1
[FBL GFX 2021-01-18 11:04:51,793] Receiving Execution Result for cartridge/1.0.
Please wait ...
[FBL GFX 2021-01-18 11:04:51,883] Received Execution Result for cartridge/1.0.
Result
stored in Client.exec result['cartridge/1.0']

A message like the above will be displayed to notify you that the result for th circuit cartridge/1.0 has been returned. Using the following method, the result will be reorganized to be referenced by the names of the neurons/synapses.

```
[25]: result = c.get_result('cartridge/1.0')
```

Plot the inputs to all photoreceptors and the response of R1 and L1 neurons.

```
[26]: client.plotExecResult('cartridge/1.0', outputs = ['R1', 'L1'])
```

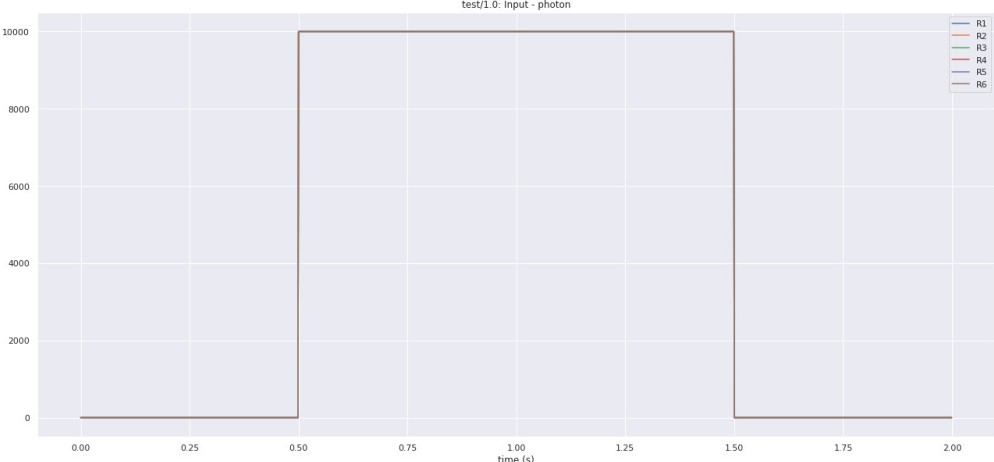

**Appendix 4—figure 5.** Inputs to the photoreceptors used during the execution of a full lamina cartridge circuit.

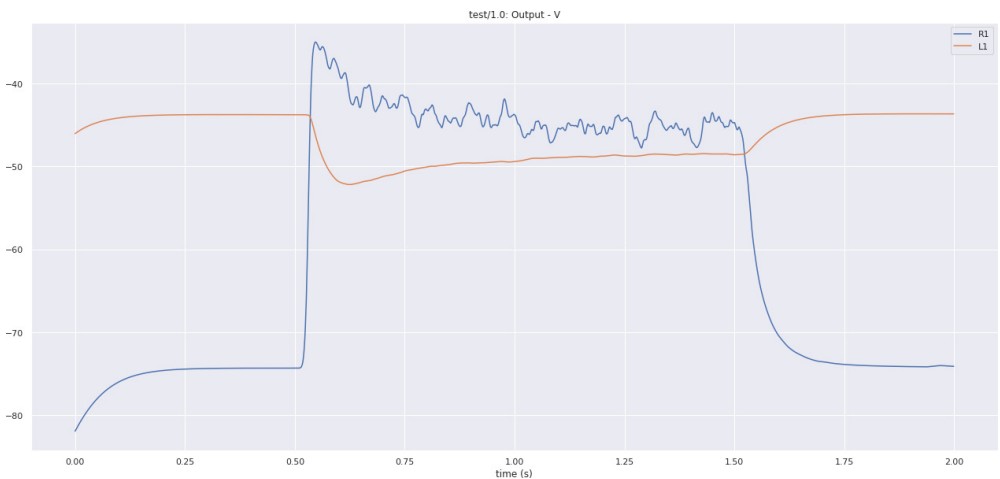

**Appendix 4—figure 6.** The output voltage of the R1 photoreceptor and L1 neuron of the lamina cartridge circuit.

## 4.5 Retrieving the executable circuit from the NeuroArch database

Now let's clear the workspace and fire the same NLP query as before.

```
[27]: res1 = client.executeNLPquery('show all')
```

[FBL NLP 2021-01-18 11:05:28,105] NLP successfully parsed query.

This time we can retrieve the executable circuit from the database that we just wrote, using the same methods as before.

```
[28]: c = circuit.ExecutableCircuit(client, res1)
```

Please select from the exisiting models to initialize the executable
circuit, or
press a to abort
0: cartridge version 1.0 (rid #457:0) 0
#457:0
Sending circuit configuration to GFX

We were asked to choose from a list of executable circuits that models the circuit displayed in NeuroNLP window. Here only one such model exists, which is the one we created in Appendix 4.3.

This time we disable R2-R6, a1-a6, L3 and T1 neurons on the circuit diagram. Equivalently, we can issue the following command: sending circuit configuration to GFX

```
[29]: c.disable_neurons(['R{}'.format(i) for i in range(2, 7)] + \
                        ['a{}'.format(i) for i in range(1, 7)] + \
                        ['L3', 'T1'])
```

sending circuit configuration to GFX

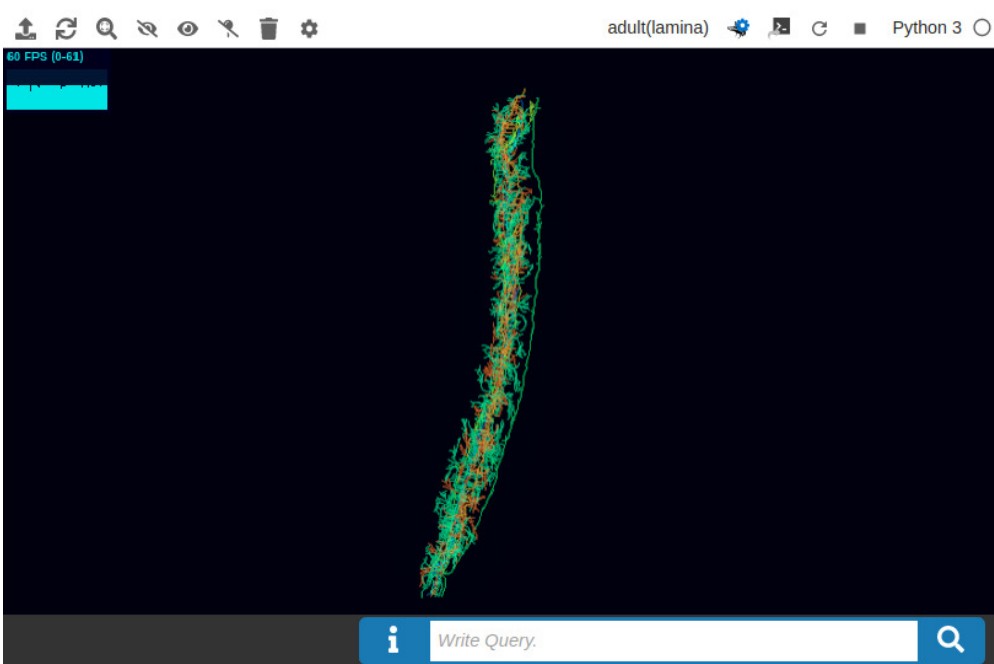

**Appendix 4—figure 7.** A lamina cartridge with several ablated neurons.

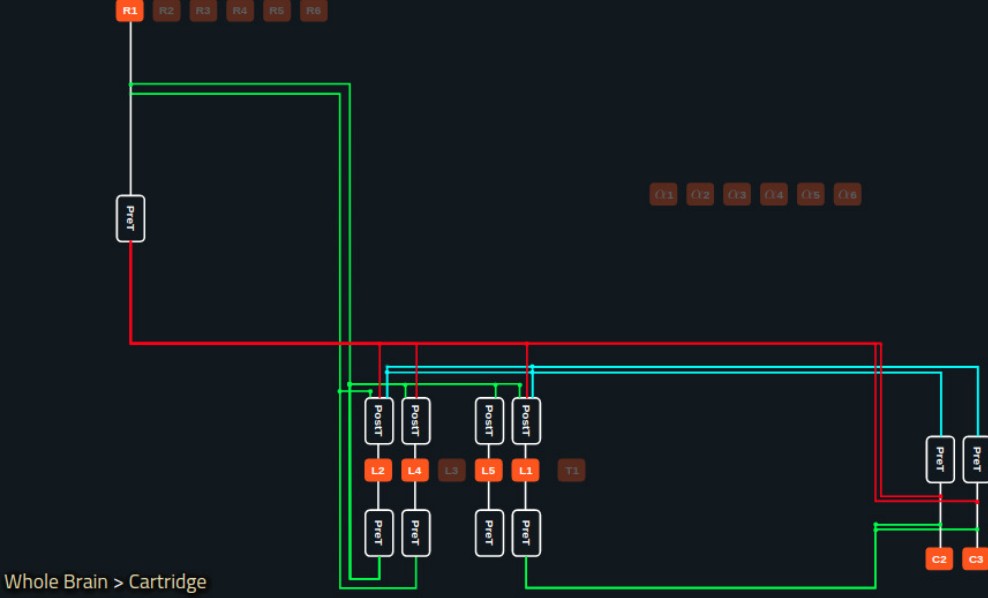

*Appendix 4—figure 8 continued on next page*

**Appendix 4—figure 8.** A reconfigured lamina cartridge obtained by disabling a number of neurons in the interactive circuit diagram.

And then reflect this change in the database before executing it.

```
[30]: res = c.remove_components()
```

```
[31]: dur = 2.0
      dt = 1e-4
      steps = int(dur/dt)

      input_processors = {'LAM(L)':
              [{'class': 'StepInputProcessor',
              'name': 'LAM(L)',
              'module': 'neurokernel.LPU.InputProcessors.StepInputProcessor',
              'variable': 'photon',
              'uids': [c.find_model(c.uname_to_rid['R1']).popitem()[0]],
              'val': 1e4,
              'start': 0.5,
              'stop': 1.5,
              'input_file': 'LAM_input.h5',
              'input_interval': 10}
              ]}
      output_processors = {'LAM(L)':
              [{'class': 'Record',
              'uid_dict': {'V': {'uids': None}},
              'sample_interval': 10}
              ]}
```

```
[32]: c.execute(input_processors = input_processors,
      output_processors = output_processors,
      steps = steps, dt = dt)
```

[FBL NK 2021-01-18 11:18:47,961] Execution request sent. Please wait.
[FBL NK 2021-01-18 11:18:47,962] Job received. Currently queued #1
[FBL GFX 2021-01-18 11:19:05,231] Receiving Execution Result for cartridge/1.0.
Please wait . . .
[FBL GFX 2021-01-18 11:19:05,280] Received Execution Result for cartridge/1.0.
Result stored in Client.exec_result['cartridge/1.0']

```
[33]: result = c.get_result('cartridge/1.0')
```

Finally, we plot the input to the R1 neuron and the responses of the R1 and L1 neurons. Here, the L1 neuron only receives input from 1 photoreceptor as compared to 6 in Appendix 4.4.

```
[34]: client.plotExecResult('cartridge/1.0', outputs = ['R1', 'L1'])
```

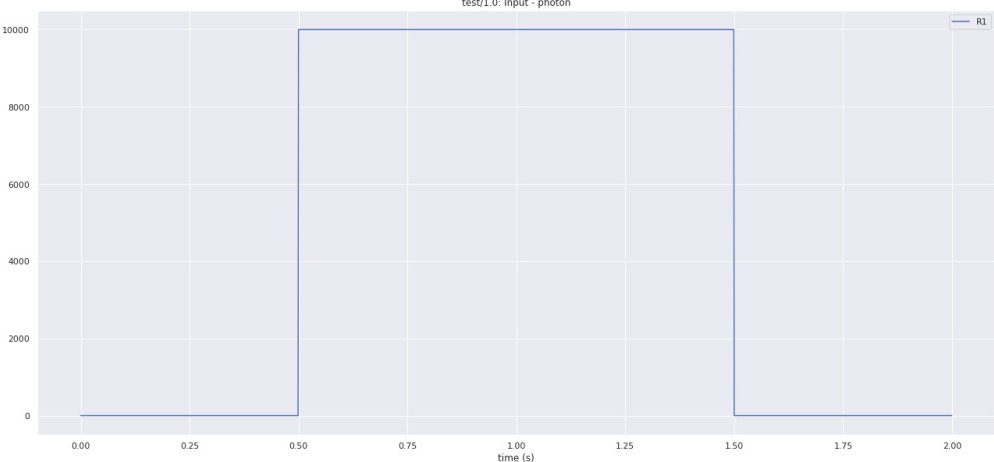

**Appendix 4—figure 9.** Input to the photoreceptor in the reconfigured lamina cartridge circuit.

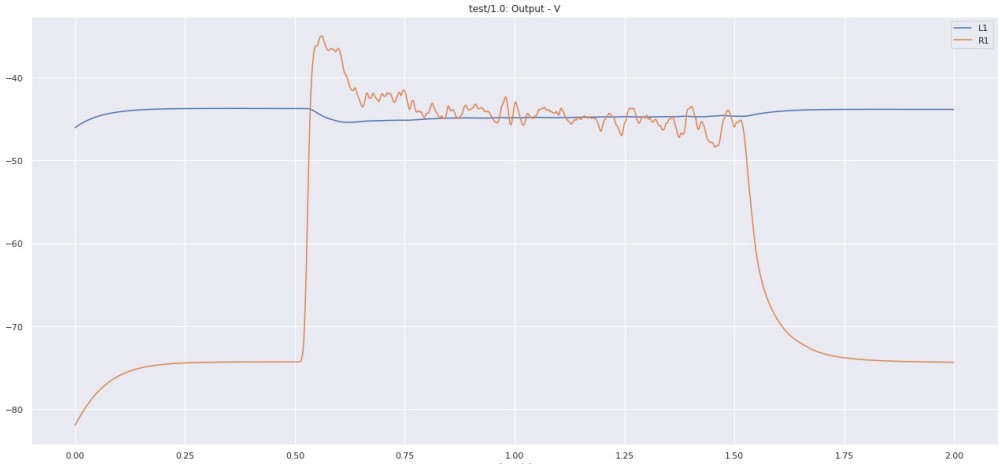

**Appendix 4—figure 10.** Voltage responses of the R1 photoreceptor and L1 neuron of the reconfigured lamina cartridge circuit.

