## [Decision Letter]

**Acceptance summary:**

FlyBrainLab is a resource for driving connectomic analyses of the *Drosophila* brain and for carrying out computational modeling based on multiple data sources. It supports 3D visualization of datasets published in the worldwide literature, and a number of libraries for integrating anatomical, sensory and physiological data with published and exploratory computational models. It will be useful for a wide range of activities, from exploring the content and intersection of datasets, to comparing circuit models in the same computational setting, to running massively parallel circuit simulations.

**Decision letter after peer review:**

Thank you for submitting your article "FlyBrainLab:Accelerating the Discovery of the Functional Logic of the *Drosophila* Brain in the Connectomic/Synaptomic Era" for consideration by *eLife*. Your article has been reviewed by three peer reviewers, and the evaluation has been overseen by a Reviewing Editor and Ronald Calabrese as the Senior Editor. The following individual involved in review of your submission has agreed to reveal their identity: Padraig Gleeson (Reviewer #1).

The reviewers have discussed the reviews with one another and the Reviewing Editor has drafted this decision to help you prepare a revised submission.

Summary:

This manuscript outlines the FlyBrainLab platform, which brings together a number of software packages from the authors to provide a unified interface for viewing data and simulating neuronal activity related to *Drosophila*. The reviewers felt that the paper had promise but there was substantial work still to be done.

Essential revisions:

1) Could the authors provide substantially more detail on how an experimentalist would use the package? It should be clear why they would want to do so.

2) The manuscript must provide transparency on the data processing and

integration.

3) The package would be far more user-friendly if it had much simpler installation. Detailed instructions would help too.

4) Users would benefit from a process to keep the packages up-to-date, such as for the “hemibrain” module.

In addition, the reviewers have provided many helpful comments to help the authors with their revision.

Reviewer #1:

This manuscript outlines the FlyBrainLab platform, which brings together a number of software packages from the authors to provide a unified interface for viewing data and simulating neuronal activity related to *Drosophila*.

The application is well described and examples of its use are given. The code for the application components is open source and installation instructions and documented are provided. The suite of components clearly work well together providing a very good example of a user focussed computational neuroscience application for working with advanced data and models.

1) While much of the technical/implementation detail is reserved for the Materials and methods section, the main body of the manuscript would benefit from a high level diagram of the structure of the application (like Supplementary figure 1, or even simpler), or a table defining/summarising the various components mentioned in the main text (NeuroMinerva/CxCircuit/NeuroArch etc.) and how they related to each other.

Reviewer #2:

FlyBrainLab by Lazar et al. provides the ability to set up, execute, and analyze *Drosophila* neural circuits, while integrating/exploring connectomics data, in a single platform. Such a unified framework has the potential to advance our understanding of the functional logic of the fly brain. The authors show that FlyBrainLab tools can be used to execute models developed previously in the literature. What is missing, however, is a clear demonstration that the platform can be used for de novo exploration and guidance on how the tools offered by the platform will enable new discoveries. In particular, the case is not made that using this library provides an easier path to discovery than the normal ad-hoc approach. The work does not fully describe what a user needs to do to deploy it for their own studies, nor does it clearly show how its own examples were generated.

1) Across the circuit examples supplied in the manuscript, it is not clear what features need to be manually coded up for the particular circuit/question of interest vs. what features can be pulled from FlyBrainLab and directly used. At present, the discussion of the different libraries in the supplement lists capabilities, but there is no guidance or examples of how the libraries can be used in practice. We could not find documentation for CXcircuits, EOScircuits, and MolTrans online. Similarly, the supplementary video illustrates the interactive capabilities of the platform, but the manuscript does not guide the user in replicating these capabilities on their own. To fix this, we advise the authors to include the notebooks used to generate all of the figures/analysis in the main results as supplementary files, with detailed annotation so that a user can use them as starting points for their own analyses.

2) More must be included in the manuscript to describe how the tool can be used for exploratory analysis. Consider including a simple annotated code walkthrough that, starting with some list of neurons, perhaps from the Hemibrain, answers what utilities are available/what code is needed to visualize neuron morphologies, what code is needed to generate an interactive circuit diagram, what code is needed to set up a simple leaky integrate and fire model, what is needed to execute a circuit, and whether resultant firing rate outputs look reasonable. The panels in Supplementary figure 3 are close, but they show the results of the above workflow, and there is no demonstration on how one can get there. Such an example need not (and perhaps is better not to) focus on a well-characterized circuit. The simple examples found in FlyBrainLab/Neuroballad are promising.

3) More work can be done to lower the barrier of entry for FlyBrainLab. Even as a researcher with a few years of Python experience that is currently using the Hemibrain to set up, run, and analyze neural circuits, I had difficulty installing FlyBrainLab and knowing what steps to take to replicate the examples shown in the manuscript. In particular, the installation instructions seem inconsistent/not fully developed on https://github.com/FlyBrainLab/FlyBrainLab. It took hours to figure out which instructions to follow to end up with a Jupyter Lab configuration that resembles the supplementary video, with a notebook, a morphology viewer, and a circuit viewer in the same window. The installation instructions within NeuroMinerva, built on JupyterLab version >2, helped get me to that point, but the instructions on FlyBrainLab, built on JupyterLab version <2, did not get me to that point. In addition, the "Starting Up FlyBrainLab" section on https://github.com/FlyBrainLab/FlyBrainLab should have material on what to do if you do not see an FFBO section or cannot run the example notebook, perhaps in some troubleshooting page.

Reviewer #3:

Lazar and colleagues present a platform, FlyBrainLab that integrates *Drosophila* neuron and circuit modelling data with neuroanatomy, from morphology to synaptic resolution information. Their desktop system is modular and stand-alone, providing the ability to query, run and visualise particular circuits and models. To demonstrate the functionality of their platform they present 3 specific examples that cover the use of published models, light and electron-microscopy (EM) data and the comparison between larva and adult.

Although the need their platform is addressing is real, the manuscript does not present the work in a compelling way, particularly for this journal's audience. Furthermore, the methods used to integrate data, and how data are used are not described properly. If a system such as this aims to become a standard analytical tool for neuroscientists, it is essential that data integration and processing are transparent.

Please find below a number of concerns. I do not comment on the technical details of the FlyBrainLab platform modules, as that is not my expertise.

1) The structure of the manuscript and the way the examples are presented are not compelling for the average neuroscientist that wants to start using the public data (models, connectome and synaptome). Especially if the one of the main draws of this type of platform is for neuroscientists to start testing models based on real data. The main reason for this is that very little information is given on how experimental data is curated and integrated (see below for more).

2) What neuroanatomical data is being used and in what way is completely opaque. It is assumed that different modalities of data will have been processed in different ways, but very little information is given in this regard. How is the light-level FlyCircuit data processed to infer connectivity and how is this process validated? How are cell types identified and validated, in FlyCircuit and the hemibrain? How many neurons and types are used for each use case?

For example, regarding the CX circuit example, the authors say, "The innervation pattern of each neuron was visually examined in the NeuroNLP window and a standard name assigned according to the naming scheme adopted in the CXcircuit Library." How do these standard names relate to the cell type names used by the community? Identifying cell types from morphological data requires expertise when this is done to the highest resolution, and thus this process should be described in detail. In addition, it becomes very difficult to assess the use cases presented when there is no clarity on what neurons and types are being used.

3) Related to the point above, the authors list the hemibrain data used is from version 1.0.1 (gs://hemibrain-release/neuprint/hemibrain_v1.0.1_neo4j_inputs.zip). However, a new version of the data (1.1) was released online in May, with the data dumps available at least from the end of June (according to https://dvid.io/blog/release-v1.1/). The latest version significantly improves the cell typing that had been released (see https://docs.google.com/document/d/1vae3ClHR8z8uekqwrOHtqiux3oY5-Y_xw6W2srCi3PI/edit?usp=sharing). The authors should update their manuscript to use the latest version of data. This should highlight issues of how data can be kept up to date in these types of platforms and how integration of versions can be achieved. The authors should comment on the processes they use for this.

4) Presenting this platform as a Resource, it becomes essential that it is easy to install. I attempted to install FlyBrainLab according to the instructions in https://github.com/FlyBrainLab/FlyBrainLab. Using miniconda on macOS, which I already had installed for other purposes, I unfortunately ran into errors, and the installation was unsuccessful (seemingly caused by msgpack not being found). The instructions mention that the platform has only been tested in Ubuntu but that it "should work" in other platforms. I understand that it is not possible to test for and avoid, all possible errors, but the authors should test the installation in at least one other OS, if they want the average neuroscientist to start using it.

The tutorials listed in https://github.com/FlyBrainLab/Tutorials are certainly a very useful introduction, although they suffer from the issues in points 2 and 3.

[Editors' note: further revisions were suggested prior to acceptance, as described below.]

Thank you for submitting your article "Accelerating with FlyBrainLab the Discovery of the Functional Logic of the *Drosophila* Brain in the Connectomic Era" for consideration by *eLife*. Your article has been overseen by a Reviewing Editor and Ronald Calabrese as the Senior Editor.

The Reviewing Editor has drafted this decision to help you prepare a revised submission.

As the editors have judged that your manuscript is of interest, but as described below that additional work is required before it is published, we would like to draw your attention to changes in our revision policy that we have made in response to COVID-19 (https://elifesciences.org/articles/57162). First, because many researchers have temporarily lost access to the labs, we will give authors as much time as they need to submit revised manuscripts. We are also offering, if you choose, to post the manuscript to bioRxiv (if it is not already there) along with this decision letter and a formal designation that the manuscript is "in revision at *eLife*". Please let us know if you would like to pursue this option. (If your work is more suitable for medRxiv, you will need to post the preprint yourself, as the mechanisms for us to do so are still in development.)

Summary:

The revised manuscript has addressed some of the technical issues, but has not addressed the core issues of readability of the manuscript , and usability of the software, by a regular fly neurobiologist. This was stated in the Essential revisions, point 1: "1. Could the authors provide substantially more detail on how an experimentalist would use the package? It should be clear why they would want to do so."

While the authors have responded with some limited explanations in the cover letter, the required changes are not evident in the manuscript, and it is there that these essential points of usability must be clarified. Again, it is not sufficient to refer the reader to the website to do this. The appendices, and much of the text, still mostly tell the reader what can be done, rather than how to do it. This should be rather early in the manuscript to motivate what follows.

Similarly, on essential point 2, the reviewers would like to know how their data goes in and is manipulated, and how they can be confident that what the program does is faithful to the original. Issues of installation, which have been presented, are relevant, but of secondary technical importance.

[Editors' note: further revisions were suggested prior to acceptance, as described below.]

Thank you for submitting your article "Accelerating with FlyBrainLab the Discovery of the Functional Logic of the *Drosophila* Brain in the Connectomic Era" for consideration by *eLife*. Your article has been reviewed by three peer reviewers, and the evaluation has been overseen by a Reviewing Editor and Ronald Calabrese as the Senior Editor. The following individuals involved in review of your submission have agreed to reveal their identity: Padraig Gleeson (Reviewer #1); Danylo Lavrentovich (Reviewer #2).

Essential Revisions:

The reviewers and I felt that the paper and FlyBrainLab provide a userful resource for the field. The revised version is considerably improved and the reviewers would like to suggest a few essential but straightforward revisions to make it even more accessible to the readers and users of this resource.

1) Update key references (indicated in the detailed reviews).

2) Clarify Figures and their legends, especially Figure 2 and 3.

There are several further important suggestions by the reviewers to strengthen the presentation and improve the accessibility of the paper and resource for readers. These are provided in the detailed reviewer comments below.

Reviewer #1:

This new version of the manuscript has a better layout and will be a more useful introduction to the application for new users. However there are still some issues with how the structure of the application is presented which may be difficult for readers.

Figure 2, especially the legend is quite minimal and there is nothing here to give a reader the key idea that this is a graphical application which a user would interact with through their browser. I suggest to move the screenshot of the application from Appendix 1—figure 2 to a panel in the main figure 2, and make sure these panels are well integrated and explained, e.g. NeuroMynerva in the top panel is what you see in the bottom. Refer the reader here to Appendix 1—figure 1 for more details (some of the colors of the blocks match between the simplified/full versions, e.g. green NeuroArch, there's no reason they all shouldn't for ease of readability).

NeuroNLP and NeuroGFX (window) are mentioned in the text without any context. These need to be shown/explained in Figure 2 and also described briefly where NeuroMynerva etc. are first defined in the Introduction. I would suggest highlighting all important component names in bold where they are first introduced so a user can go back to the definition as they are discussed later in the text.

It is strange that the actual short English language queries used for Figure 3 are not mentioned in the legend or main text. This is an important feature of the application and adding (at least some of) the sequence of commands for one of the panels (e.g. 3a, "show T4a", "color red", "add cholinergic presynaptic neurons" etc.) in another panel/table in the figure would be quite informative for readers. In the main text "(see also Materials and methods)" could be replaced with something better like: (the full sequence of queries which created this panel can be found in the Materials and methods). Also explain that the panels in Figure 3 are screenshots of the NeuroNLP window in Figure 2B, etc.

It is good having a section in the Materials and methods for each of other figures related to the main use cases/examples, but these could be tied together better also, making it clearer that the details of how the figure panels were generated can be found in the Materials and methods. Also some parts of the Materials and methods do not refer back to the figures, e.g. "Model A [26], Model B [27] and Model C [28]" could refer to Figure 6A, B, C etc. Small things like this would improve the readability of the paper significantly.

It might also be worth numbering the use cases/analysis types, e.g. Use Case 1-6, and adding these to subheadings to make it easier to move between the main text and Materials and methods.

Overall the manuscript is a good introduction to the range of features FlyBrainLab offers and is structured such that a user can see what can be accomplished, and is given some guidance how they would achieve it themselves.

Reviewer #2:

The text is clearer and more inviting for a general audience. The enumeration of capabilities in the Introduction is effective. The Results section is structured well, displaying different use cases of FlyBrainLab. The accompanying tutorials online serve as good launching points for researchers.

Thank you to the authors for the additions in the main text, the code walkthroughs in the appendices, and the improved installation instructions. The basic tutorials are simple to follow. My only suggestion on the code side is to be more verbose in the introduction to the lamina cartridge executable circuit notebook and in the limitations of the user-side-only installation.

Reviewer #3:

The revised version of the manuscript addresses many of the concerns previously reported. Thank you to the authors for providing much clearer information about the data that is ready to use in the FlyBrain Lab platform, how it can be used, installed and the components of the FlyBrainLab. There are still some corrections that are needed regarding the source of some of the datasets.

Throughout the paper, reference 4 (Xu et al., 2020) is used as the citation for the hemibrain dataset. This is a preprint that has been superseded by the publication in September 2020 of the peer-reviewed paper (Scheffer et al., 2020, https://doi.org/10.7554/*eLife*.57443). It also needs updating in GitHub (https://github.com/FlyBrainLab/Datasets#ref-1)

The reference to the larval L1EM dataset also needs correcting. For example this is given as reference 2 (Berck et al., 2016). The correct reference, as correctly shown in https://github.com/FlyBrainLab/Datasets#ref-3, is Ohyama et al., 2015 (reference 69). There might be other instances in the text that use the wrong citation.

The section added to the beginning of the Results, which includes Figure 3, provides readers with some examples on how they can start exploring the data in the platform (published datasets) using plain English queries. However, I do not think the added Figure 3 currently presents the data in a way that makes it easy for readers to link the relevant text and figure legend that describe the connectivity, to the panels. Each of the 4 examples (a-d) displays a neuron plot (left) and a connectivity matrix (right); other than reading each of the row/column names it is not possible to link the neurons plotted on the left to the data plotted on the right. Adding a colored annotation bar or even coloring the row/column names of the connectivity matrices according to the neuron plots would certainly help, or perhaps adding some clustering.

Example 2 refers to a possible direct connection between the mushroom body and the fan-shaped body ("raising the question whether the two memory centers are directly connected"). Some of the neurons directly connecting these 2 neuropils (and possible pathways for visual information in addition to reference 17), have been described already, in Li et al., 2020 (December 2020, https://doi.org/10.7554/*eLife*.62576), one of the recent papers based on the hemibrain dataset. Could the authors please rephrase?

---

## [Author Response]

Essential revisions:1) Could the authors provide substantially more detail on how an experimentalist would use the package? It should be clear why they would want to do so.

The FlyBrainLab platform has a number of capabilities that can be used by experimentalists with widely different backgrounds in computing.

a) For experimentalists with limited programming experience, the platform can be used for extensive visualization, interactive search and building of simple brain circuits of interest. The FlyBrainLab interactive capabilities only require here basic knowledge of terminology in neurobiology. The user interacts with the UI through natural language queries without the need to go through button-clicking or to learn a new, sophisticated database query language. For example, the NeuroNLP Window (see Appendix 1—figure 2) supports the construction of novel brain circuits (not just displaying individual neuron/cell types) on the morphological level of abstraction. Simple utilities enable the graphical display of connectivity diagrams (graphs). The neurons displayed can be subsequently targeted for genetic manipulation, optogenetic ablation and/or recording. As these properties are not the main focus of our manuscript, we created a notebook to guide the user through the use of English queries and illustrate its effectiveness.

b) Experimentalists with some computer background, say Python programming, have the capability to explore and analyze novel circuits as demonstrated in Figure 6. The level of computing knowledge required is akin to Matlab programming. Here, the exploration and analysis of yet unknown brain circuits may be of interest. For example, if the experimentalist has detailed morphology and/or neural activity data of a cell of interest, he/she can build with FlyBrainLab a circuit containing post and presynaptic neurons and computationally analyze the circuit connectivity and the effect of neuron ablation on its function. The same effect can often be evaluated by experimental means, say by optogenetic ablation of neurons suggested by the computational model. The same methodology can be followed through when choosing a circuit initially investigated in the literature. FlyBrainLab immediately extends the capabilities to visualize and evaluate the functionality of a chosen circuit in a larger context than previously published. In addition, the user can benefit from the Circuit Libraries written by more computationally-advanced users. Since FlyBrainLab bridges the gap between fly brain data and executable circuits, interactive computational models are much easier for an experimentalist to explore than an ad-hoc program. This is clearly reflected in the CX example depicted in Figure 3 where multiple windows synchronize morphology visualization data with circuits diagrams. Furthermore, these libraries provide an easier path for experimentalists to computationally explore the circuits under study and to validate or invalidate models using collected data. For example, users can intuitively silence neurons using an interactive circuit diagram (rather than digging into someone’s code) and compare its response to a recorded circuit where the same neuron is silenced genetically. Ultimately, the exact mechanism underlying the function of brain circuits is what is largely lacking.

c) Experimentalists with more advanced computational neuroscience background, can take full advantage of the capabilities to generate circuit diagrams, and run parallel programs with Neurokernel on GPUs. In this scenario, the experimentalist goes well beyond what is possible today on the bench. The key reason is scaling. An arbitrary number of interconnected neuropils of interest implemented by the same or different research groups can be interconnected and structurally and functionally explored, as these circuit models are also represented in the NeuroArch Database and can be easily retrieved for execution. Scaling circuits say by considering larger and larger brain regions, is clearly of paramount importance in the quest of understanding the logic of brain function. As in classical Computer Science, the question of scaling leads to deep questions of complexity. Here, the FlyBrainLab offers a platform for accelerating the discovery of the functional logic of the *Drosophila* brain something ad-hoc methods cannot deliver. By analogy, we can of course build cars in a garage, but in order to accelerate the building of cars, the car makers moved on long time ago to the assembly line.

To lower the bar of entry for systems and computational neuroscientists, we now provide a number of tutorials online at https://github.com/FlyBrainLab/Tutorials organized along the technical proficiency required. We are aware, however, that we do not/cannot accommodate the needs of all the users in the neuroscience research community interested in using the FlyBrainLab computing platform.

2) The manuscript must provide transparency on the data processing andintegration.

We would like to stress that FlyBrainLab is not positioned as a data provisioning platform, but rather a computing platform that provides utilities to visualize fly brain data and explore, analyze, evaluate and compare executable circuit models in a common environment.

To evaluate the transparency of the data integration and circuit execution capabilities presented here clearly requires that users have FlyBrainLab fully operational. We strongly recommend that if, for whatever reason, the reviewers run into installation problems, they contact us through the editors. In this context, we would like to emphasize that the complexity of FlyBrainLab platform is well beyond what has been typically attempted in the past computational neuroscience literature. The complexity involved is reminiscent to that of an operating system where processes running on a CPU (Neurokernel) interact with a database (NeuroArch) and are flexibly invoked by the UI through NeuroMynerva.

To further enhance the transparency of the computing platform, we included 2 new figures into the manuscript:

a) Figure 2 in the Introduction section gives an overview of the main components and hints at the complexity of the overall FlyBrainLab architecture.

b) Figure 6 in the Results section demonstrates the capability to effectively explore the structure and function of yet to be discovered brain circuits.

Figure 2 and Appendix 1—figure 1 and 2 help clarify the complexity of the FlyBrainLab architecture. Understanding the underlying design choices of the systems architecture of FlyBrainLab was previously detailed in papers describing the NeuroArch (Givon et al., 2015: http://dx.doi.org/10.5281/zenodo.44225) and Neurokernel (Givon and Lazar, 2016: https://doi.org/10.1371/journal.pone.0146581) components. The NeuroMynerva user-side front end, built on top of JupyterLab is new. We are avoiding listing more details here as they are too technical. They are of course available on Github.

We included now a number of tutorials to help the user take advantage of the Circuit and Utility libraries. Detailed notebooks provide users with an overview and a number of examples how to invoke the libraries. For example, we published the CXcircuits Library at https://github.com/FlyBrainLab/CXcircuits, and included notebooks for each of the three CX models analyzed in the main text. In addition, the notebooks and libraries of the other figures/results will be published once the paper is accepted. They will be listed in the “Publications and Talks” and ”Libraries” sections of the FlyBrainLab Wiki page https://github.com/FlyBrainLab/FlyBrainLab/ wiki/FlyBrainLab-Resources.

Although we do not consider the content of the datasets as being central to capabilities of FlyBrainLab as a computing tool, we welcome the use of our platform with (not instead of) other data provision platforms and have provided examples of how FlyBrainLab can be used with new datasets (https://github.com/FlyBrainLab/Tutorials/blob/ master/tutorials/swc_loading_tutorial/swc_loading.ipynb) However, in order to demonstrate the capabilities of FlyBrainLab to work with a wide range of datasets, we’ve included several publicly available datasets (listed and tracked in the FlyBrainLab Wiki) in the default FlyBrainLab package, which have been loaded into the NeuroArch database with minimal changes (apart from formatting edits for compatibility with the NeuroArch schema, which is specified in Givon et al., 2015 (http://dx.doi. org/10.5281/zenodo.44225)). Notebooks for loading these datasets into NeuroArch databases are provided in https://github.com/FlyBrainLab/datasets.

3) The package would be far more user-friendly if it had much simpler installation. Detailed instructions would help too.

We substantially improved upon and taken the installation process to the next level.

a) We corrected, and significantly expanded the installation instructions at https: //github.com/FlyBrainLab/FlyBrainLab, and added a Wiki page for troubleshooting (https://github.com/FlyBrainLab/FlyBrainLab/wiki/Troubleshooting).

b) We now provide multiple installation options for users of different expertise: 1) using a script, 2) using our docker image, and 3) using an Amazon AWS machine image. All these options typically only require a single command line for installation and a single command line for starting the application. We also clearly listed the system requirements for each installation option. Dependencies are fully described in the installation scripts.

c) We tested the installation procedure on Linux (Ubuntu and CentOS), macOS and Windows.

d) We asked a diverse range of users, including colleagues, collaborators, undergraduate students to test the installation and received positive feedback.

e) We created a number of “get started” tutorials to guide through the basic features of the system (https://github.com/FlyBrainLab/Tutorials/tree/master/ tutorials/getting_started).

f) We now provide users more information about available resources (https:// github.com/FlyBrainLab/FlyBrainLab/wiki/FlyBrainLab-Resources), including i) a list of the latest version of the components, and ii) a list of publicly available datasets that we loaded into the NeuroArch database.

4) Users would benefit from a process to keep the packages up-to-date, such as for the “hemibrain” module.

For installation purposes, we have published and will keep updating a list of the latest versions of all FlyBrainLab components https://github.com/FlyBrainLab/ FlyBrainLab/wiki/FlyBrainLab-Resources#repositories. The installation scripts provide the version number of the dependencies whenever applicable.

For datasets loaded into the NeuroArch Database, we

a) Provided a Datasets Version Tracker https://github.com/FlyBrainLab/datasets, that also includes the latest Hemibrain dataset,

b) Published the code/notebooks used to load the NeuroArch Database with different datasets, thereby helping users to load the NeuroArch Database independently of the main developers,

c) Will periodically and timely update the database, and welcome community contributions.

In addition, the reviewers have provided many helpful comments to help the authors with their revision.Reviewer #1:This manuscript outlines the FlyBrainLab platform, which brings together a number of software packages from the authors to provide a unified interface for viewing data and simulating neuronal activity related to *Drosophila*.The application is well described and examples of its use are given. The code for the application components is open source and installation instructions and documented are provided. The suite of components clearly work well together providing a very good example of a user focussed computational neuroscience application for working with advanced data and models.1) While much of the technical/implementation detail is reserved for the Materials and methods section, the main body of the manuscript would benefit from a high level diagram of the structure of the application (like Supplementary figure 1, or even simpler), or a table defining/summarising the various components mentioned in the main text (NeuroMinerva/CxCircuit/NeuroArch etc.) and how they related to each other.

Thank you for the suggestion! We added a new figure (Figure 2) to the manuscript to provide an early overview of the main FlyBrainLab components, i.e., NeuroArch, Neurokernel and NeuroMynerva. As suggested by the reviewer, Figure 2 is a simpler version of Appendix 1—figure 1. It provides an overview of the main datasets currently available in NeuroArch, and the circuit execution capabilities supported by Neurokernel. To put the FlyBrainLab components in a better context, we also added text relating the levels of abstraction underlying Figure 2 and Appendix 1—figure 1.

Reviewer #2:FlyBrainLab by Lazar et al. provides the ability to set up, execute, and analyze *Drosophila* neural circuits, while integrating/exploring connectomics data, in a single platform. Such a unified framework has the potential to advance our understanding of the functional logic of the fly brain. The authors show that FlyBrainLab tools can be used to execute models developed previously in the literature. What is missing, however, is a clear demonstration that the platform can be used for de novo exploration and guidance on how the tools offered by the platform will enable new discoveries. In particular, the case is not made that using this library provides an easier path to discovery than the normal ad-hoc approach. The work does not fully describe what a user needs to do to deploy it for their own studies, nor does it clearly show how its own examples were generated.

We thank the reviewer for the constructive comments on the manuscript. At the end of the Results section, we now present several examples and ways for exploring an unknown circuit by expanding upon the Supplementary figure 3 of the original submission.

What FlyBrainLab offers may not be an easier path to discovery, but certainly a faster one. What does a car assembly line provide? Does one still need to design a car? Yes, and it’s a difficult problem. But assembly line makes the production much faster. Similarly, FlyBrainLab provides an essential workflow for building circuits from data, visualizing circuits, comparing existing models within the same platform, interactively manipulating circuits, creating new models that can all be completed within a single environment that provides the much needed integration to accelerate discoveries. Furthermore, we have substantially expanded the tutorials https://github.com/FlyBrainLab/Tutorials, specifically https://github.com/FlyBrainLab/Tutorials/tree/master/tutorials/getting_started to get users started with all aspects of the FlyBrainLab. Notebooks for the CX example have been published, and the notebooks for the other results in the manuscript will be published once the paper is accepted for publication.

The reviewer seems to suggest that comparison between models is not a novel exploration of circuit function. We argue the opposite. Comparison of models in the literature is one of the keys for an in-depth understanding of the exact assumptions, structure and function of the published models, operating under the same setting/environment. This is how fields like machine learning, signal processing, computer vision thrive, where comparisons and being able to run every line of code is a standard practice. However, this is not the norm in systems neuroscience and computational neuroscience, with very few exceptions (see this paper https://doi.org/10.1523/JNEUROSCI.3374-12.2013 for an example). In the two examples involving early olfactory circuits, we do not just “execute models developed previously in the literature”. Rather, the models considered have been adapted to new contexts to explore the function of the circuits under consideration, either due to the more precise connectivity information brought by the Hemibrain dataset, or due to a downsizing of the larva circuit. We consider these also as de novo exploration of the functional logic of the underlying circuits.

1) Across the circuit examples supplied in the manuscript, it is not clear what features need to be manually coded up for the particular circuit/question of interest vs. what features can be pulled from FlyBrainLab and directly used. At present, the discussion of the different libraries in the supplement lists capabilities, but there is no guidance or examples of how the libraries can be used in practice. We could not find documentation for CXcircuits, EOScircuits, and MolTrans online. Similarly, the supplementary video illustrates the interactive capabilities of the platform, but the manuscript does not guide the user in replicating these capabilities on their own. To fix this, we advise the authors to include the notebooks used to generate all of the figures/analysis in the main results as supplementary files, with detailed annotation so that a user can use them as starting points for their own analyses.

We thank the reviewer for this suggestion. To address the concern raised, we published the CXcircuits Library at https://github.com/FlyBrainLab/CXcircuits, and included notebooks for each of the three CX models analyzed in the main text. The notebooks are mentioned in the revised manuscript. To avoid duplication, we chose to publish the code directly on GitHub instead of using supplementary manuscript files. In this way, the code can constantly be updated to include new features and users can benefit from the most up-to-date version. We also note that the exact commit provides access to the code for reproducing the figures/results of the manuscript.

In addition, the notebooks and libraries of the other figures/results will be published once the paper is accepted. They will be listed in the “Publications and Talks” and ”Libraries” sections of the FlyBrainLab Wiki page https://github.com/FlyBrainLab/FlyBrainLab/wiki/ FlyBrainLab-Resources.

We would like to take this opportunity to further clarify the current division of the roles taken by FlyBrainLab platform and its Libraries.

The main components of the FlyBrainLab, as described in the Appendix 1, provide core functionalities required for data storage/retrieval, visualization, user interface and code execution of brain circuits. Examples of capabilities that users can directly invoke are given below:

1) Query using plain English and 3D graphics for building and visualizing brain circuits;

2) Retrieval of connectivity of the brain circuit built/visualized;

3) User interface and API for circuit diagram interaction;

4) Specification of models for each circuit components.

5) Execution of the circuits represented/stored in the NeuroArch Database.

The Utility Libraries provide tools for 1. analyzing data that are retrieved using the core FlyBrainLab functionality, 2. creating circuit diagrams semi-automatically.

The Circuit Libraries are built on top of the core FlyBrainLab functionality and provide tools to study functions of a specific brain region/circuit. An analogy of the relation between Circuit Libraries and the FlyBrainLab platform is that between the toolboxes and core functions of Matlab. In Matlab, the former provide high-level, application-specific functionality realized with some of the core built-in functions (such as a digital signal processing toolbox). In other words, the Circuit Libraries are examples of how the features of FlyBrainLab can/should be used.

2) More must be included in the manuscript to describe how the tool can be used for exploratory analysis. Consider including a simple annotated code walkthrough that, starting with some list of neurons, perhaps from the Hemibrain, answers what utilities are available/what code is needed to visualize neuron morphologies, what code is needed to generate an interactive circuit diagram, what code is needed to set up a simple leaky integrate and fire model, what is needed to execute a circuit, and whether resultant firing rate outputs look reasonable. The panels in Supplementary figure 3 are close, but they show the results of the above workflow, and there is no demonstration on how one can get there. Such an example need not (and perhaps is better not to) focus on a well-characterized circuit. The simple examples found in FlyBrainLab/Neuroballad are promising.

Thank you for your suggestion. We substantially expanded upon the exploratory example in Supplementary figure 3 and moved it to the end of the Results section. It appears now as Figure 6 in the subsection entitled “Exploring the Structure and Function of Yet to be Discovered Brain Circuits”. Three examples are described that demonstrate the use of the Utility Libraries in the quest of novel discoveries.

Furthermore, we have substantially expanded upon the online documentation and tutorials. We have added notebooks showing examples in each of the main steps of the workflow, including:

1) How to ask questions in English to obtain a list of neurons of interest (including visualizing their morphology) and how to build upon an initial set of neurons,

2) How to generate circuit diagrams,

3) How to interact with the circuit diagram layout, build a model and execute it, obtain the result and plot the response.

We will continue to release more examples in the future.

3) More work can be done to lower the barrier of entry for FlyBrainLab. Even as a researcher with a few years of Python experience that is currently using the Hemibrain to set up, run, and analyze neural circuits, I had difficulty installing FlyBrainLab and knowing what steps to take to replicate the examples shown in the manuscript. In particular, the installation instructions seem inconsistent/not fully developed on https://github.com/FlyBrainLab/FlyBrainLab. It took hours to figure out which instructions to follow to end up with a Jupyter Lab configuration that resembles the supplementary video, with a notebook, a morphology viewer, and a circuit viewer in the same window. The installation instructions within NeuroMinerva, built on JupyterLab version >2, helped get me to that point, but the instructions on FlyBrainLab, built on JupyterLab version <2, did not get me to that point. In addition, the "Starting Up FlyBrainLab" section on https://github.com/FlyBrainLab/FlyBrainLab should have material on what to do if you do not see an FFBO section or cannot run the example notebook, perhaps in some troubleshooting page.

We thank the reviewer for the valuable feedback and apologize for the confusion. We substantially improved upon the installation process:

1) We corrected, and significantly expanded the installation instructions at https:// github.com/FlyBrainLab/FlyBrainLab, and added a Wiki page for troubleshooting (https://github.com/FlyBrainLab/FlyBrainLab/wiki/Troubleshooting).

2) We tested the installation procedure on Linux (Ubuntu and CentOS), macOS and Windows.

3) We have already published and will keep updating a Docker image that has the full FlyBrainLab installed (https://hub.docker.com/r/fruitflybrain/fbl). We also included an Amazon Machine Image to be used on the AWS EC2 service https://github.com/ FlyBrainLab/FlyBrainLab#14-amazon-machine-image. Additional images for other cloud services will be provided upon request.

4) We asked a diverse range of users, including colleagues, collaborators, undergraduate students to test the installation and received positive feedback.

Reviewer #3:Lazar and colleagues present a platform, FlyBrainLab that integrates *Drosophila* neuron and circuit modelling data with neuroanatomy, from morphology to synaptic resolution information. Their desktop system is modular and stand-alone, providing the ability to query, run and visualise particular circuits and models. To demonstrate the functionality of their platform they present 3 specific examples that cover the use of published models, light and electron-microscopy (EM) data and the comparison between larva and adult.Although the need their platform is addressing is real, the manuscript does not present the work in a compelling way, particularly for this journal's audience. Furthermore, the methods used to integrate data, and how data are used are not described properly. If a system such as this aims to become a standard analytical tool for neuroscientists, it is essential that data integration and processing are transparent.

We thank the reviewer for providing this perspective and we would like to stress that FlyBrainLab is not positioned as a data provisioning platform, but rather as a computing platform that provides utilities to visualize fly brain data and explore, analyze, evaluate and compare executable circuit models in a common environment. We’d like to clarify that we consider data curation, integration and processing as porting currently publicly available datasets into the NeuroArch database, and we fully respect the expertise of researchers generating and curating fly brain data. As such, apart from formatting changes for consistency with NeuroArch database’s schema, minimal changes are done for any data saved to and loaded from the NeuroArch database into the Neurokernel Execution Engine. Although 6 default datasets are provided as starting points for users, the FlyBrainLab platform was developed to be agnostic of the content of the underlying datasets. We welcome the use of any additional datasets that can be loaded locally on machines running FlyBrainLab without communicating with our publicly hosted data servers. We do acknowledge, however, that in addition to providing the reference to the previous NeuroArch publications, transparency in terms of how data is saved to and loaded from the NeuroArch database could be make clearer. To that end, we’ve done the following:

1) We published the code used to create the NeuroArch database, which can be used directly or indirectly and independently of us to load new datasets and update upstream datasources,

2) We have updated the latest version of NeuroArch database that provides the Hemibrain version 1.1, and

3) We published a webpage tracking versioning of NeuroArch database.

Please find below a number of concerns. I do not comment on the technical details of the FlyBrainLab platform modules, as that is not my expertise.1) The structure of the manuscript and the way the examples are presented are not compelling for the average neuroscientist that wants to start using the public data (models, connectome and synaptome). Especially if the one of the main draws of this type of platform is for neuroscientists to start testing models based on real data. The main reason for this is that very little information is given on how experimental data is curated and integrated (see below for more).

We thank the reviewer for this feedback. While, as we already mentioned, we do not curate datasets, we included additional tutorials and examples of how individual components of the FlyBrainLab platform can be used to access/visualize/manipulate data types such as neuroanatomical connectomics/synaptomics. We also note that in the example shown in Figure 4 of the Results section, the workflow employed for modifying a previous (FlyCircuit-based) model of the Antennal Lobe to reflect new public (Hemibrain-based) data is intended as an illustration of how models can be updated using more recent (connectomics/synamptomics) data. Furthermore, we would like to clarify that the purpose of the FlyBrainLab platform is to provide an integrated system to experiment with fly brain data that are publicly available worldwide. The developers of FlyBrainLab do minimal modifications to the datasets apart from porting them into the NeuroArch database using a specified schema (as specified in Givon et al., 2015 (http://dx.doi.org/10.5281/zenodo.44225)). We do acknowledge the need for clearer examples of how newly published data can be integrated into the database. For end-users who intend to use other datasets not currently provided as default in FlyBrainLab, we included additional example notebooks detailing how such datasets can be loaded into the FlyBrainLab. Finally, although not generally recommended, we note that components in NeuroMynerva (the user front-end) can be independently invoked without communicating with the database. The neuroanatomy visualizer (Neu3D-Widget), for example, can load arbitrary swc or mesh files, where the corresponding 3D data can be accessed in the associated python kernel once the files are loaded into the widget. An example notebook (link) has been added to highlight this use case as well.

2) What neuroanatomical data is being used and in what way is completely opaque. It is assumed that different modalities of data will have been processed in different ways, but very little information is given in this regard. How is the light-level FlyCircuit data processed to infer connectivity and how is this process validated? How are cell types identified and validated, in FlyCircuit and the hemibrain? How many neurons and types are used for each use case?For example, regarding the CX circuit example, the authors say, "The innervation pattern of each neuron was visually examined in the NeuroNLP window and a standard name assigned according to the naming scheme adopted in the CXcircuit Library." How do these standard names relate to the cell type names used by the community? Identifying cell types from morphological data requires expertise when this is done to the highest resolution, and thus this process should be described in detail. In addition, it becomes very difficult to assess the use cases presented when there is no clarity on what neurons and types are being used.

We believe that two types of questions are raised above by the reviewer. One relates to how the anatomical data from the original dataset are processed and stored, and the other relates to how the data is interpreted from a model’s perspective and used in modeling.

To address the first type of questions, we would like to reiterate the position we take on connectome data. We do not generate connectome data, including the morphology of the neurons and their connectivities, nor are we in the position to identify large quantities of cell types. The utility that the platform provides is the APIs to read/write the NeuroArch Database. We do not provide any additional interpretations of these datasets. For transparency, we now provide the code that we used to create NeuroArch labeled datasets from their original source (see also the response to your comment #3). For the Hemibrain dataset, the cell types and names of individual neurons have always been assigned according to the original dataset, along with a reference ID pointing to the ID used in the original dataset. For the FlyCircuit dataset, we included original neurons from FlyCircuit 1.2 (http://flycircuit.tw), and the inferred connectivity according to the algorithm published and made available to us by the authors (https://doi.org/10.3389/fninf.2018.00099 now cited in the Data Availability section). For the Larva L1EM dataset, we included detailed description of how the publicly served dataset (https://l1em.catmaid.virtualflybrain.org) is loaded into the NeuroArch Database (https://github.com/FlyBrainLab/datasets#README.md), including a CSV file that shows the mapping between original neuron labels in the raw Dataset to the labels used in FlyBrainLab. For transparency, IDs of the neurons of the original data source are always available/displayed in the Info Panel (see also Appendix 1—figure 2).

To address the second type of questions, we would like to note the following. First, even though a substantial amount of hard work has been put into annotating datasets by their original creators, often there may be missing data/labels, and some labels are simply not really useful. In order to create an executable circuit, a user may need to make additional assumptions to assign labels/names/types. One example is provided by the FlyCircuit dataset which contains no cell type information. Additional information must be brought in from the literature by a user or he/she needs to make further assumptions. Second, the naming scheme provided by the original dataset or even in the literature is not the best possible.

For example, as the reviewer pointed out, the standard naming scheme we used in the CX example is not the same as the one used by the research community (we also have to point out that there are many names used by the community for these neuron types, and depending on the researcher, the names used appear to be largely random). We have adopted a naming scheme (in the original submission) that is both human readable and easily machine-parsable. The latter property has never been the focus of naming schemes used by neurobiologists but is critical when it comes to specifying neurons for code execution. These two points highlight the need for flexibility in processing publicly available data. The FlyBrainLab provides users full access to the NeuroArch Database to update any of the neuron’s information/metadata as desired.

Finally, we would like to address the concern raised in the last sentence in the reviewer’s comment. The types of neurons in the CX example are already provided in the Materials and methods section, including in Figure 7 (in the revised manuscript). We also corrected our statement on visually examining these neurons. The visual examination was aided by the published neuron types in the paper (https://doi.org/10.1016/j.celrep.2013.04.022). For the neurons in the FlyCircuit dataset but not mentioned in the paper, we made assumptions in modeling according to the available evidence in the literature. The details are omitted here as they are largely out of scope. In the revised manuscript, however, we added further information on each individual neuron modeled. Similarly, the types of neurons and the number of neurons of each type is explicitly mentioned in the two examples of the early olfactory system.

3) Related to the point above, the authors list the hemibrain data used is from version 1.0.1 (gs://hemibrain-release/neuprint/hemibrain_v1.0.1_neo4j_inputs.zip). However, a new version of the data (1.1) was released online in May, with the data dumps available at least from the end of June (according to https://dvid.io/blog/release-v1.1/). The latest version significantly improves the cell typing that had been released (see https://docs.google.com/document/d/1vae3ClHR8z8uekqwrOHtqiux3oY5-Y_xw6W2srCi3PI/edit?usp=sharing). The authors should update their manuscript to use the latest version of data. This should highlight issues of how data can be kept up to date in these types of platforms and how integration of versions can be achieved. The authors should comment on the processes they use for this.

We thank the reviewer for this comment. Again, as mentioned in the responses to earlier questions, the main purpose of this manuscript is to describe the FlyBrainLab as a platform of which the NeuroArch database, not an individual dataset, is a critical component. Incidentally, we presented a usage case in which the Hemibrain dataset version 1.0.1 is the main source of data. This should bear no difference in showcasing the capabilities of the FlyBrainLab than using version 1.1. Therefore, we assert that there is no need to update the database in the example we presented in the paper.

To benefit the community, however, we do feel the need to periodically and in a timely fashion update the database for the purpose of general usage. We did the following: (1) we have updated the latest version of the NeuroArch database and currently provide the Hemibrain version 1.1, (2) we published a webpage tracking versions of the NeuroArch database, and 3) we published the code used to create the NeuroArch Database. The code can be used directly or indirectly and independently of us once an update of the upstream datasource is available.

4) Presenting this platform as a Resource, it becomes essential that it is easy to install. I attempted to install FlyBrainLab according to the instructions in https://github.com/FlyBrainLab/FlyBrainLab. Using miniconda on macOS, which I already had installed for other purposes, I unfortunately ran into errors, and the installation was unsuccessful (seemingly caused by msgpack not being found). The instructions mention that the platform has only been tested in Ubuntu but that it "should work" in other platforms. I understand that it is not possible to test for and avoid, all possible errors, but the authors should test the installation in at least one other OS, if they want the average neuroscientist to start using it.The tutorials listed in https://github.com/FlyBrainLab/Tutorials are certainly a very useful introduction, although they suffer from the issues in points 2 and 3.

We thank the reviewer for the valuable comment. To address the concern, we did the following:

1) We significantly expanded the installation instruction on https://github.com/FlyBrainLab/FlyBrainLab, added a Wiki page for troubleshooting

(https://github.com/FlyBrainLab/FlyBrainLab/wiki/Troubleshooting), and mentioned that the Issue Trackers on GitHub can be useful in this case (we also understand that the reviewer needs the remain anonymous).

2) We tested the installation procedure on Linux (Ubuntu and CentOS), macOS and Windows.

3) We have already published and will keep updating a Docker image that has the full FlyBrainLab installed. We also provide an Amazon Machine Image to be used on the AWS EC2 service. Additional images on other services can be provided if requested.

4) We asked a diverse range of users, including colleagues, collaborators, undergraduate students to test the installation and received positive feedback.

[Editors' note: further revisions were suggested prior to acceptance, as described below.]

The revised manuscript has addressed some of the technical issues, but has not addressed the core issues of readability of the manuscript , and usability of the software, by a regular fly neurobiologist. This was stated in the essential revisions, point 1: "1. Could the authors provide substantially more detail on how an experimentalist would use the package? It should be clear why they would want to do so."While the authors have responded with some limited explanations in the cover letter, the required changes are not evident in the manuscript, and it is there that these essential points of usability must be clarified. Again, it is not sufficient to refer the reader to the website to do this. The appendices, and much of the text, still mostly tell the reader what can be done, rather than how to do it. This should be rather early in the manuscript to motivate what follows.

Thank you for the clarification. In this revision, we substantially expanded the text to address the readability issue of the manuscript and how an experimentalist can use the platform. We added in the Results section two more examples showing the type of questions FlyBrainLab can be used effectively to answer questions raised by a regular fly neurobiologist, and in the Materials and methods section the steps needed to answer these with FlyBrainLab.

Specifically, we added in the Results section the entry “Building Fly Brain Circuits with English Queries”, showing the versatility of the English query interface of FlyBrainLab, also technically known as NeuroNLP. NeuroNLP enables users, without any programming knowledge, to perform complex queries to build, visualize and explore biological circuits, a capability that none of the current data provisioning services is designed to do or can provide to neurobiologists/neuroscientists. In the corresponding part in the Materials and methods section, the English queries employed are listed in full detail.

We also moved a part of the text previously included in the supplement/appendix to the new entry entitled “Exploring the Structure and Function of Yet to be Discovered Brain Circuits” in the Results section. Here, we provided several examples on how to analyze connectome/synaptome datasets using FlyBrainLab to identify structures and cell types, and create circuit diagrams modeling brain pathways. We describe the steps to achieve these results in the corresponding part of the Materials and methods section.

In the newly added entry “Interactive Exploration of Executable Fruit Fly Brain Circuits” in the Results section, we present the construction of an interactive circuit diagram for rapidly developing circuit models. The capability to remove or reenable a neuron in the circuit diagram is akin to, respectively, silencing and rescuing neurons in an experiment, and is of particular interest to systems neurobiologists. Such a capability quickly enable the exploration of biological findings by means of computational models, and it is highly flexible and scalable beyond the typical experimental settings.

Finally, we added Appendix 4 with a walk through of code highlighting the main capabilities of the FlyBrainLab regarding model creation and circuit execution, including: (1) loading the NeuroArch Database from connectome datasets, (2) building and exploring biological circuits, (3) interactively exploring circuit diagrams, (4) execution of circuits retrieved from the NeuroArch Database.

Concluding, the manuscript now comprehensively describes how neurobiologists and computational neuroscientists alike can leverage the power of FlyBrainLab, whether they want to simply visualize neural circuit through complex English queries, analyze the connectivity data, or construct executable circuit models for exploration, analysis, comparison and evaluation.

Similarly, on essential point 2, the reviewers would like to know how their data goes in and is manipulated, and how they can be confident that what the progam does is faithful to the original. Issues of installation, which have been presented, are relevant, but of secondary technical importance.

Thank you for the clarification. We added the entry “Loading Publicly Available Datasets into NeuroArch Database” in the Materials and methods section. We provided details of how each dataset is handled using the NeuroArch API for loading into the NeuroArch Database. We also provided some high-level statistics of the loaded datasets. The scripts for loading these datasets have been published on GitHub: https://github.com/FlyBrainLab/ Datasets. In addition, in Appendix 4, we now provide a complete walk through some of the core FlyBrainLab capabilities with a simple example, including data loading. Some basic usages of data loading are exemplified.

[Editors' note: further revisions were suggested prior to acceptance, as described below.]

Essential Revisions:The reviewers and I felt that the paper and FlyBrainLab provide a userful resource for the field. The revised version is considerably improved and the reviewers would like to suggest a few essential but straightforward revisions to make it even more accessible to the readers and users of this resource.1) Update key references (indicated in the detailed reviews).

We updated the references according to the suggestions of reviewer #3. We also checked and updated references to preprints of other peer-reviewed publications.

2) Clarify Figures and their legends, especially Figure 2 and 3.

We updated Figure 2 and Figure 3 according to the suggestions from the reviewers.

There are several further important suggestions by the reviewers to strengthen the presentation and improve the accessibility of the paper and resource for readers. These are provided in the detailed reviewer comments below.Reviewer #1:This new version of the manuscript has a better layout and will be a more useful introduction to the application for new users. However there are still some issues with how the structure of the application is presented which may be difficult for readers.Figure 2, especially the legend is quite minimal and there is nothing here to give a reader the key idea that this is a graphical application which a user would interact with through their browser. I suggest to move the screenshot of the application from Appendix 1—figure 2 to a panel in the main Figure 2, and make sure these panels are well integrated and explained, e.g. NeuroMynerva in the top panel is what you see in the bottom. Refer the reader here to Appendix 1—figure 1 for more details (some of the colors of the blocks match between the simplified/full versions, e.g. green NeuroArch, there's no reason they all shouldn't for ease of readability).

We updated Figure 2 as well as expanded on its caption as suggested.

NeuroNLP and NeuroGFX (window) are mentioned in the text without any context. These need to be shown/explained in Figure 2 and also described briefly where NeuroMynerva etc. are first defined in the Introduction. I would suggest highlighting all important component names in bold where they are first introduced so a user can go back to the definition as they are discussed later in the text.

As suggested, we highlighted the component names with bold font at their first instance.

It is strange that the actual short English language queries used for Figure 3 are not mentioned in the legend or main text. This is an important feature of the application and adding (at least some of) the sequence of commands for one of the panels (e.g. 3a, "show T4a", "color red", "add cholinergic presynaptic neurons" etc.) in another panel/table in the figure would be quite informative for readers. In the main text "(see also Materials and methods)" could be replaced with something better like: (the full sequence of queries which created this panel can be found in the Materials and methods). Also explain that the panels in Figure 3 are screenshots of the NeuroNLP window in Figure 2B, etc.

We added English queries in the first example (Use Case 1) and refer readers to the Materials and methods section for a full sequence of queries for the rest of the examples.

It is good having a section in the Materials and methods for each of other figures related to the main use cases/examples, but these could be tied together better also, making it clearer that the details of how the figure panels were generated can be found in the Materials and methods. Also some parts of the Materials and methods do not refer back to the figures, e.g. "Model A [26], Model B [27] and Model C [28]" could refer to Figure 6A, B, C etc. Small things like this would improve the readability of the paper significantly.It might also be worth numbering the use cases/analysis types, e.g. Use Case 1-6, and adding these to subheadings to make it easier to move between the main text and Materials and methods.

We added Use Case 1-6 to each of the subheadings in the Results as well as in the Materials and methods section.

Overall the manuscript is a good introduction to the range of features FlyBrainLab offers and is structured such that a user can see what can be accomplished, and is given some guidance how they would achieve it themselves.Reviewer #2:The text is clearer and more inviting for a general audience. The enumeration of capabilities in the Introduction is effective. The Results section is structured well, displaying different use cases of FlyBrainLab. The accompanying tutorials online serve as good launching points for researchers.Thank you to the authors for the additions in the main text, the code walkthroughs in the appendices, and the improved installation instructions. The basic tutorials are simple to follow. My only suggestion on the code side is to be more verbose in the introduction to the lamina cartridge executable circuit notebook and in the limitations of the user-side-only installation.

We thank the reviewer for the valuable comments and suggestions. We added more detailed instructions in the lamina cartridge tutorial, in particular, on the exact steps for starting the backend servers and creating a workspace so that the code can be readily executed. We further clarified the limitations of the user-side-only installation in the Code Availability and Installation section in Materials and methods .

Reviewer #3:The revised version of the manuscript addresses many of the concerns previously reported. Thank you to the authors for providing much clearer information about the data that is ready to use in the FlyBrain Lab platform, how it can be used, installed and the components of the FlyBrainLab. There are still some corrections that are needed regarding the source of some of the datasets.Throughout the paper, reference 4 (Xu et al., 2020) is used as the citation for the hemibrain dataset. This is a preprint that has been superseded by the publication in September 2020 of the peer-reviewed paper (Scheffer et al., 2020, https://doi.org/10.7554/eLife.57443). It also needs updating in GitHub (https://github.com/FlyBrainLab/Datasets#ref-1)

As requested, we updated all citations to the paper above.

The reference to the larval L1EM dataset also needs correcting. For example this is given as reference 2 (Berck et al., 2016). The correct reference, as correctly shown in https://github.com/FlyBrainLab/Datasets#ref-3, is Ohyama et al., 2015 (reference 69). There might be other instances in the text that use the wrong citation.

We corrected the citation and checked to make sure that the reference is cited correctly in the rest of the manuscript.

The section added to the beginning of the Results, which includes Figure 3, provides readers with some examples on how they can start exploring the data in the platform (published datasets) using plain English queries. However, I do not think the added Figure 3 currently presents the data in a way that makes it easy for readers to link the relevant text and figure legend that describe the connectivity, to the panels. Each of the 4 examples (a-d) displays a neuron plot (left) and a connectivity matrix (right); other than reading each of the row/column names it is not possible to link the neurons plotted on the left to the data plotted on the right. Adding a colored annotation bar or even coloring the row/column names of the connectivity matrices according to the neuron plots would certainly help, or perhaps adding some clustering.

In the new Figure 3, we matched the color of row/column neuron names in the adjacency matrix to the color of the visualized neurons. Note that in the interactive user interface, each neuron can be highlighted/addressed by their name. It is, however, not possible to reflect this feature in the printed version of the manuscript.

Example 2 refers to a possible direct connection between the mushroom body and the fan-shaped body ("raising the question whether the two memory centers are directly connected"). Some of the neurons directly connecting these 2 neuropils (and possible pathways for visual information in addition to reference 17), have been described already, in Li et al., 2020 (December 2020, https://doi.org/10.7554/eLife.62576), one of the recent papers based on the hemibrain dataset. Could the authors please rephrase?

We rephrased the paragraph and referenced the paper.